# NON-ASYMPTOTIC ANALYSIS OF EFFICIENCY IN CONFORMALIZED REGRESSION

**Yunzhen Yao**[1]    **Lie He**[2*]   **Michael Gastpar**[1]
[1]LINX, EPFL
[2]MoE Key Laboratory of Interdisciplinary Research of Computation and Economics,
  Shanghai University of Finance and Economics
{yunzhen.yao,michael.gastpar}@epfl.ch, {helie}@sufe.edu.cn

## ABSTRACT

Conformal prediction provides prediction sets with coverage guarantees. The informativeness of conformal prediction depends on its efficiency, typically quantified by the expected size of the prediction set. Prior work on the efficiency of conformalized regression commonly treats the miscoverage level $\alpha$ as a fixed constant. In this work, we establish non-asymptotic bounds on the deviation of the prediction set length from the oracle interval length for conformalized quantile and median regression trained via SGD, under mild assumptions on the data distribution. Our bounds of order $\mathcal{O}(1/\sqrt{n} + 1/(\alpha^2 n) + 1/\sqrt{m} + \exp(-\alpha^2 m))$ capture the joint dependence of efficiency on the proper training set size $n$, the calibration set size $m$, and the miscoverage level $\alpha$. The results identify phase transitions in convergence rates across different regimes of $\alpha$, offering guidance for allocating data to control excess prediction set length. Empirical results are consistent with our theoretical findings.

## 1 INTRODUCTION

Deploying machine learning models in safety-critical domains, such as health care (Allgaier et al., 2023; Gui et al., 2024), finance (Wisniewski et al., 2020; Bastos, 2024), and autonomous systems (Lindemann et al., 2023; Ren et al., 2023), requires not only accurate predictions but also reliable uncertainty quantification. *Conformal prediction* (CP) is a principled, distribution-free framework for this purpose, equipping black-box models with prediction sets achieving *coverage guarantees* or *validity* (Vovk et al., 2005; Balasubramanian et al., 2014). Formally, given a set of data $\{(X_j, Y_j)\}_{j=1}^m$ drawn from a distribution $\mathcal{P}$ over $\mathcal{X} \times \mathcal{Y}$, for any user-specified *miscoverage level* $\alpha \in (0, 1)$ and a predictive model, conformal prediction constructs a set-valued function $\mathcal{C} : \mathcal{X} \to 2^{\mathcal{Y}}$ such that, for a test pair $(X_{m+1}, Y_{m+1}) \sim \mathcal{P}$, the prediction set $\mathcal{C}(X_{m+1})$ covers the label $Y_{m+1}$ with probability

$$\mathbb{P}\left[Y_{m+1} \in \mathcal{C}(X_{m+1})\right] \geq 1 - \alpha. \tag{1}$$

*Split conformal prediction* is a computationally efficient variant that incorporates training predictive models. It splits data into a *proper training set* and a calibration set; the model is first trained on the former, and its uncertainty is then quantified using the latter. During calibration, *nonconformity score functions* are constructed to measure the discrepancy between model predictions and true labels. The distribution of these scores is estimated over the calibration set, and a *quantile* of them defines a threshold. The prediction set $\mathcal{C}$ is then obtained by collecting all candidate labels whose nonconformity scores are no larger than this threshold.

A central focus of conformal prediction is *efficiency*, commonly quantified by the expected measure of the prediction set (Shafer & Vovk, 2008). For classification tasks, efficiency relates to the cardinality of the predicted label set; for regression, it corresponds to the length (or volume) of the prediction interval (or region). Under the validity condition (1), smaller prediction sets are more informative. Early works primarily evaluated efficiency empirically, whereas recent research has shifted toward *asymptotic* efficiency, demonstrating that prediction sets converge to the oracle sets as the sample

---

*Corresponding author.

size increases (Sesia & Candès, 2020; Chernozhukov et al., 2021; Izbicki et al., 2022). In contrast, *non-asymptotic* efficiency, or finite-sample guarantees on the expected measure or excess measure of the prediction set, remains much less understood, with only partial results available (Lei & Wasserman, 2014; Lei et al., 2018; Dhillon et al., 2024; Bars & Humbert, 2025). Existing non-asymptotic bounds are typically expressed based on the calibration set size $m$, whereas the effect of training set size $n$ and miscoverage level $\alpha$ remains an open question in split conformalized regression.

In this work, we analyze the efficiency of split conformal prediction in regression, focusing on *conformalized median regression* (CMR) and *conformalized quantile regression* (CQR) (Romano et al., 2019). CMR uses the absolute residual as the nonconformity score, and the quantile of the calibration residuals then determines the half-width of a symmetric prediction interval centered at the estimated conditional median. In contrast, CQR estimates both upper and lower conditional quantiles, defining nonconformity scores relative to these estimates. After calibration, CQR yields adaptive, asymmetric prediction intervals that naturally capture heteroscedasticity without assuming symmetric conditional quantiles.

**Contributions.** We present a non-asymptotic theoretical analysis of the efficiency of conformalized quantile regression and conformalized median regression under stochastic gradient descent (SGD) training. Our main contributions are as follows:

- **Finite-sample bounds for CQR.** For CQR-SGD (Algorithm 1), we derive an upper bound of order $\mathcal{O}(1/\sqrt{n} + 1/(\alpha^2 n) + 1/\sqrt{m} + \exp(-\alpha^2 m))$ on the expected deviation of the prediction set length from the oracle interval, where $n$ is the proper training set size, $m$ is the calibration set size, and $\alpha$ is the miscoverage level (Theorem 3.2). Unlike prior work that relies on assumptions on intermediate quantities, our analysis places assumptions directly on the data distribution.

- **Finite-sample bounds for CMR.** For homoscedastic tasks, CMR-SGD produces symmetric intervals of constant length across inputs, enabling us to derive a non-asymptotic upper bound of analogous order (Theorem 4.1) to CQR.

- **Theoretical guidance.** To the best of our knowledge, our work is the first analysis establishing upper bounds on interval length deviation as a function of $(n, m, \alpha)$, revealing phase transitions across different $\alpha$ regimes (Section 3.2.1). Our results thus offer guidance on allocating data between training and calibration to control excess length at a desired miscoverage level. These theoretical insights are further validated through experiments.

Finally, while our theorems are presented for models trained with SGD, the analytical framework developed in this paper is not tied to a specific optimizer: the bounds extend directly to other optimization algorithms by substituting their corresponding estimation error rates.

## 2 PRELIMINARIES

**Quantiles of random variables.** Let $F$ be the cumulative distribution function (c.d.f.) of a random variable $Z$. For $\gamma \in (0, 1)$, the $\gamma$-quantile of $Z$ is defined as

$$q_\gamma(Z) := \inf\{u \in \mathbb{R} : F(u) \geq \gamma\}.$$

**Conditional quantile function.** For $(X, Y) \sim \mathcal{P}$ over $\mathcal{X} \times \mathcal{Y}$, the conditional $\gamma$-quantile function $q_\gamma(Y \mid X) : \mathcal{X} \to \mathbb{R}$ is defined as

$$q_\gamma(Y \mid X = x) := \inf\left\{u \in \mathbb{R} : F_{Y \mid X=x}(u) \geq \gamma\right\}, \quad \text{for all } x \in \mathcal{X}. \tag{2}$$

**Split conformal prediction.** In split conformal prediction, the data are partitioned into the *proper training set* $\mathcal{D}_{\text{train}}$ and the *calibration set* $\mathcal{D}_{\text{cal}}$. The training set is first used to train a model $h$. With the trained model $h$, a nonconformity score function $\psi_h : \mathcal{X} \times \mathcal{Y} \to \mathbb{R}$ is then defined to quantify the discrepancy between a candidate label $y$ and the input $x$, where higher scores indicate worse conformity. The nonconformity scores $S_m := \{\psi_h(x_j, y_j)\}_{j=1}^m$ are computed for all calibration samples in $\mathcal{D}_{\text{cal}} = \{(x_j, y_j)\}_{j=1}^m$. The sample quantile $\hat{q}_{(1-\alpha)_m}$ is calculated at level:

$$(1 - \alpha)_m := \lceil (1 - \alpha)(m + 1) \rceil / m,$$

corresponding to the $\lceil (1-\alpha)(m+1) \rceil$-th smallest value in $S_m$, which is also known as the empirical quantile. The prediction set for a new input $x$ is then defined as

$$\mathcal{C}(x) \;=\; \{\, y \in \mathcal{Y} : \psi_h(x,y) \leq \hat{q}_{(1-\alpha)_m} \,\}.$$

**Bachmann–Landau notation.** We employ Bachmann–Landau (or Big O) notation in the limit as $n, m \to \infty$. For positive sequences or functions $f, g$, we write $f = O(g)$ if there exists $C, N > 0$ such that $|f(k)| \leq C\,|g(k)|$ for all $k \geq N$; we write $f = \Omega(g)$ if there exists $c, N > 0$ such that $|f(k)| \geq c\,|g(k)|$ for all $k \geq N$. We write $f = o(g)$ if $f/g \to 0$, and $f = \omega(g)$ if $f/g \to \infty$.

## 3 ANALYSIS OF CONFORMALIZED QUANTILE REGRESSION (CQR)

### 3.1 PROBLEM SETUP FOR CQR-SGD

**Data model.** We consider a random design setting where training, calibration, and test samples are drawn i.i.d. from an unknown distribution $\mathcal{P}$ over $\mathcal{X} \times \mathcal{Y}$. Formally, for all $i \in [n], j \in [m]$

$$(X_i^{\text{train}}, Y_i^{\text{train}}),\ (X_j^{\text{cal}}, Y_j^{\text{cal}}),\ (X^{\text{test}}, Y^{\text{test}}) \quad \text{i.i.d.} \sim \mathcal{P}.$$

We assume the covariate space $\mathcal{X} \subset \mathbb{R}^d$ is bounded: there exists a constant $B > 0$ such that

$$\|x\|_2 \leq B, \quad \forall\, x \in \mathcal{X}. \tag{3}$$

Similarly, the response space $\mathcal{Y} \subset \mathbb{R}$ is assumed to be a bounded interval $[y_{\min}, y_{\max}]$.

**Learning objective.** In CQR, the training set $\mathcal{D}_{\text{train}}$ is used to estimate the conditional $\gamma$-quantile function $q_\gamma (Y \mid X)$ defined in (2), where $\gamma = 1 - \alpha/2, \alpha/2$. The estimated function $t_\gamma(\cdot\,; \theta_n(\gamma))$ is obtained by solving the *stochastic pinball loss minimization* problem (Koenker & Bassett Jr, 1978):

$$\min_{\theta \in \Theta} \ \ell_\gamma(\theta) := \mathbb{E}_{(X,Y) \sim \mathcal{P}_{X \times Y}}\big[ L_\gamma\big(t_\gamma(X; \theta),\, Y\big) \big], \tag{4}$$

where the *pinball loss* takes the form

$$L_\gamma(t, y) = \gamma(y - t)\,\mathbf{1}\{y \geq t\} + (1 - \gamma)(t - y)\,\mathbf{1}\{y < t\}. \tag{5}$$

We consider a linear function class with a convex and compact parameter space:

$$t_\gamma(x; \theta) = \theta^\top x, \quad \theta \in \Theta \subset \mathbb{R}^d, \quad \sup_{\theta \in \Theta} \|\theta\|_2 \leq K < \infty. \tag{6}$$

Without loss of generality, we assume $K \leq \max\{|y_{\min}|, |y_{\max}|\}/B$. The linear model represents a standard setting for theoretical analysis of quantile regression (Koenker, 2005; Pan & Zhou, 2021), ensuring convexity of the objective function in (4).

**Learning algorithm.** To solve (4), we consider the *stochastic approximation* framework (Robbins & Monro, 1951), focusing on stochastic gradient descent (SGD). The $\theta$ is updated according to

$$\theta_{k+1} = \Pi_\Theta (\theta_k - \eta_k \hat{g}_k), \tag{7}$$

where $\eta_k$ is the step size, $\Pi_\Theta$ denotes the Euclidean projection onto $\Theta$, and $\hat{g}_k$ is a stochastic sub-gradient satisfying $\mathbb{E}[\hat{g}_k \mid \theta_k] = g_k$, with $g_k$ a subgradient of the population objective in (4) at $\theta_k$.

Let $\theta_n(\gamma)$ denote the parameter learned by solving (4) via SGD on the training set $\mathcal{D}_{\text{train}}$. For convenience, we introduce the shorthand notations for the learned parameters

$$\underline{\theta}_n := \theta_n(\alpha/2), \quad \bar{\theta}_n := \theta_n(1 - \alpha/2), \quad \vartheta_n := \big(\underline{\theta}_n, \bar{\theta}_n\big).$$

**Conformalized quantile regression.** CQR employs two estimated conditional quantile functions, $t_{\alpha/2}(\cdot\,; \underline{\theta}_n)$ and $t_{1-\alpha/2}(\cdot\,; \bar{\theta}_n)$. Given the learned parameters $\vartheta_n = \big(\underline{\theta}_n, \bar{\theta}_n\big)$, the score for $(X, Y)$ is

$$S(X, Y; \vartheta_n) := \max\big\{ t_{\alpha/2}(X; \underline{\theta}_n) - Y,\ Y - t_{1-\alpha/2}(X; \bar{\theta}_n) \big\}. \tag{8}$$

Thus $S > 0$ if $Y$ lies outside the interval $[t_{\alpha/2}(X; \underline{\theta}_n)), t_{1-\alpha/2}(X; \bar{\theta}_n)]$, and $S \leq 0$ otherwise. Let $S_m(\mathcal{D}_{\text{cal}}; \vartheta_n)$ denote the $m$ scores on the calibration data, and let $\hat{q}_{(1-\alpha)_m}(S_m \mid \vartheta_n)$ be their empirical $(1-\alpha)_m$-quantile, i.e., the $\lceil (1-\alpha)(m+1) \rceil$-th smallest value of $S_m(\mathcal{D}_{\text{cal}}; \vartheta_n)$. The prediction set for a test covariate $X$ is then

$$\mathcal{C}(X) = \big[ t_{\alpha/2}\big(X; \underline{\theta}_n\big) - \hat{q}_{(1-\alpha)_m}(S_m \mid \vartheta_n),\ t_{1-\alpha/2}\big(X; \bar{\theta}_n\big) + \hat{q}_{(1-\alpha)_m}(S_m \mid \vartheta_n) \big], \tag{9}$$

if $t_{1-\alpha/2}\big(X; \underline{\theta}_n\big) - t_{\alpha/2}\big(X; \bar{\theta}_n\big) + 2\hat{q}_{(1-\alpha)_m}(S_m \mid \vartheta_n) \geq 0$; otherwise, $\mathcal{C}(X) = \emptyset$.

**Remark 3.1.** *The phenomenon where the lower quantile estimate exceeds the upper quantile estimate is known as* quantile crossing *(Romano et al., 2019; Bassett Jr & Koenker, 1982). We show in the proof of Proposition A.4 that, quantile crossing does not occur with high probability once the training set size $n$ is sufficiently large. Moreover, because the covariate space $\mathcal{X}$ is bounded, the ground-truth lower and upper quantile functions cannot cross, even if they are not parallel.*

The whole pipeline of CQR with SGD training is summarized in Algorithm 1.

---

**Algorithm 1** Conformalized Quantile Regression with SGD Training (CQR-SGD)

---

1: **Input:** Dataset of size $(n + m)$, miscoverage level $\alpha$, new input $x$
2: Split the dataset into a proper training set $\mathcal{D}_{\text{train}}$ of size $n$ and a calibration set $\mathcal{D}_{\text{cal}}$ of size $m$
3: Train quantile regressors $t_{\alpha/2}(\cdot; \underline{\theta}_n)$ and $t_{1-\alpha/2}(\cdot; \bar{\theta}_n)$ on $\mathcal{D}_{\text{train}}$ by solving (4) via SGD
4: Compute $m$ nonconformity scores on $\mathcal{D}_{\text{cal}}$ according to (8)
5: $\hat{q}_{(1-\alpha)_m} \leftarrow$ the $(1 - \alpha)_m$-quantile of the scores on $\mathcal{D}_{\text{cal}}$
6: $\mathcal{C}(x) \leftarrow \left[ t_{\alpha/2}(x; \underline{\theta}_n) - \hat{q}_{(1-\alpha)_m}, t_{1-\alpha/2}(x; \bar{\theta}_n) + \hat{q}_{(1-\alpha)_m} \right]$
7: **Output:** Prediction set $\mathcal{C}(x)$ for a new input $x$

---

### 3.2 THEORETICAL RESULTS FOR EFFICIENCY OF CQR

To establish upper bounds on the expected length deviation of the prediction sets, we introduce the following assumptions.

**Assumption 3.1** (Well-specification in CQR). *For $\gamma \in \{\alpha/2, 1 - \alpha/2\}$, there exists $\theta^*(\gamma) \in \Theta$ such that*

$$q_\gamma(Y \mid X = x) = t_\gamma(x; \theta^*(\gamma)) = x^\top \theta^*(\gamma), \quad \text{for all } x \in \mathcal{X} \subset \mathbb{R}^d.$$

Assumption 3.1 ensures that $\theta^*(\gamma)$ is a minimizer of (4) (Takeuchi et al., 2006; Steinwart & Christmann, 2011).

Similar to $\underline{\theta}_n, \bar{\theta}_n$, and $\vartheta_n$, we introduce the shorthand notations for the ground-truth parameters

$$\underline{\theta}^* := \theta^*(\alpha/2), \quad \bar{\theta}^* := \theta^*(1 - \alpha/2), \quad \vartheta^* := \left( \underline{\theta}^*, \bar{\theta}^* \right).$$

**Assumption 3.2** (Bounded covariance). *There exist constants $0 < \lambda_{\min} \le \lambda_{\max} < \infty$ such that*

$$\lambda_{\min} I \preceq \Sigma := \mathbb{E}[XX^\top] \preceq \lambda_{\max} I, \tag{10}$$

*where $I$ is the identity matrix, and $A \preceq B$ means that $(B - A)$ is positive semi-definite for two symmetric matrices $A, B$.*

Note that $\lambda_{\max} \le B^2$, since $\|x\|_2 \le B$ for all $x \in \mathcal{X}$.

**Assumption 3.3** (Regularity of the conditional density). *For any $x \in \mathcal{X}$, the conditional probability density function (p.d.f.) $f_{Y|X}(\cdot \mid x)$ exists and is continuous. Moreover, there exist constants $0 < f_{\min} \le f_{\max} < \infty$ such that*

$$f_{\min} \le f_{Y|X}(y \mid x) \le f_{\max}, \quad \forall x \in \mathcal{X}, \ \forall y \in \mathcal{Y}. \tag{11}$$

We notice that Assumption 3.3 concerns only the underlying data distribution $\mathcal{P}$. In particular, our assumptions are agnostic to the induced nonconformity scores, unlike prior works which impose assumptions on the induced distribution of nonconformity scores, which depends on the trained predictive model. Assumption 3.3 is satisfied by many common continuous distributions once truncated to a bounded support and normalized, including the truncated normal distribution.

Assumption 3.3 implies that the conditional support of $Y$ given any $x \in \mathcal{X}$ is the common set $\mathcal{Y}$. The lower bound $f_{Y|X}(y \mid x) \ge f_{\min}$ guarantees that $\mathcal{Y}$ is bounded, while the upper bound $f_{Y|X}(y \mid x) \le f_{\max}$ ensures that $\mathcal{Y}$ has non-empty interior. A constant $H$ is defined to characterize the flatness of conditional distribution, i.e.

$$H(f_{\max}, f_{\min}) := f_{\max} / f_{\min}. \tag{12}$$

In particular, the Lebesgue measure of $\mathcal{Y}$ satisfies $1/f_{\max} \leq |\mathcal{Y}| \leq 1/f_{\min}$. Together with $B$ in (3), $K$ in (6), and Assumption 3.1, it yields

$$|y| \leq BK + 1/f_{\min}, \qquad \forall y \in \mathcal{Y}. \tag{13}$$

The score $S$ has a bounded support, since $|t_{1/2}(X; \theta_n)| \leq BK$ and $|Y| \leq BK + 1/f_{\min}$, i.e.,

$$|S| \leq R := 2BK + 1/f_{\min}.$$

As a first step toward bounding the expected length deviation, Theorem 3.1 establishes upper bounds on both the prediction error of the quantile regressor and the parameter estimation error under SGD training, expressed in terms of the training sample size $n$.

**Theorem 3.1** (Quantile regression error of SGD-trained models). *If Assumptions 3.1–3.3 hold, taking step size $\eta_k = 1/(\lambda_{\min} f_{\min} k)$ in SGD update (7), then*

$$\mathbb{E}_{X, \theta_n} \left[ (t_\gamma(X; \theta_n(\gamma)) - t_\gamma(X; \theta^*(\gamma)))^2 \right] \leq \frac{4\lambda_{\max}^2 f_{\max} d}{\lambda_{\min}^3 f_{\min}^2 n}, \tag{14}$$

$$\mathbb{E}_{\theta_n} \left[ \|\theta_n(\gamma) - \theta^*(\gamma)\|_2^2 \right] \leq \frac{4\lambda_{\max}^2 f_{\max} d}{\lambda_{\min}^4 f_{\min}^2 n}. \tag{15}$$

The proof of Theorem 3.1 is deferred to Appendix A.1.

**Remark 3.2.** *The results of Theorem 3.1 are established under a strongly-convex assumption as they rely on Theorem A.1 from Rakhlin et al. (2012). Comparable rates can also be obtained for non-strongly-convex objectives under the assumptions in Bach & Moulines (2013), where Assumption 3.2 can be weakened to requiring only the invertibility of $\mathbb{E}[XX^\top]$.*

Theorem 3.2 establishes a non-asymptotic efficiency guarantee for CQR-SGD (Algorithm 1), bounding the expected length deviation of the prediction set from the oracle conditional quantile interval

$$\mathcal{C}^*(X) := \left[ q_{\alpha/2}(Y \mid X), \, q_{1-\alpha/2}(Y \mid X) \right]. \tag{16}$$

We measure the efficiency of conformalized regression methods by the expected length deviation

$$\mathbb{E}_{X, \vartheta_n, \mathcal{D}_{\mathrm{cal}}} \left[ \big| |\mathcal{C}(X)| - |\mathcal{C}^*(X)| \big| \right]. \tag{expected length deviation}$$

**Theorem 3.2** (Efficiency of CQR-SGD). *For CQR-SGD, suppose Assumptions 3.1–3.3 hold. If $m > 8H/\min\{\alpha, 1 - \alpha\}$, then for test sample $(X, Y)$ and $0 < \alpha \leq 1/2$,*

$$\mathbb{E}_{X, \vartheta_n, \mathcal{D}_{\mathrm{cal}}} \left[ \big| |\mathcal{C}(X)| - |\mathcal{C}^*(X)| \big| \right] \leq \mathcal{O}\left( n^{-1/2} + (\alpha^2 n)^{-1} + m^{-1/2} + \exp(-\alpha^2 m) \right), \tag{17}$$

*where $H$ is the constant defined in (12).*

The explicit upper bound (41) and the full proof of Theorem 3.2 are presented in Appendix B, with a proof sketch illustrated in Figure 1.

**Remark 3.3.** *While Theorem 3.2 is presented for CQR trained using SGD, the analysis strategy applies to other optimization algorithms. In particular, one can replace the SGD error bound in Theorem 3.1 with that of the chosen optimizer. This replacement modifies only the terms in the overall bound that depend on the training set size $n$. Formally, suppose the upper bound in Theorem 3.1 is replaced by $\varphi_n$ where $\varphi_n \to 0$ as $n \to \infty$, then the upper bound in Theorem 3.2 becomes $\mathcal{O}\left( \varphi_n^{1/2} + \alpha^{-2}\varphi_n + m^{-1/2} + \exp(-\alpha^2 m) \right)$.*

**Remark 3.4.** *For a random variable $Z$, the density level set $\mathcal{L}(u_{1-\alpha})$ is the optimal prediction set with coverage probability $1 - \alpha$ (Lei et al., 2011), i.e.,*

$$\mathcal{L}(u_{1-\alpha}) := \{z \in \mathcal{Z} : f_Z(z) \geq u_{1-\alpha}\} = \underset{\mathbb{P}[Z \in \mathcal{C}] \geq 1-\alpha}{\arg\min} |\mathcal{C}|,$$

*where $u_{1-\alpha} = \inf\{u : \mathbb{P}[Z \in \mathcal{L}(u)] \geq 1 - \alpha\}$. The oracle interval $\mathcal{C}^*(x)$ coincides with the optimal prediction set if for any $y \in \mathcal{C}^*(x)$ and any $y' \in \mathcal{Y} \setminus \mathcal{C}^*(x)$, it holds that $f_{Y|X=x}(y) \geq f_{Y|X=x}(y')$.*

$$\mathbb{E}_{X,\vartheta_n,\mathcal{D}_{\mathrm{cal}}}\big[\big||\mathcal{C}(X)| - |\mathcal{C}^*(X)|\big|\big]$$

$$= \mathbb{E}_{X,\vartheta_n,\mathcal{D}_{\mathrm{cal}}}\Big[\Big|\big|\max\big\{t_{1-\alpha/2}\left(X;\bar{\theta}_n\right) - t_{\alpha/2}\left(X;\underline{\theta}_n\right) + 2\hat{q}_{(1-\alpha)_m}(S_m \mid \vartheta_n),\, 0\big\}\big|$$
$$- \big|\big(t_{1-\alpha/2}\left(X;\bar{\theta}^*\right) - t_{\alpha/2}\left(X;\underline{\theta}^*\right)\big)\big|\Big|\Big]$$

$$\leq \underbrace{\mathbb{E}_{X,\vartheta_n}\big[\big|t_{1-\alpha/2}\left(X;\bar{\theta}_n\right) - t_{1-\alpha/2}\left(X;\bar{\theta}^*\right)\big| + \big|t_{\alpha/2}\left(X;\underline{\theta}_n\right) - t_{\alpha/2}\left(X;\underline{\theta}^*\right)\big|\big]}$$

$$= \mathcal{O}\left(\sqrt{1/n}\right) \quad \textit{Quantile regression errors of trained model (Thm. 3.1)}$$

$$+ \qquad \underbrace{\mathbb{E}_{\vartheta_n}\big[\big|q_{1-\alpha}\left(S \mid \vartheta_n\right)\big|\big]}$$

$$= \mathcal{O}\left(\sqrt{1/n}\right) \quad \textit{Population quantile of the score (Prop. A.2)}$$

$$+ \qquad \underbrace{\mathbb{E}_{\vartheta_n}\big[\big|q_{1-\alpha}\left(S \mid \vartheta_n\right) - q_{(1-\alpha)_m}\left(S \mid \vartheta_n\right)\big|\big]}$$

$$= \mathcal{O}\left(1/m + 1/(\alpha^2 n)\right) \quad \textit{Population finite-sample score-quantile gap (Prop. A.3)}$$

$$+ \qquad \underbrace{\mathbb{E}_{\vartheta_n,\mathcal{D}_{\mathrm{cal}}}\big[\big|q_{(1-\alpha)_m}\left(S \mid \vartheta_n\right) - \hat{q}_{(1-\alpha)_m}\left(S_m \mid \vartheta_n\right)\big|\big]}$$

$$= \mathcal{O}\left(\sqrt{1/m} + \exp(-\alpha^2 m) + 1/(\alpha^2 n)\right) \quad \textit{Empirical score-quantile concentration (Prop. A.5)}$$

Figure 1: Proof outline of Theorem 3.2. Full proof deferred to Section A.

### 3.2.1 PHASE TRANSITIONS OF THE UPPER BOUND

In Theorem 3.2, the upper bound on the expected absolute deviation between the prediction set length $|\mathcal{C}(X)|$ and the oracle interval length $|\mathcal{C}^*(X)|$ is expressed explicitly as a function of the training size $n$, calibration size $m$, and miscoverage level $\alpha$. Unlike prior analyses that treat $\alpha$ as a fixed constant, our result reveals its critical role in efficiency. Specifically, the terms $(\alpha^2 n)^{-1}$ and $\exp(-\alpha^2 m)$ in the bound imply a fundamental scaling relationship as follows.

**Regimes of $\alpha$ in general cases.**

- The length deviation converges to zero whenever $\alpha$ decays slower than $n^{-1/2}$ and $m^{-1/2}$, i.e., $\alpha = \omega(\max\{n^{-1/2}, m^{-1/2}\})$. Thus, Theorem 3.2 implies that if the expected prediction set length is required to remain within a fixed tolerance of the oracle length, $\alpha$ is not supposed to be chosen arbitrarily small.

- For the two $n$-dependent terms in (17), if $\alpha = \Omega(n^{-1/4})$, then they are of order $\mathcal{O}(n^{-1/2})$; otherwise they are of order $\mathcal{O}\left((\alpha^2 n)^{-1}\right)$.

- For the two $m$-dependent terms, if $\alpha = \Omega(\sqrt{\log m/m})$, then they are of order $\mathcal{O}(m^{-1/2})$; otherwise they are of order $\mathcal{O}(\exp(-\alpha^2 m))$.

- Thus, if $\alpha = \Omega(\max\{n^{-1/4}, \sqrt{\log m/m}\})$, the upper bound scales as $\mathcal{O}(n^{-1/2} + m^{-1/2})$, which coincides with the rate in Bars & Humbert (2025) assuming a finite function class.

**Regimes of $\alpha$ when $n, m$ of the same order.** When $n = \Theta(m)$, the upper bound simplifies to $\mathcal{O}(n^{-1/2} + (\alpha^2 n)^{-1})$. Figure 2 shows it in different regimes of $\alpha = \Omega(n^{-1})$, consistent with the assumption $m > 8H/\min\{\alpha, 1-\alpha\}$ in Theorem 3.2.

**Data Allocation.** If $\alpha = \Omega(\max\{n^{-1/4}, \sqrt{\log m/m}\})$, the bound reduces to $\mathcal{O}(n^{-1/2} + m^{-1/2})$, so a natural choice is to set $n$ and $m$ to be of the same order. If $\alpha = \Omega(\sqrt{\log m/m})$ and $\alpha = \omega(n^{-1/4})$, the trade-off is between $\mathcal{O}(m^{-1/2})$ and $\mathcal{O}(1/(\alpha n^2))$, and balancing them yields $m = \Theta(\alpha^4 n^4)$.

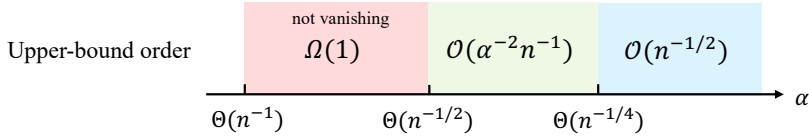

Figure 2: Upper bound orders in Theorem 3.2 in different regimes of $\alpha$ when $n = \Theta(m)$. Results in Lei et al. (2018); Bars & Humbert (2025) lie in the right most regime (blue).

## 4 ANALYSIS OF CONFORMALIZED MEDIAN REGRESSION (CMR)

### 4.1 PROBLEM SETUP FOR CMR-SGD

For conformalized median regression (CMR), we consider the same i.i.d. data model and learning algorithm (SGD) as CQR in Section 3.1.

**Learning objective.** In CMR, the training set $\mathcal{D}_{\text{train}}$ is used to estimate the conditional median function $q_{1/2}(Y \mid X)$, which is the special case for conditional $\gamma$-quantile estimation with $\gamma = 1/2$ (see (2)). The estimated conditional median function $t_{1/2}(\cdot; \theta)$ is learned by solving the minimization of the expected absolute error (stochastic pinball loss with $\gamma = 1/2$) via SGD:

$$\min_{\theta \in \Theta} \ell_{1/2}(\theta) := \mathbb{E}_{(X,Y) \sim \mathcal{P}_{X \times Y}} \left[ |t_{1/2}(X; \theta) - Y| \right]. \tag{18}$$

We adopt the same linear model class as in CQR, namely (6).

The shorthand notations for the learned parameter $\theta_n(1/2)$ and the true parameter $\theta^*(1/2)$ are:

$$\check{\theta}_n := \theta_n(1/2), \quad \check{\theta}^* := \theta^*(1/2).$$

**Conformalized median regression.** In CMR, given the trained regressor $t_{1/2}(\cdot; \check{\theta}_n)$, the nonconformity score for $(X, Y)$ is

$$S\left(X, Y; \check{\theta}_n\right) := \left| t_{1/2}(X; \check{\theta}_n) - Y \right|, \tag{19}$$

which corresponds to the absolute prediction error of the estimated conditional median $t1/2(\cdot; \check{\theta}_n)$.

For the calibration set $\mathcal{D}_{\text{cal}}$, let $S_m(\mathcal{D}_{\text{cal}}; \check{\theta}_n)$ denote the $m$ scores on calibration data, and let $\hat{q}_{(1-\alpha)_m}(S_m \mid \check{\theta}_n)$ be the empirical $(1-\alpha)_m$-quantile of $S$ given $\check{\theta}_n$, i.e., the $\lceil (1-\alpha)(m+1) \rceil$-th smallest element in $S_m(\mathcal{D}_{\text{cal}}; \check{\theta}_n)$. The prediction set for a test covariate $X$ is then

$$\mathcal{C}(X) = \left[ t_{1/2}(X; \check{\theta}_n) - \hat{q}_{(1-\alpha)_m}(S_m \mid \check{\theta}_n), \; t_{1/2}(X; \check{\theta}_n) + \hat{q}_{(1-\alpha)_m}(S_m \mid \check{\theta}_n) \right]. \tag{20}$$

### 4.2 THEORETICAL RESULTS FOR EFFICIENCY OF CMR

The well-specification assumption in CMR assumes a linear $q_{1/2}$:

**Assumption 4.1** (Well-specification in CMR). *There exists $\theta^*(1/2) \in \Theta$ such that*

$$q_{1/2}(Y \mid X = x) = t_{1/2}(x; \theta^*(1/2)), \quad \text{for all } x \in \mathcal{X}.$$

For the CMR setting, we make an additional assumption on top of Assumptions 4.1, 3.2, and 3.3:

**Assumption 4.2** (Symmetry of quantiles). *There exists $\zeta > 0$ such that for every $x \in \mathcal{X}$,*

$$q_{1-\alpha/2}(Y \mid X = x) - q_{1/2}(Y \mid X = x) = q_{1/2}(Y \mid X = x) - q_{\alpha/2}(Y \mid X = x) = \zeta. \tag{21}$$

**Remark 4.1.** *Assumption 4.2 is standard in the analysis of conformalized regression based on a single regressor, following the precedent set by Assumption A1 of Lei et al. (2018).*

**Theorem 4.1** (Efficiency of CMR). *For CMR-SGD, suppose Assumption 4.1,3.2,3.3,4.2 hold. If $m > 8H/\min\{\alpha, 1 - \alpha\}$, then for test sample $(X, Y)$ and $0 < \alpha \leq 1/2$,*

$$\mathbb{E}_{X, \vartheta_n, \mathcal{D}_{\text{cal}}} \left[ \left| |\mathcal{C}(X)| - |\mathcal{C}^*(X)| \right| \right] \leq \mathcal{O}\left( n^{-1/2} + (\alpha^2 n)^{-1} + m^{-1/2} + \exp(-\alpha^2 m) \right), \tag{22}$$

*where $H$ is the constant defined in (12).*

The explicit upper bound (42) and the full proof of Theorem 4.1 are presented in Appendix B.

## 5 RELATED WORKS

**Quantile regression.** Quantile regression has attracted significant attention since the seminal work of Koenker & Bassett Jr (1978) due to its robustness to outliers and ability to capture distributional heterogeneity. Early works derived the $\sqrt{n}$-consistency and asymptotic normality of quantile regressors in the linear model (Bassett Jr & Koenker, 1978; 1982; Portnoy & Koenker, 1989; Pollard, 1991). Other works established statistical properties under fixed designs, where covariates are treated as deterministic (He & Shao, 1996; Koenker, 2005). More recent works have shifted toward non-asymptotic analysis with convergence rate $\mathcal{O}(1/\sqrt{n})$ under random designs, where covariates are random and prediction performance on unseen data is emphasized (Steinwart & Christmann, 2011; Catoni, 2012; Hsu et al., 2014; Loh & Wainwright, 2015; Pan & Zhou, 2021; He et al., 2023; Liu et al., 2023; Sasai & Fujisawa, 2025). Median regression is a special case of quantile regression, has also been extensively studied (Chen et al., 2008). Shen et al. (2025) analyze online quantile regression with linear models trained via SGD, under regularity conditions closely related to ours, including a local lower bound on the conditional density. These methods form the basis for conformalized median regression and conformalized quantile regression (Romano et al., 2019).

**Efficiency analysis of conformal prediction.** Conformal prediction was developed to equip point predictions with confidence regions that provide finite-sample coverage guarantees (Papadopoulos et al., 2002; Vovk et al., 2005; 2009; Vovk, 2026). Research on its efficiency (Vovk et al., 2016; Gasparin & Ramdas, 2025) has evolved from early asymptotic convergence analyses, which established convergence rates toward the oracle prediction region (Chajewska et al., 2001; Li & Liu, 2008; Sadinle et al., 2019; Sesia & Candès, 2020; Chernozhukov et al., 2021; Izbicki et al., 2022), to generalization error-based bounds on expected set size Zecchin et al. (2024), and recently volume-minimization methods using data-driven norms (Sharma et al., 2023; Correia et al., 2024; Kiyani et al., 2024; Braun et al., 2025; Bars & Humbert, 2025; Gao et al., 2025; Srinivas, 2026). Relatedly, Gauthier et al. (2025) propose backward conformal prediction, which directly controls the size of prediction sets while relaxing the classical marginal coverage formulation. Complementary to marginal coverage, Duchi (2025) investigates sample-conditional coverage guarantees in split conformal prediction. We note that our analysis focuses on the i.i.d. setting; robustness under distribution shift has been studied separately, e.g., Joshi et al. (2025).

For conditional density estimation, under $\beta$-Hölder class and $\gamma$-exponent margin conditions of the conditional density, Lei & Wasserman (2014) derived minimax-optimal rates of order $\mathcal{O}((\log m/m)^{\beta/(3\beta+1)})$ when $\gamma = 1$, and showed that conditional coverage cannot generally be guaranteed in finite samples. When the quantile of $Y$ is symmetric and independent of $X$ (analogous to Assumption 4.2), Lei et al. (2018) incorporated training error into the efficiency analysis, treating $\alpha$ as a fixed constant. In contrast, our results for CQR and CMR make no assumptions on the training error and provide explicit upper bounds (41, 42) as functions of $(n, m, \alpha)$, applicable also to adaptive prediction sets.

Under the assumptions that the quantile function of the nonconformity score is locally $\beta$-Hölder continuous, and that the worst-case empirical estimation error of the function class is bounded, Bars & Humbert (2025) derived convergence rates of the order $\mathcal{O}(m^{-\beta\kappa/2} + n^{-\beta\iota/2})$ for some $0 < \iota, \kappa < 1$ when the function class is finite. In the case of $\beta = 1$, this rate matches our bound when $\alpha$ is treated as a fixed constant, namely $\mathcal{O}(m^{-1/2} + n^{-1/2})$. Different from analysis in Bars & Humbert (2025) that focuses on methods based on volume minimization, our work develops efficiency guarantees for CQR and CMR, without imposing assumptions on the score distribution induced by the trained model or on the estimation error. Instead, we demonstrate in the proof (especially Proposition B.2) that the required regularity conditions of the score are satisfied with high probability under mild assumptions on the underlying data distribution.

## 6 EXPERIMENTS

This section presents evaluations of length deviation using synthetic data to access our theoretical results. Additional synthetic experiments and real-world experiments are deferred to Appendix C and D due to space constraints. An overview of all experiments conducted in this paper can be found in Section 6.1.

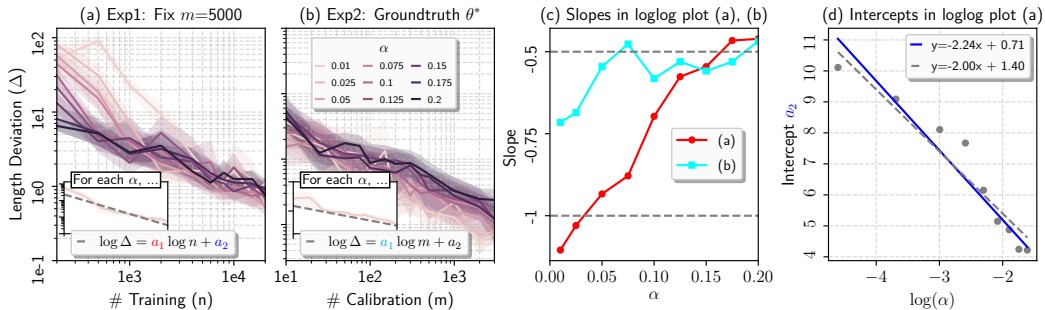

Figure 3: The length deviation of conformalized quantile regression in synthetic data experiments.

**Experiment setup.** The data generation procedure is described in Appendix C.1. All experiments employ linear models trained with SGD for one epoch using a batch size of $64$. Learning rates are selected via successive halving over the range $[10^{-5}, 1]$. We evaluate miscoverage levels $\alpha \in \{0.01, 0.025, 0.05, 0.075, 0.1, 0.125, 0.15, 0.175, 0.2\}$. Reported results are averaged over 20 independent trials, and length deviations are computed on 2000 test samples.

We denote the expected length deviation as $\Delta$. We empirically assess the upper bound of $\Delta$ in Theorem 3.2, of order $\mathcal{O}(\frac{1}{\sqrt{n}} + \frac{1}{n\alpha^2} + \frac{1}{\sqrt{m}} + \exp(-\alpha^2 m))$ from three perspectives.

- **Effect of training size** $n$. With a large calibration set ($m = 5000$), the calibration error is negligible, and the theoretical bound simplifies to $\mathcal{O}(1/\sqrt{n} + 1/(n\alpha^2))$. The theory predicts that a linear regression of $\log \Delta$ on $\log n$, i.e.,

$$\log \Delta \sim a_1 \log n + a_2, \tag{23}$$

  yields a slope $a_1$ that transitions from $-1$ to $-1/2$ as $\alpha$ increases. We confirm this trend empirically. For each $\alpha$, we train models over $n$ ranging from 200 to 20000 (Figure 3a) and fit the regression model (23) (the inset in Figure 3a shows an example) to obtain slope $a_1$ and intercept $a_2$. The resulting $(\alpha, a_1)$ pairs, shown by the red curve in Figure 3c, validate that the slope shifts from approximately $-1$ to $-1/2$ as $\alpha$ grows, reflecting the transition of the dominant term in the bound from $\mathcal{O}(1/(n\alpha^2))$ to $\mathcal{O}(1/\sqrt{n})$. The intercept $a_2$ depends on $\log \alpha$, as discussed below.

- **Effect of miscoverage level** $\alpha$. In the regime where $(n\alpha^2)^{-1}$ dominates, $\Delta$ is expected to follow a power-law scaling of order $\alpha^{-2}$. To examine this, we further regress the fitted intercepts $a_2$ in (23) on $\log \alpha$:

$$a_2 \sim b_1 \log \alpha + b_2.$$

  Together with (23), the estimated coefficient $b_1 = -2.24$ (Figure 3d) implies that $\Delta \sim \alpha^{-2.24}$. This aligns with the theoretical upper bound of order $\mathcal{O}(\alpha^{-2})$. Appendix C.2 provides an additional verification for the existence of this regime.

- **Effect of calibration size** $m$. Using the ground-truth parameter $\theta^*$, we vary the calibration set size $m$ ranging from 100 to 3000, ensuring that the resulting length deviation depends only on $m$ and $\alpha$. As illustrated in Figure 3b, the deviation decreases consistently with larger calibration sets. On a log–log scale, the slope approximately approaches $-0.5$, reflecting the increasing dominance of the $\mathcal{O}(1/\sqrt{m})$ term in the bound. Meanwhile, the exponential term $\exp(-\alpha^2 m)$ decays quickly for modest values of $m$ and becomes negligible thereafter.

## 6.1 ROADMAP OF EXPERIMENTS

We here outline the structure of all experiments conducted in the paper.

**Synthetic experiments.** Figure 3 in Section 6 and Figure 5 in Appendix C.2 assess the theoretical results developed in this paper. Appendix C.3 further examines optimization effects: Figure 6 investigates SGD with heavy-ball momentum, and Figure 7 reports the case of AdamW (Loshchilov & Hutter, 2019). In Appendix C.4, Figure 8 presents results under nonlinear conditional quantile functions. Finally, in Appendix C.5, Figures 9–10 evaluate alternative convex loss models.

**Real-world experiments.**  In Appendix D.2.2, Figure 11 presents an empirical evaluation of length deviation of CMR and CQR under different optimizers on five real-world datasets, comparing SGD, SGD with momentum, Adam, and AdamW. In Appendix D.2.3, Figure 12 evaluates non-linear models. Appendix D.3 empirically investigates data-allocation strategies in Figure 13.

## 7 LIMITATIONS, DISCUSSION, AND FUTURE WORK

**Oracle intervals may not be optimal under certain distributions.**  Our theoretical analysis shows that the prediction sets produced by CQR and CMR converge to the oracle quantile interval (16) as the training and calibration sample sizes $n$ and $m$ grow. However, the oracle interval itself is not always efficiency-optimal. It is optimal only when the condition in Remark 3.4 holds, which depends on the structure of the conditional distribution. For instance, when the conditional density is multimodal or basin-shaped, the optimal prediction set is not a single interval. In such cases, the prediction sets produced by standard conformal methods such as CMR and CQR do not approximate the optimal set. This limitation stems inherently from the standard non-conformity scores, which are restricted to producing single-interval prediction sets and therefore cannot capture complex distributional structures. One way to improve efficiency in these settings is to move beyond fixed score functions and consider parameterized nonconformity scores that adapt to the data. For instance, recent work such as Braun et al. (2025) employs an optimization-driven framework targeting volume minimization to learn the parametrization. Such approaches could potentially learn transformations that adapt to complex conditional distributions, leading to more efficient prediction sets. This is a promising direction for future research.

**Role and limitations of the linearity assumption.**  Our theoretical analysis builds on the linearity assumption of the conditional quantiles. This assumption is standard in the theoretical analysis of quantile regression (Koenker, 2005; Pan & Zhou, 2021; Shen et al., 2025), as it ensures convexity of the objective and therefore the consistency of the SGD estimator as the training data size $n$ grows. While relaxing this assumption is in principle possible, it typically requires additional assumptions on the complexity of the function class or on the estimation error bounds, which may be difficult to verify in practice (Bars & Humbert, 2025).

## 8 CONCLUSION

This paper studies the efficiency of conformalized quantile regression (CQR) and conformalized median regression (CMR) through the lens of the expected length deviation, defined as the discrepancy between the coverage-guaranteed prediction set size and the oracle interval length. Our analysis explicitly accounts for randomness introduced by training, finite-sample calibration, and test evaluation. Under mild assumptions on the data distribution, we provide, to the best of our knowledge, the first non-asymptotic convergence rate of the form: $\mathcal{O}(n^{-1/2} + n^{-1}\alpha^{-2} + m^{-1/2} + \exp(-\alpha^2 m))$, which highlights a fine-grained effect of the miscoverage level $\alpha$. Empirical results closely align with the theoretical findings.

**Ethics statement.**  This work raises no ethical concerns to declare.

**Reproducibility statement.**  For our theoretical results, we provide complete proofs in Appendix A and B. The empirical setups are detailed in Section 6, Appendix C, and Appendix D. To facilitate full reproducibility, we include the source code and datasets as part of our submission. The provided repository contains scripts to install the required Python environment, run all experiments, and generate the figures presented in the manuscript.

ACKNOWLEDGMENTS

The work in this manuscript was partially supported by the Swiss National Science Foundation under Grant 200364.

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

## A    PROOFS OF RESULTS IN CQR

To proceed, we first define some notations as follows.

$$\mathcal{E}_\gamma\left(X, \theta_n\left(\gamma\right)\right) := \left|t_\gamma\left(X; \theta_n\left(\gamma\right)\right) - t_\gamma\left(X; \theta^*\left(\gamma\right)\right)\right| \geq 0; \tag{24}$$

$$\Delta\left(X, \vartheta_n\right) := \max\left\{\mathcal{E}_{\alpha/2}\left(X, \underline{\theta}_n\right), \mathcal{E}_{1-\alpha/2}\left(X, \bar{\theta}_n\right)\right\} \geq 0; \tag{25}$$

$$S^*\left(X, Y\right) := \max\left\{t_{\alpha/2}\left(X; \underline{\theta}^*\right) - Y, Y - t_{1-\alpha/2}\left(X; \bar{\theta}^*\right)\right\} \tag{26}$$

$$= \max\left\{q_{\alpha/2}\left(Y \mid X\right) - Y, Y - q_{1-\alpha/2}\left(Y \mid X\right)\right\};$$

$$M\left(\vartheta_n\right) := \max\left\{\left\|\left(\underline{\theta}_n - \underline{\theta}^*\right)\right\|_2, \left\|\left(\bar{\theta}_n - \bar{\theta}^*\right)\right\|_2\right\}. \tag{27}$$

Let $\hat{F}_{S|\vartheta_n}^{(m)}$ denote the empirical c.d.f. from $m$ i.i.d. calibration scores given $\vartheta_n$, i.e.,

$$\hat{F}_{S|\vartheta_n}^{(m)}\left(s\right) = \frac{1}{m}\sum_{j=1}^m \mathbb{1}\{S_j \leq s\}, \qquad S_j \overset{\text{i.i.d.}}{\sim} F_{S|\vartheta_n}.$$

### A.1    PROOF OF THEOREM 3.1

**Theorem 3.1** (Quantile regression error of SGD-trained models)**.** *If Assumptions 3.1–3.3 hold, taking step size $\eta_k = 1/\left(\lambda_{\min}f_{\min}k\right)$ in SGD update (7), then*

$$\mathbb{E}_{X, \theta_n}\left[\left(t_\gamma\left(X; \theta_n\left(\gamma\right)\right) - t_\gamma\left(X; \theta^*\left(\gamma\right)\right)\right)^2\right] \leq \frac{4\lambda_{\max}^2 f_{\max}d}{\lambda_{\min}^3 f_{\min}^2 n}, \tag{14}$$

$$\mathbb{E}_{\theta_n}\left[\left\|\theta_n\left(\gamma\right) - \theta^*\left(\gamma\right)\right\|_2^2\right] \leq \frac{4\lambda_{\max}^2 f_{\max}d}{\lambda_{\min}^4 f_{\min}^2 n}. \tag{15}$$

To prove Theorem 3.1, we first show that $\ell_\gamma\left(\theta\right)$ in (4) is strongly convex and smooth with respect to $\theta^*(\gamma)$, as stated below in Proposition A.1. The proof of Proposition A.1 further relies on Lemma A.1 and Lemma A.2 for the gradient and the Hessian of $\ell_\gamma\left(\theta\right)$.

**Proposition A.1.** *Under Assumption 3.3, and if $\mathbb{E}\left[\|X\|^2\right] < \infty$, the objective $\ell_\gamma\left(\theta\right)$ in (4) satisfies*

$$\frac{f_{\min}}{2}\|\theta - \theta^*\left(\gamma\right)\|_\Sigma^2 \leq \ell_\gamma\left(\theta\right) - \ell_\gamma\left(\theta^*\left(\gamma\right)\right) \leq \frac{f_{\max}}{2}\|\theta - \theta^*\left(\gamma\right)\|_\Sigma^2. \tag{28}$$

*If Assumption 3.2 furthermore holds, then*

$$\frac{f_{\min}\lambda_{\min}}{2}\|\theta - \theta^*\left(\gamma\right)\|_2^2 \leq \ell_\gamma\left(\theta\right) - \ell_\gamma\left(\theta^*\left(\gamma\right)\right) \leq \frac{f_{\max}\lambda_{\max}}{2}\|\theta - \theta^*\left(\gamma\right)\|_2^2, \tag{29}$$

*where $\|\cdot\|_\Sigma$ denotes the $\Sigma$-induced norm, i.e., $\|\theta\|_\Sigma := \sqrt{\theta^\top \Sigma \theta}$.*

*Proof.* To prove this proposition, we first need Lemma A.1 and Lemma A.2 to calculate the gradient and the Hessian of $\ell_\gamma(\theta)$. By Lemma A.1,

$$\nabla\ell_\gamma(\theta^*(\gamma)) = \mathbb{E}_X\left[\left(F_{Y|X}\left((\theta^*(\gamma))^\top X \mid X\right) - \gamma\right) X\right]$$
$$= \mathbb{E}_X\left[(F_{Y|X}(q_\gamma(Y \mid X)) - \gamma) X\right]$$
$$= 0.$$

By Lemma A.2, $\nabla^2\ell_\gamma(\theta) = \mathbb{E}_X\left[f_{Y|X}(\theta^\top X \mid X) XX^\top\right]$. By Assumption 3.3, $\forall v \in \mathbb{R}^d$,

$$f_{\min}\|v\|_\Sigma^2 = f_{\min}\mathbb{E}_X\left[(X^\top v)^2\right] \le \mathbb{E}_X\left[f_{Y|X}(\theta^\top X \mid X)(X^\top v)^2\right]$$
$$\le f_{\max}\mathbb{E}_X\left[(X^\top v)^2\right] = f_{\max}\|v\|_\Sigma^2.$$

Hence, $f_{\min}\Sigma \preceq \nabla^2\ell_\gamma(\theta) \preceq f_{\max}\Sigma$ for any $\theta \in \Theta$. By Taylor's Formula,

$$\ell_\gamma(\theta) - \ell_\gamma(\theta^*(\gamma)) = \int_0^1 (1-u)(\theta - \theta^*(\gamma))^\top \nabla^2\ell_\gamma(\theta^* + u(\theta - \theta^*(\gamma)))(\theta - \theta^*(\gamma))\, du.$$

Since

$$f_{\min}\|\theta - \theta^*(\gamma)\|_\Sigma \le (\theta - \theta^*(\gamma))^\top \nabla^2\ell_\gamma(\theta^* + u(\theta - \theta^*(\gamma)))(\theta - \theta^*(\gamma))$$
$$\le f_{\max}\|\theta - \theta^*(\gamma)\|_\Sigma$$

and $\int_0^1 (1-u)\, du = 1/2$, we have

$$\frac{f_{\min}}{2}\|\theta - \theta^*(\gamma)\|_\Sigma^2 \le \ell_\gamma(\theta) - \ell_\gamma(\theta^*(\gamma)) \le \frac{f_{\max}}{2}\|\theta - \theta^*(\gamma)\|_\Sigma^2.$$

$$\square$$

**Lemma A.1.** *Suppose (11) in Assumption 3.3 is true, if $\mathbb{E}[\|X\|_2] < \infty$, then*

$$\nabla\ell_\gamma(\theta) = \mathbb{E}_{X,Y}\left[\left(\mathbb{1}\left\{Y < \theta^\top X\right\} - \gamma\right) X\right] = \mathbb{E}_X\left[\left(F_{Y|X}\left(\theta^\top X \mid X\right) - \gamma\right) X\right]. \quad (30)$$

*Proof.* The key idea is to show that the interchange of differentiation and expectation is valid according to the dominated convergence theorem. For $\theta \in \Theta$, it holds that

$$\mathbb{P}\left[Y = \theta^\top X\right] = \mathbb{E}_{(X,Y)}\left[\mathbb{1}\left\{Y = \theta^\top X\right\}\right]$$
$$= \mathbb{E}_X\left[\mathbb{E}_{Y|X}\left[\mathbb{1}\left\{Y = \theta^\top X\right\} \mid X\right]\right]$$
$$= \mathbb{E}_X\left[\mathbb{P}\left[Y = \theta^\top X \mid X\right]\right].$$

Since (11) in Assumption 3.3 is true, the p.d.f $f_{Y|X}(Y \mid X)$ exists for each $x \in \mathcal{X}$. Thus,

$$\mathbb{P}\left[Y = \theta^\top x \mid X = x\right] = \int_{\{\theta^\top x\}} f_{Y|X}(Y \mid X)\, dy = 0.$$

Thus, $\mathbb{P}[Y = t_\gamma(X;\theta)] = \mathbb{P}[Y = \theta^\top X] = \mathbb{E}[0] = 0$.

For $(x,y) \in \mathcal{X} \times \mathcal{Y}$, if $y \ne t_\gamma(x;\theta)$, the directional derivative of $L_\gamma(\theta^\top x, y)$ at $\theta$ along vector $v$ is

$$D_v L_\gamma(\theta^\top x, y) = \lim_{\rho\to 0}\frac{L_\gamma\left((\theta + \rho v)^\top x, y\right) - L_\gamma(\theta^\top x, y)}{\|v\|_2\rho}$$
$$= \frac{1}{\|v\|}\frac{d}{d\rho}L_\gamma\left((\theta + \rho v)^\top x, y\right)\Big|_{\rho=0}$$
$$= \left(\mathbb{1}\left\{y < \theta^\top x\right\} - \gamma\right) x^\top \frac{v}{\|v\|}.$$

Moreover, since $L_\gamma(t, y)$ is 1-Lipschitz with respect to $t$,

$$\left| \frac{L_\gamma\left((\theta + \rho v)^\top x, y\right) - L_\gamma\left(\theta^\top x, y\right)}{\|v\|_2 \rho} \right| = \frac{1}{\|v\|_2 \rho} \left| L_\gamma\left((\theta + \rho v)^\top x, y\right) - L_\gamma\left(\theta^\top x, y\right) \right|$$

$$\leq \frac{1}{\|v\|_2 \rho} \|(\theta + \rho v)^\top x - \theta^\top x\|_2$$

$$\leq \|x\|.$$

Since we assume $\mathbb{E}\left[\|X\|_2\right] < \infty$, by the dominated convergence theorem,

$$D_v \ell_\gamma(\theta) = D_v \mathbb{E}_{X,Y}\left[L_\gamma\left(\theta^\top X, Y\right)\right]$$

$$= \lim_{\rho \to 0} \frac{\mathbb{E}_{X,Y}\left[L_\gamma\left((\theta + \rho v)^\top X, Y\right)\right] - \mathbb{E}_{X,Y}\left[L_\gamma\left(\theta^\top X, Y\right)\right]}{\|v\|_2 \rho}$$

$$= \lim_{\rho \to 0} \mathbb{E}_{X,Y}\left[\frac{L_\gamma\left((\theta + \rho v)^\top X, Y\right) - L_\gamma\left(\theta^\top X, Y\right)}{\|v\|_2 \rho}\right]$$

$$= \mathbb{E}_{X,Y}\left[\lim_{\rho \to 0} \frac{L_\gamma\left((\theta + \rho v)^\top X, Y\right) - L_\gamma\left(\theta^\top X, Y\right)}{\|v\|_2 \rho}\right]$$

$$= \mathbb{E}_{X,Y}\left[D_v L_\gamma\left(\theta^\top X, Y\right)\right]$$

$$= \mathbb{E}_{X,Y}\left[\left(\mathbb{1}\left\{Y < \theta^\top X\right\} - \gamma\right) X\right]^\top \frac{v}{\|v\|}.$$

Hence,

$$\nabla \ell_\gamma(\theta) = \mathbb{E}_{X,Y}\left[\left(\mathbb{1}\left\{Y < \theta^\top X\right\} - \gamma\right) X\right]$$

$$= \mathbb{E}_X\left[\mathbb{E}_{Y|X}\left[\left(\mathbb{1}\left\{Y < \theta^\top X\right\} - \gamma\right) X \mid X\right]\right]$$

$$= \mathbb{E}_X\left[\mathbb{E}_{Y|X}\left[\left(\mathbb{1}\left\{Y < \theta^\top X\right\} - \gamma\right) \mid X\right] X\right]$$

$$= \mathbb{E}_X\left[\left(F_{Y|X}\left(\theta^\top X \mid X\right) - \gamma\right) X\right].$$

$\square$

**Lemma A.2.** *Under Assumption 3.3, if $\mathbb{E}\left[\|X\|^2\right] < \infty$, then*

$$\nabla^2 \ell_\gamma(\theta) = \mathbb{E}_X\left[f_{Y|X}\left(\theta^\top X \mid X\right) XX^\top\right]. \tag{31}$$

*Proof.* By Assumption, $\mathbb{E}\left[\|X\|_2\right] \leq \sqrt{\mathbb{E}\left[\|X\|^2\right]} < \infty$. Then, by Lemma A.1,

$$\nabla \ell_\gamma(\theta) = \mathbb{E}_{X,Y}\left[\left(\mathbb{1}\left\{Y < \theta^\top X\right\} - \gamma\right) X\right]$$

$$= \mathbb{E}_X\left[\mathbb{E}_{Y|X}\left[\left(\mathbb{1}\left\{Y < \theta^\top X\right\} - \gamma\right) X \mid X\right]\right]$$

$$= \mathbb{E}_X\left[\mathbb{E}_{Y|X}\left[\left(\mathbb{1}\left\{Y < \theta^\top X\right\} - \gamma\right) \mid X\right] X\right]$$

$$= \mathbb{E}_X\left[\left(F_{Y|X}\left(\theta^\top X \mid X\right) - \gamma\right) X\right].$$

To prove the lemma, the key point is to show that the interchange of differentiation and expectation is valid, as in the proof of Lemma A.1.

$$\lim_{\rho \to 0} \frac{\left(F_{Y|X}\left(\theta^\top x + \rho v^\top x \mid X\right) - \gamma\right) x - \left(F_{Y|X}\left(\theta^\top x \mid X\right) - \gamma\right) x}{\|v\|_2 \rho}$$

$$= \lim_{\rho \to 0} \frac{1}{\|v\|_2 \rho} \left(F_{Y|X}\left(\theta^\top x + \rho v^\top x \mid X\right) - F_{Y|X}\left(\theta^\top x \mid X\right)\right) x$$

$$= x \cdot \frac{v^\top x}{\|v\|} \lim_{\rho \to 0} \frac{1}{\rho v^\top x} \left(F_{Y|X}\left(\theta^\top x + \rho v^\top x \mid X\right) - F_{Y|X}\left(\theta^\top x \mid X\right)\right).$$

According to the mean value theorem, there exists $\xi(x)$ in $\left(\theta^\top x, \theta^\top x + \rho v^\top x\right)$ such that

$$\frac{1}{\rho v^\top x}\left(F_{Y|X}\left(\theta^\top x + \rho v^\top x \mid X\right) - F_{Y|X}\left(\theta^\top X \mid X\right)\right) = f_{Y|X}\left(\xi(x) \mid X\right).$$

Hence,

$$\lim_{\rho \to 0} \frac{1}{\rho v^\top x}\left(F_{Y|X}\left(\theta^\top x + \rho v^\top x \mid X\right) - F_{Y|X}\left(\theta^\top X \mid X\right)\right) = \lim_{\rho \to 0} f_{Y|X}\left(\xi(x) \mid X\right).$$

Since $f_{Y|X}(Y \mid X)$ is continuous for $\mathcal{P}_X$-almost every $x \in \mathcal{X}$, we have for $\mathcal{P}_X$-almost every $x \in \mathcal{X}$,

$$\lim_{\rho \to 0} f_{Y|X}\left(\xi(x) \mid X\right) = f_{Y|X}\left(\theta^\top X \mid X\right).$$

Hence, for $\mathcal{P}_X$-almost every $x \in \mathcal{X}$,

$$\lim_{\rho \to 0} \frac{\left(F_{Y|X}\left(\theta^\top x + \rho v^\top x \mid X\right) - \gamma\right) x - \left(F_{Y|X}\left(\theta^\top X \mid X\right) - \gamma\right) x}{\|v\|_2 \rho} = f_{Y|X}\left(\theta^\top X \mid X\right)\frac{xx^\top v}{\|v\|}.$$

Since (11) in Assumption 3.3 is true, for any $x \in \mathcal{X}$, $F_{Y|X}$ is $f_{\max}$-Lipschitz.

$$\left|\frac{\left(F_{Y|X}\left(\theta^\top x + \rho v^\top x \mid X\right) - \gamma\right) x - \left(F_{Y|X}\left(\theta^\top X \mid X\right) - \gamma\right) x}{\|v\|_2 \rho}\right|$$

$$= \frac{1}{\|v\|_2 \rho}\left|\left(F_{Y|X}\left(\theta^\top x + \rho v^\top x \mid X\right) - F_{Y|X}\left(\theta^\top X \mid X\right)\right)\right| \|x\|_2$$

$$\leq \frac{1}{\|v\|_2 \rho} f_{\max}\rho\|v\|_2\|x\|^2 = f_{\max}\|x\|^2.$$

Since $\mathbb{E}\left[\|X\|^2\right] < \infty$, it holds that $\mathbb{E}\left[f_{\max}\|X\|^2\right] < \infty$. Therefore, by the dominated convergence theorem, the directional derivative of $\nabla \ell_\gamma(\theta)$ at $\theta$ along vector $v$ is

$$D_v\left(\nabla \ell_\gamma(\theta)\right)$$
$$= D_v \mathbb{E}_X\left[\left(F_{Y|X}\left(\theta^\top X \mid X\right) - \gamma\right) X\right]$$
$$= \lim_{\rho \to 0} \frac{\mathbb{E}_X\left[\left(F_{Y|X}\left(\theta^\top X + \rho v^\top X \mid X\right) - \gamma\right) X\right] - \mathbb{E}_X\left[\left(F_{Y|X}\left(\theta^\top X \mid X\right) - \gamma\right) X\right]}{\|v\|_2 \rho}$$
$$= \lim_{\rho \to 0} \mathbb{E}_X\left[\frac{1}{\|v\|_2 \rho}\left(F_{Y|X}\left(\theta^\top X + \rho v^\top X \mid X\right) - F_{Y|X}\left(\theta^\top X \mid X\right)\right) X\right]$$
$$= \mathbb{E}_X\left[\lim_{\rho \to 0} \frac{1}{\|v\|_2 \rho}\left(F_{Y|X}\left(\theta^\top X + \rho v^\top X \mid X\right) - F_{Y|X}\left(\theta^\top X \mid X\right)\right) X\right]$$
$$= \mathbb{E}_X\left[f_{Y|X}\left(\theta^\top X \mid X\right) XX^\top\right]\frac{v}{\|v\|}.$$

Hence, $\nabla^2 \ell_\gamma(\theta) = \mathbb{E}_X\left[f_{Y|X}\left(\theta^\top X \mid X\right) XX^\top\right]$. $\qquad\square$

With Proposition A.1, we are ready to apply Theorem A.1 for SGD and get Corollary A.1.

**Theorem A.1** (Section 3 in Rakhlin et al. (2012)). *Suppose the loss function $\ell$ is $\lambda$-strongly convex and $\mu$-smooth with respect to a minimizer $\theta^*$ over $\Theta$, and $\mathbb{E}[\|g_t\|^2] \leq G^2$. Then taking $\eta_t = 1/\lambda t$, it holds for any $n$ that*

$$\mathbb{E}_{\theta_n}\left[f(\theta_n) - f(\theta^*)\right] \leq \frac{2\mu G^2}{\lambda^2 n}. \tag{32}$$

**Corollary A.1** (Upper Bound of Extra Loss). *Suppose Assumption 3.1, 3.2 and 3.3 hold. Let $\mathcal{D}_{train} := \{(X_i, Y_i)\}_{i=1}^n$ be the set of training samples and $\theta_n$ be the estimator by optimizing stochastic pinball loss (4) produced by SGD (7). Taking $\eta_k = 1/(\lambda_{\min} f_{\min} k)$, it holds that*

$$\mathbb{E}_{\theta_n}\left[\ell_\gamma(\theta_n(\gamma)) - \ell_\gamma(\theta^*(\gamma))\right] \leq \frac{2\lambda_{\max}^2 f_{\max} d}{\lambda_{\min}^2 f_{\min}^2 n}. \tag{33}$$

*Proof.* In this proof, we denote $\theta_n(\gamma)$ by $\theta_n$ for simplicity. By Lemma A.1, $\nabla \ell_\gamma(\theta) = \mathbb{E}_X \left[ \left( \mathbb{1}\left\{ Y < \theta^\top X \right\} \right) X \right]$. Then,

$$
\begin{aligned}
\mathbb{E}_{X,\theta_n} \left[ \left\| \nabla \ell_\gamma(\theta_n) \right\|^2 \right] &= \mathbb{E}_{\theta_n} \left[ \left\| \mathbb{E}_X \left[ \left( \mathbb{1}\left\{ Y < \theta_n^\top X \right\} \right) X \right] \right\|^2 \right] \\
&= \mathbb{E}_{\theta_n} \left[ \mathbb{E}_X \left[ \left\| \left( \mathbb{1}\left\{ Y < \theta_n^\top X \right\} \right) X \right\| \right]^2 \right] \\
&\leq \mathbb{E}_X \left[ \|X\| \right]^2 \leq \lambda_{\max} d.
\end{aligned}
$$

where the last inequality is by Assumption 3.2,

$$
\mathbb{E} \left[ \|X\| \right]^2 \leq \mathbb{E} \left[ \|X\|^2 \right] = \mathbb{E} \left[ \text{trace} \left( XX^\top \right) \right] = \text{trace} \left( \mathbb{E} \left[ XX^\top \right] \right) \leq \text{trace} \left( \lambda_{\max} I \right) = d\lambda_{\max}.
$$

The corollary then follows from Proposition A.1 and Theorem A.1. $\qquad \square$

Now we are ready to prove Theorem 3.1. In this proof, we denote $\theta_n(\gamma), \theta^*(\gamma)$ by $\theta_n, \theta^*$, respectively, for simplicity. By Proposition A.1,

$$
\begin{aligned}
\|\theta_n - \theta^*\|_\Sigma^2 &\leq \frac{2}{f_{\min}} \left( \ell(\theta_n) - \ell(\theta^*) \right); \\
\|\theta_n - \theta^*\|_2^2 &\leq \frac{2}{f_{\min} \lambda_{\min}} \left( \ell(\theta_n) - \ell(\theta^*) \right).
\end{aligned}
$$

Since the test sample $(X, Y)$ is sampled independently of the set of the training samples $\{(X_i, Y_i)\}_{i=1}^n$, and $\theta_n$ is a function of $\{(X_i, Y_i)\}_{i=1}^n$, $\theta_n$ is independent of $X$.

$$
\begin{aligned}
\mathbb{E}_{\theta_n, X} \left[ \left( t(X; \theta_n) - t(X; \theta^*) \right)^2 \right] &= \mathbb{E}_{\theta_n, X} \left[ \left( \left( \theta_n - \theta^* \right)^\top X \right)^2 \right] \\
&= \mathbb{E}_{\theta_n} \left[ \mathbb{E}_X \left[ \left( \theta_n - \theta^* \right)^\top XX^\top \left( \theta_n - \theta^* \right) | \theta_n \right] \right] \\
&= \mathbb{E}_{\theta_n} \left[ \left( \theta_n - \theta^* \right)^\top \mathbb{E}_X \left[ XX^\top \right] \left( \theta_n - \theta^* \right) \right] \\
&= \mathbb{E}_{\theta_n} \left[ \|\theta_n - \theta^*\|_\Sigma^2 \right].
\end{aligned}
$$

Hence, by Corollary A.1, $\mathbb{E}_{\theta_n} [\|\theta_n - \theta^*\|_\Sigma^2] \leq \frac{2}{f_{\min}} \mathbb{E}_{\theta_n} [(\ell(\theta_n) - \ell(\theta^*))] \leq \frac{4\lambda_{\max}^2 f_{\max} d}{\lambda_{\min}^3 f_{\min}^2 n}$.

This completes the proof of Theorem 3.1.

## A.2   PROOF OF PROPOSITION A.2

**Proposition A.2** (Population quantile of the score). *In CQR, if $F_{Y|X}(Y \mid X = x)$ is continuous for all $x \in \mathcal{X}$, then*

$$
|q_{1-\alpha}(S \mid \vartheta_n)| \leq B \max \left\{ \|\underline{\theta}_n - \underline{\theta}^*\|_2, \|\bar{\theta}_n - \bar{\theta}^*\|_2 \right\}. \tag{34}
$$

*Suppose Assumptions 3.1–3.3 hold,*

$$
\mathbb{E}_{\vartheta_n} \left[ |q_{1-\alpha}(S \mid \vartheta_n)| \right] \leq \frac{2B \, \lambda_{\max} \sqrt{2f_{\max} d}}{\lambda_{\min}^2 f_{\min}} \sqrt{\frac{1}{n}}. \tag{35}
$$

The proof of Proposition A.2 relies on the following lemma.

**Lemma A.3.** *Suppose $F_{Y|X}(Y \mid X)$ is continuous for each $x \in \mathcal{X}$. Then,*

$$
|q_{1-\alpha}(S \mid X, \vartheta_n)| \leq \Delta(X, \vartheta_n), \tag{36}
$$

*where $q_{1-\alpha}(S \mid X, \vartheta_n)$ denotes the $(1-\alpha)$-quantile of $S$ given $X, \vartheta_n$.*

*Proof.* By the definitions (24, 25, 26),

$$
\begin{aligned}
S(X, Y; \vartheta_n) &:= \max \left\{ t_{\alpha/2}(X; \underline{\theta}_n) - Y, \ Y - t_{1-\alpha/2}(X; \bar{\theta}_n) \right\} \\
&\leq \max \left\{ \mathcal{E}_{\alpha/2}(X, \underline{\theta}_n) + q_{\alpha/2}(Y \mid X) - Y, \ \mathcal{E}_{1-\alpha/2}(X, \bar{\theta}_n) + Y - q_{1-\alpha/2}(Y \mid X) \right\} \\
&\leq \Delta(X, \vartheta_n) + S^*(X, Y), \tag{37}
\end{aligned}
$$

where the last inequality is because $\max\{u_1 + v_1, u_2 + v_2\} \leq \max\{u_1, u_2\} + \max\{v_1, v_2\}$.

Similarly,

$$
\begin{aligned}
S\left(X, Y; \vartheta_n\right) &:= \max\left\{t_{\alpha/2}\left(X; \underline{\theta}_n\right) - Y, \; Y - t_{1-\alpha/2}\left(X; \bar{\theta}_n\right)\right\} \\
&\geq \max\left\{q_{\alpha/2}\left(Y \mid X\right) - Y - \mathcal{E}_{\alpha/2}\left(X, \underline{\theta}_n\right), \; Y - q_{1-\alpha/2}\left(Y \mid X\right) - \mathcal{E}_{1-\alpha/2}\left(X, \bar{\theta}_n\right)\right\} \\
&= S^*\left(X, Y\right) - \Delta\left(X, \vartheta_n\right),
\end{aligned}
\tag{38}
$$

where the last inequality is because $\max\{u_1 - v_1, u_2 - v_2\} \geq \max\{u_1, u_2\} - \max\{v_1, v_2\}$.

Note that $S^*\left(X, Y\right) \leq 0$ is equivalent to $q_{\alpha/2}\left(Y \mid X\right) \leq Y \leq q_{1-\alpha/2}\left(Y \mid X\right)$. Since $F_{Y|X}$ is continuous,
$$
\mathbb{P}\left[q_{\alpha/2}\left(Y \mid X\right) \leq Y \leq q_{1-\alpha/2}\left(Y \mid X\right) \mid X\right] = 1 - \alpha.
$$
Hence, $\mathbb{P}[S^*\left(X, Y\right) \leq 0 | X] = 1 - \alpha$. Let $q_{1-\alpha}\left(S^* \mid X\right)$ be the $(1 - \alpha)$-quantile of $S^*$ given $X$. Since $X$ is given, and $F_{Y|X}$ is continuous, $F_{S^*|X}$ is continuous. Then, $q_{1-\alpha}\left(S^* \mid X\right) = 0$. Conditional on $X, \vartheta_n, \Delta\left(X, \vartheta_n\right)$ is deterministic. By (37), we have

$$
\begin{aligned}
&\mathbb{P}\left[S\left(X, Y; \vartheta_n\right) \leq u \mid X, \vartheta_n\right] \geq \mathbb{P}\left[\Delta\left(X, \vartheta_n\right) + S^*\left(X, Y\right) \leq u \mid X, \vartheta_n\right] \\
&\implies \mathbb{P}\left[S\left(X, Y; \vartheta_n\right) \leq \Delta\left(X, \vartheta_n\right) \mid X, \vartheta_n\right] \geq \mathbb{P}\left[S^*\left(X, Y\right) \leq 0 \mid X\right] = 1 - \alpha.
\end{aligned}
$$

Then, $q_{1-\alpha}\left(S \mid X, \vartheta_n\right) \leq \Delta\left(X, \vartheta_n\right)$. By (38), we have

$$
\begin{aligned}
&\mathbb{P}\left[S\left(X, Y; \vartheta_n\right) \leq u \mid X, \vartheta_n\right] \leq \mathbb{P}\left[S^*\left(X, Y\right) - \Delta\left(X, \vartheta_n\right) \leq u \mid X, \vartheta_n\right] \\
&\implies \mathbb{P}\left[S\left(X, Y; \vartheta_n\right) \leq -\Delta\left(X, \vartheta_n\right) \mid X, \vartheta_n\right] \leq \mathbb{P}\left[S^*\left(X, Y\right) \leq 0 \mid X\right] = 1 - \alpha.
\end{aligned}
$$

Then, $q_{1-\alpha}\left(S \mid X, \vartheta_n\right) \geq -\Delta\left(X, \vartheta_n\right)$. □

For $\gamma \in \left\{\frac{\alpha}{2}, 1 - \frac{\alpha}{2}\right\}$,

$$
\mathcal{E}_\gamma\left(X, \theta_n\left(\gamma\right)\right) = \left|\left(\theta_n\left(\gamma\right) - \theta^*\left(\gamma\right)\right)^\top X\right| \leq \left\|\left(\theta_n\left(\gamma\right) - \theta^*\left(\gamma\right)\right)\right\|_2 \left\|X\right\|_2 \leq B \left\|\left(\theta_n\left(\gamma\right) - \theta^*\left(\gamma\right)\right)\right\|_2,
$$

where the last inequality is from the fact that the norm of $x \in \mathcal{X}$ is bounded by $B$. Then,

$$
\Delta\left(X, \vartheta_n\right) \leq B \max\left\{\left\|\left(\underline{\theta}_n - \underline{\theta}^*\right)\right\|_2, \; \left\|\left(\bar{\theta}_n - \bar{\theta}^*\right)\right\|_2\right\} = B \cdot M\left(\vartheta_n\right).
$$

By Lemma A.3, $\left|q_{1-\alpha}\left(S \mid X, \vartheta_n\right)\right| \leq \Delta\left(X, \vartheta_n\right) \leq B \cdot M\left(\vartheta_n\right)$. Then,

$$
\begin{aligned}
\mathbb{P}\left[S\left(X, Y; \vartheta_n\right) \leq B \cdot M\left(\vartheta_n\right) \mid X, \vartheta_n\right] &\geq 1 - \alpha; \\
\mathbb{P}\left[S\left(X, Y; \vartheta_n\right) \geq -B \cdot M\left(\vartheta_n\right) \mid X, \vartheta_n\right] &\leq 1 - \alpha.
\end{aligned}
$$

Then, removing the conditioning on $X$,

$$
\begin{aligned}
&\mathbb{P}\left[S\left(X, Y; \vartheta_n\right) \leq B \cdot M\left(\vartheta_n\right) \mid \vartheta_n\right] \\
&= \mathbb{E}_{X,Y|\vartheta_n}\left[\mathbb{1}\left\{S\left(X, Y; \vartheta_n\right) \leq B \cdot M\left(\vartheta_n\right)\right\} \mid \vartheta_n\right] \\
&= \mathbb{E}_{X|\vartheta_n}\left[\mathbb{E}_{Y|X,\vartheta_n}\left[\mathbb{1}\left\{S\left(X, Y; \vartheta_n\right) \leq B \cdot M\left(\vartheta_n\right)\right\} \mid X, \vartheta_n\right] \mid \vartheta_n\right] \\
&= \mathbb{E}_{X|\vartheta_n}\left[\mathbb{P}\left[S\left(X, Y; \vartheta_n\right) \leq B \cdot M\left(\vartheta_n\right) \mid X, \vartheta_n\right] \mid \vartheta_n\right] \\
&\geq \mathbb{E}_{X|\vartheta_n}\left[1 - \alpha \mid \vartheta_n\right] = 1 - \alpha.
\end{aligned}
$$

Hence, $q_{1-\alpha}\left(S \mid \vartheta_n\right) \leq B \cdot M\left(\vartheta_n\right)$. And by similar arguments as below, $q_{1-\alpha}\left(S \mid \vartheta_n\right) \geq -B \cdot M\left(\vartheta_n\right)$.

$$
\begin{aligned}
&\mathbb{P}\left[S\left(X, Y; \vartheta_n\right) \geq -B \cdot M\left(\vartheta_n\right) \mid \vartheta_n\right] \\
&= \mathbb{E}_{X,Y|\vartheta_n}\left[\mathbb{1}\left\{S\left(X, Y; \vartheta_n\right) \geq -B \cdot M\left(\vartheta_n\right)\right\} \mid \vartheta_n\right] \\
&= \mathbb{E}_{X|\vartheta_n}\left[\mathbb{E}_{Y|X,\vartheta_n}\left[\mathbb{1}\left\{S\left(X, Y; \vartheta_n\right) \geq -B \cdot M\left(\vartheta_n\right)\right\} \mid X, \vartheta_n\right] \mid \vartheta_n\right] \\
&= \mathbb{E}_{X|\vartheta_n}\left[\mathbb{P}\left[S\left(X, Y; \vartheta_n\right) \geq -B \cdot M\left(\vartheta_n\right) \mid X, \vartheta_n\right] \mid \vartheta_n\right] \\
&\leq \mathbb{E}_{X|\vartheta_n}\left[1 - \alpha \mid \vartheta_n\right] = 1 - \alpha.
\end{aligned}
$$

Therefore, $|q_{1-\alpha}(S \mid \vartheta_n)| \leq B \cdot M(\vartheta_n)$. Then,

$$\mathbb{E}_{\vartheta_n}[|q_{1-\alpha}(S \mid \vartheta_n)|] \leq B\, \mathbb{E}_{\vartheta_n}[M(\vartheta_n)]$$

$$\leq B\, \mathbb{E}_{\vartheta_n}\left[\sqrt{\|(\underline{\theta}_n - \underline{\theta}^*)\|_2^2 + \|(\bar{\theta}_n - \bar{\theta}^*)\|_2^2}\right]$$

$$\leq B\sqrt{\mathbb{E}_{\vartheta_n}\left[\|(\underline{\theta}_n - \underline{\theta}^*)\|_2^2 + \|(\bar{\theta}_n - \bar{\theta}^*)\|_2^2\right]}$$

$$\leq B\sqrt{\mathbb{E}_{\vartheta_n}\left[\|(\underline{\theta}_n - \underline{\theta}^*)\|_2^2\right] + \mathbb{E}_{\vartheta_n}\left[\|(\bar{\theta}_n - \bar{\theta}^*)\|_2^2\right]}$$

$$\leq B\sqrt{\frac{8\lambda_{\max}^2 f_{\max}d}{\lambda_{\min}^4 f_{\min}^2 n}} = \frac{2B\,\lambda_{\max}\sqrt{2f_{\max}d}}{\lambda_{\min}^2 f_{\min}}\sqrt{\frac{1}{n}},$$

where the second inequality is from $\max\{a, b\} \leq \sqrt{a^2 + b^2}$, the third inequality is by Jensen's inequality, and the last inequality is from Theorem 3.1.

This completes the proof of Proposition A.2.

## A.3 PROOF OF PROPOSITION A.3

**Proposition A.3** (Population finite-sample score-quantile gap)**.** *In CQR, Suppose Assumptions 3.1–3.3 hold, if $m > 8H/\min\{\alpha, 1 - \alpha\}$ for $H$ in (12), then*

$$\mathbb{E}_{\vartheta_n}\left[|q_{(1-\alpha)_m}(S \mid \vartheta_n) - q_{1-\alpha}(S \mid \vartheta_n)|\right] \leq \frac{1}{f_{\min}m} + \frac{1056R f_{\max}^3 \lambda_{\max}^2 B^2 d}{\min\{\alpha^2, (1-\alpha)^2\}\lambda_{\min}^4 f_{\min}^2 n}.$$

To prove Proposition A.3, we first need the following critical proposition:

**Proposition A.4.** *Suppose $\alpha \in (0, 1)$ is a constant. Define*

$$\beta := \min\left\{\frac{\alpha}{2f_{\max}}, \frac{1-\alpha}{2f_{\max}}\right\}, \qquad A := \frac{4\lambda_{\max}^2 f_{\max}d}{\lambda_{\min}^4 f_{\min}^2}, \qquad \epsilon_n := B\sqrt{\frac{2A}{n\delta}}.$$

*Under the same setting of Theorem 3.1, if $\epsilon_n < \beta/4$ (equivalently $n > \frac{32AB^2}{\beta^2\delta}$), then for $\delta \in (0, 1)$, with probability at least $1 - \delta$ over $\vartheta_n$, the following (denoted by event $V$) hold simultaneously:*

*1. For $s$ with $|s| < \beta - \epsilon_n$, $f_{S|\vartheta_n}(s \mid \vartheta_n) \geq 2f_{\min}$.*

*2. $|q_{1-\alpha}(S \mid \vartheta_n)| \leq \epsilon_n < \beta/4$.*

*Proof.* By the definition of $S$ in (8),

$$\mathbb{P}[S \leq s | X, \vartheta_n] = \mathbb{P}\left[\begin{array}{c} t_{\alpha/2}(x; \underline{\theta}_n) - s \leq Y \leq t_{1-\alpha/2}(x; \bar{\theta}_n) + s] \\ \text{and } s \geq \frac{t_{\alpha/2}(x;\underline{\theta}_n) - t_{1-\alpha/2}(x;\bar{\theta}_n)}{2} \end{array} \Bigg| X, \vartheta_n\right].$$

Hence,

$$F_{S|X,\vartheta_n}(s) = \begin{cases} 0, & \text{if } s < \frac{t_{\alpha/2}(x;\underline{\theta}_n) - t_{1-\alpha/2}(x;\bar{\theta}_n)}{2}, \\ F_{Y|X,\vartheta_n}(t_{1-\alpha/2}(x; \bar{\theta}_n) + s) \\ \quad - F_{Y|X,\vartheta_n}(t_{\alpha/2}(x; \underline{\theta}_n) - s), & \text{otherwise.} \end{cases} \tag{39}$$

We now show that with high probability, it holds for $s$ in the neighbourhood of $0$ that

$$s \geq \frac{t_{\alpha/2}(x; \underline{\theta}_n) - t_{1-\alpha/2}(x; \bar{\theta}_n)}{2}, \quad t_{1-\alpha/2}(x; \bar{\theta}_n) + s \in \mathcal{Y}, \quad t_{\alpha/2}(x; \underline{\theta}_n) - s \in \mathcal{Y}.$$

Let $y_{\max} := \sup\{y \in \mathcal{Y}\}$ and $y_{\min} := \inf\{y \in \mathcal{Y}\}$. Then, under Assumption 3.3, $y_{\max} > y_{\min}$.

$$q_{\alpha/2}(Y \mid X = x), q_{1-\alpha/2}(Y \mid X = x) \in [y_{\min}, y_{\max}],$$

$$q_{\alpha/2}(Y \mid X = x) - y_{\min} \geq \frac{\alpha}{2f_{\max}} \geq \beta, \qquad y_{\max} - q_{1-\alpha/2}(Y \mid X = x) \geq \frac{\alpha}{2f_{\max}} \geq \beta,$$

$$\frac{q_{1-\alpha/2}(Y \mid X = x) - q_{\alpha/2}(Y \mid X = x)}{2} \geq \frac{1-\alpha}{2f_{\max}} \geq \beta.$$

By Theorem 3.1, $\mathbb{E}_{\theta_n}\left[\|\theta_n(\gamma) - \theta^*(\gamma)\|_2^2\right] \le \frac{A}{n}$ for $\gamma \in \{\frac{\alpha}{2}, 1-\frac{\alpha}{2}\}$. By Markov's inequality,

$$\mathbb{P}\left[\|\theta_n(\gamma) - \theta^*(\gamma)\|_2 \le \sqrt{\frac{2A}{n\delta}}\right] \ge 1 - \frac{\delta}{2}.$$

Applying the union bound, we have

$$\mathbb{P}\left[\max_{\gamma \in \{\frac{\alpha}{2}, 1-\frac{\alpha}{2}\}} \|\theta_n(\gamma) - \theta^*(\gamma)\|_2 \le \sqrt{\frac{2A}{n\delta}}\right] \ge 1 - \delta.$$

Since for each $x \in \mathcal{X}$,

$$\mathcal{E}_\gamma(x, \theta_n(\gamma)) = |t_\gamma(x; \theta_n(\gamma)) - t_\gamma(x; \theta^*(\gamma))| = \left|(\theta_n(\gamma) - \theta^*(\gamma))^\top x\right|$$
$$\le \|(\theta_n(\gamma) - \theta^*(\gamma))\|_2 \|x\|_2 \le B \|(\theta_n(\gamma) - \theta^*(\gamma))\|_2.$$

we have that with probability at least $1 - \delta$,

$$\sup_x \Delta(x, \vartheta_n) \le B \max_{\gamma \in \{\frac{\alpha}{2}, 1-\frac{\alpha}{2}\}} \|\theta_n(\gamma) - \theta^*(\gamma)\|_2 \le B\sqrt{\frac{2A}{n\delta}} =: \epsilon_n.$$

and by Proposition A.2, it also holds that

$$|q_{1-\alpha}(S \mid \vartheta_n)| \le \epsilon_n. \tag{40}$$

Then, w.p. $\ge 1 - \delta$, for any $x \in \mathcal{X}$,

$$t_{\alpha/2}(x; \underline{\theta}_n) \ge q_{\alpha/2}(Y \mid X = x) - \Delta(x, \vartheta_n) \ge y_{\min} + \beta - \epsilon_n;$$
$$t_{1-\alpha/2}(x; \bar{\theta}_n) \le q_{1-\alpha/2}(Y \mid X = x) + \Delta(x, \vartheta_n) \le y_{\max} - \beta + \epsilon_n;$$
$$\frac{t_{1-\alpha/2}(x; \bar{\theta}_n) - t_{\alpha/2}(x; \underline{\theta}_n)}{2} \ge \frac{q_{1-\alpha/2}(Y \mid X = x) - q_{\alpha/2}(Y \mid X = x)}{2} - \Delta(x, \vartheta_n) \ge \beta - \epsilon_n.$$

The last inequality above shows that with high probability, quantile crossing will not occur given $n$ is large enough.

In this case, for any $s$ with $|s| < r_n := \beta - \epsilon_n$, we have $\forall x \in \mathcal{X}$,

$$t_{\alpha/2}(x; \underline{\theta}_n) - s > y_{\min} + \beta - \epsilon_n - r_n \ge y_{\min};$$
$$t_{\alpha/2}(x; \underline{\theta}_n) - s < q_{\alpha/2}(Y \mid X = x) + \epsilon_n + r_n \le q_{1-\alpha/2}(Y \mid X = x) + \beta \le y_{\max};$$
$$t_{1-\alpha/2}(x; \bar{\theta}_n) + s < y_{\max} - \beta + \epsilon_n + r_n \le y_{\max};$$
$$t_{1-\alpha/2}(x; \bar{\theta}_n) + s > q_{1-\alpha/2}(Y \mid X = x) - \epsilon_n - r_n \ge q_{\alpha/2}(Y \mid X = x) - \beta \ge y_{\min};$$
$$s \ge -|s| \ge -r_n = \epsilon_n - \beta \ge \frac{t_{\alpha/2}(x; \underline{\theta}_n) - t_{1-\alpha/2}(x; \bar{\theta}_n)}{2}.$$

Since $\mathcal{Y}$ is an interval,

$$t_{\alpha/2}(x; \underline{\theta}_n) - s \in \mathcal{Y}, \qquad t_{1-\alpha/2}(x; \bar{\theta}_n) + s \in \mathcal{Y}.$$

Therefore, by (39), conditioning on $\vartheta_n$, for $s$ with $|s| < r_n = \beta - \epsilon_n$,

$$f_{S|\vartheta_n}(s \mid \vartheta_n) = \mathbb{E}_{X|\vartheta_n}[\; f_{Y|X,\vartheta_n}(t_{\alpha/2}(x; \underline{\theta}_n) - s \mid X, \vartheta_n)$$
$$+ f_{Y|X,\vartheta_n}(t_{1-\alpha/2}(x; \bar{\theta}_n) + s \mid X, \vartheta_n)]$$
$$\ge 2f_{\min}.$$

Suppose $n > \frac{32AB^2}{\beta^2\delta}$, which is equivalent to $\epsilon_n < \beta/4$. Then, $r_n = \beta - \epsilon_n \ge 3\beta/4 \ge \epsilon_n$. By (40), $|q_{1-\alpha}(S \mid \vartheta_n)| \le \beta - \epsilon_n$. $\qquad\square$

The proof of Proposition A.3 also relies on the following useful lemma, which is a classical result (Bobkov & Ledoux, 2019). We include the proof here for completeness.

**Lemma A.4.** *Let $F$ be a c.d.f. with p.d.f. $f$. Suppose there exists an interval $\mathcal{I} \in \mathbb{R}$ and a constant $c_0 > 0$ such that $f(s) \geq c_0$ for all $s \in \mathcal{I}$. For $p \in (0,1)$, $q_p := \inf\{u : F(u) \geq p\} \in \mathcal{I}$, define $r_0 := \min\{q_p - \inf\mathcal{I}, \sup\mathcal{I} - q_p\} \geq 0$. Then, for any $p'$ such that $|p' - p| < c_0 r_0$, it holds that $q_{p'} \in \mathcal{I}$, and $|q_{p'} - q_p| \leq \frac{|p'-p|}{c_0}$.*

*Proof.* By assumption,

$$F(q_p - r_0) \leq F(q_p) - c_0 r_0 = p - c_0 r_0;$$
$$F(q_p + r_0) \geq F(q_p) + c_0 r_0 = p + c_0 r_0.$$

Since $|p' - p| < c_0 r_0$, either $p \leq p' < p + c_0 r_0$ or $p' \leq p < p' + c_0 r_0$. If $p \leq p' < p + c_0 r_0$, then $p \leq p' < F(q_p + r_0)$. Since $F$ is non-decreasing, $q_p \leq q_{p'} < q_p + r_0$. Similarly, if $p - c_0 r_0 < p' \leq p$, then $F(q_p - r_0) < p' \leq p$, and $q_p - r_0 < q_{p'} \leq q_p$. Hence, $q_{p'} \in \mathcal{I}$, and $|q_{p'} - q_p| \leq \frac{|p'-p|}{c_0}$. $\square$

With Proposition A.4, we apply Lemma A.4 and get Lemma A.5.

**Lemma A.5.** *Under the same setting of Proposition A.4, if the event in Proposition A.4 occurs, and if $m > \frac{4}{f_{\min}\beta}$, then it holds that $|q_{(1-\alpha)_m}(S \mid \vartheta_n)| \leq \beta/2$, $f_{S|\vartheta_n}(q_{(1-\alpha)_m}(S \mid \vartheta_n)) \geq 2f_{\min}$, and $|q_{(1-\alpha)_m}(S \mid \vartheta_n) - q_{1-\alpha}(S \mid \vartheta_n)| \leq \frac{1}{f_{\min}m}$.*

*Proof.* For simplicity, in the proof we denote $q_p(S \mid \vartheta_n)$ by $q_p$.

$$(1-\alpha)(m+1) \leq \lceil(1-\alpha)(m+1)\rceil < (1-\alpha)(m+1) + 1$$
$$\Rightarrow (1-\alpha)(m+1) - (1-\alpha)m \leq \lceil(1-\alpha)(m+1)\rceil - (1-\alpha)m < (1-\alpha)(m+1) + 1 - (1-\alpha)m$$
$$\Rightarrow 0 < \frac{1-\alpha}{m} \leq |(1-\alpha)_m - (1-\alpha)| < \frac{2-\alpha}{m} < \frac{2}{m}.$$

Since $\epsilon_n < \beta/4$, from Proposition A.4, with probability at least $1 - \delta$, for $s$ with $|s| < 3\beta/4$, $f_{S|\vartheta_n}(s \mid \vartheta_n) \geq 2f_{\min}$, and $|q_{1-\alpha}| < \beta/4$. In this case, $r_0 := \min\{q_{1-\alpha} + 3\beta/4, 3\beta/4 - q_{1-\alpha}\} > \beta/2$. If $m > \frac{4}{f_{\min}\beta}$, then $|(1-\alpha)_m - (1-\alpha)| < \frac{2}{m} < 2f_{\min}\frac{\beta}{4} < 2f_{\min}\frac{\beta}{2} < 2f_{\min} \cdot r_0$. Then by Lemma A.4, $|q_{(1-\alpha)_m}| \leq 3\beta/4$, $f_{S|\vartheta_n}(q_{(1-\alpha)_m}(S \mid \vartheta_n)) \geq 2f_{\min}$, and $|q_{(1-\alpha)_m} - q_{1-\alpha}| < \frac{|(1-\alpha)_m - (1-\alpha)|}{2f_{\min}} < \frac{1}{f_{\min}m} \leq \beta/4$. Hence, $|q_{(1-\alpha)_m}| \leq |q_{1-\alpha}| + |q_{(1-\alpha)_m} - q_{1-\alpha}| < \beta/4 + \beta/4 = \beta/2$. $\square$

Notice that $|q_{(1-\alpha)_m}(S \mid \vartheta_n) - q_{1-\alpha}(S \mid \vartheta_n)|$ is bounded by $2R$. Let $V$ denote the event in Proposition A.4, and $V^c$ its complement. Then, by Lemma A.5,

$$\mathbb{E}_{\vartheta_n}\left[|q_{(1-\alpha)_m}(S \mid \vartheta_n) - q_{1-\alpha}(S \mid \vartheta_n)|\right]$$
$$= \mathbb{P}[V] \cdot \mathbb{E}_{\vartheta_n}\left[|q_{(1-\alpha)_m}(S \mid \vartheta_n) - q_{1-\alpha}(S \mid \vartheta_n)| \mid V\right]$$
$$+ \mathbb{P}[V^c] \cdot \mathbb{E}_{\vartheta_n}\left[|q_{(1-\alpha)_m}(S \mid \vartheta_n) - q_{1-\alpha}(S \mid \vartheta_n)| \mid V^c\right]$$
$$\leq \frac{1}{f_{\min}m} + 2R\delta.$$

Picking $\delta = \frac{33AB^2}{\beta^2 n}$ completes the proof of Proposition A.3.

## A.4 PROOF OF PROPOSITION A.5

**Proposition A.5** (Empirical score-quantile concentration). *In CQR, Suppose Assumptions 3.1–3.3 hold, if $m > 8H/\min\{\alpha, 1-\alpha\}$ for $H$ in (12), then*

$$\mathbb{E}_{\vartheta_n, \mathcal{D}_{cal}}\left[|\hat{q}_{(1-\alpha)_m}(S_m \mid \vartheta_n) - q_{(1-\alpha)_m}(S \mid \vartheta_n)|\right]$$
$$\leq \frac{\sqrt{\pi}}{2f_{\min}\sqrt{2m}} + 4R\exp\left(-\frac{\min\{\alpha^2, (1-\alpha)^2\}f_{\min}^2}{8f_{\max}^2}m\right) + \frac{1056Rf_{\max}^3\lambda_{\max}^2 B^2 d}{\min\{\alpha^2, (1-\alpha)^2\}\lambda_{\min}^4 f_{\min}^2 n}.$$

To prove Proposition A.5, we first prove the following lemma:

**Lemma A.6.** *Under the same setting of Lemma A.5, if the high probability event $V$ in Proposition A.4 occurs, for any $u \in [0, \beta/4]$, if*

$$\sup_s \left| F_{S|\vartheta_n}(s) - \hat{F}_{S|\vartheta_n}^{(m)}(s) \right| \leq 2f_{\min}u,$$

*then $|\hat{q}_{(1-\alpha)_m}(S_m \mid \vartheta_n) - q_{(1-\alpha)_m}(S \mid \vartheta_n)| \leq u$.*

*Proof.* For simplicity, in the proof we denote $q_p(S \mid \vartheta_n)$ by $q_p$. By Lemma A.5, for $u \in [0, \beta/4]$, $|q_{(1-\alpha)_m} - u| \leq 3\beta/4$ and $|q_{(1-\alpha)_m} + u| \leq 3\beta/4$. Hence, in this case,

$$F_{S|\vartheta_n}\left(q_{(1-\alpha)_m} - u\right) \leq F_{S|\vartheta_n}\left(q_{(1-\alpha)_m}\right) - 2f_{\min}u = (1-\alpha)_m - 2f_{\min}u;$$
$$F_{S|\vartheta_n}\left(q_{(1-\alpha)_m} + u\right) \geq F_{S|\vartheta_n}\left(q_{(1-\alpha)_m}\right) + 2f_{\min}u = (1-\alpha)_m + 2f_{\min}u.$$

By assumption,

$$\left| F_{S|\vartheta_n}\left(q_{(1-\alpha)_m} - u\right) - \hat{F}_{S|\vartheta_n}^{(m)}\left(q_{(1-\alpha)_m} - u\right) \right| \leq 2f_{\min}u;$$
$$\left| F_{S|\vartheta_n}\left(q_{(1-\alpha)_m} + u\right) - \hat{F}_{S|\vartheta_n}^{(m)}\left(q_{(1-\alpha)_m} + u\right) \right| \leq 2f_{\min}u.$$

Then

$$\hat{F}_{S|\vartheta_n}^{(m)}\left(q_{(1-\alpha)_m} - u\right) \leq (1-\alpha)_m, \qquad \hat{F}_{S|\vartheta_n}^{(m)}\left(q_{(1-\alpha)_m} + u\right) \geq (1-\alpha)_m.$$

Since $\hat{F}_{S|\vartheta_n}^{(m)}$ is non-decreasing, we have

$$\hat{q}_{(1-\alpha)_m}(S_m \mid \vartheta_n) := \inf\{u' \in \mathcal{S}_m : \hat{F}_{S|\vartheta_n}^{(m)}(u') \geq (1-\alpha)_m\} \in \left[q_{(1-\alpha)_m} - u, q_{(1-\alpha)_m} + u\right],$$

where $\mathcal{S}_m$ is the set of scores of the calibration data.

Then, $|\hat{q}_{(1-\alpha)_m}(S_m \mid \vartheta_n) - q_{(1-\alpha)_m}(S \mid \vartheta_n)| \leq u$. $\qquad\square$

**Lemma A.7** (Dvoretzky–Kiefer–Wolfowitz Inequality (Dvoretzky et al., 1956; Massart, 1990)). *Given a natural number $m$, let $X_1, \ldots, X_m$ be real-valued i.i.d. random variables with c.d.f. $F(\cdot)$. Let $F^{(m)}$ denote the associated empirical distribution function defined by*

$$F^{(m)}(x) = \frac{1}{m} \sum_{j=1}^{m} \mathbb{1}\{X_j \leq x\}, \qquad x \in \mathbb{R}.$$

*Then,*

$$\mathbb{P}\left[\sup_{x \in \mathbb{R}} \left| F^{(m)}(x) - F(x) \right| > \varepsilon \right] \leq 2e^{-2m\varepsilon^2}, \qquad \forall \varepsilon \geq 0.$$

By the Dvoretzky–Kiefer–Wolfowitz Inequality (Lemma A.7),

$$\mathbb{P}\left[\sup_s \left| F_{S|\vartheta_n}(s) - \hat{F}_{S|\vartheta_n}^{(m)}(s) \right| \geq 2f_{\min}u \right] \leq 2\exp\left(-8mf_{\min}^2 u^2\right).$$

Thus, by Lemma A.6, given that the event $V$ occurs,

$$\mathbb{P}\left[|\hat{q}_{(1-\alpha)_m}(S_m \mid \vartheta_n) - q_{(1-\alpha)_m}(S \mid \vartheta_n)| \geq u \mid V \right] \leq 2\exp\left(-8mf_{\min}^2 u^2\right), \quad u \in [0, \beta/4].$$

Specifically,

$$\mathbb{P}\left[|\hat{q}_{(1-\alpha)_m}(S_m \mid \vartheta_n) - q_{(1-\alpha)_m}(S \mid \vartheta_n)| \geq \beta/4 \mid V \right] \leq 2\exp\left(-8mf_{\min}^2(\beta/4)^2\right).$$

Then, for any $u > \beta/4$,

$$\mathbb{P}\left[|\hat{q}_{(1-\alpha)_m}(S_m \mid \vartheta_n) - q_{(1-\alpha)_m}(S \mid \vartheta_n)| \geq u \mid V \right] \leq 2\exp\left(-8mf_{\min}^2(\beta/4)^2\right).$$

Since $|S| \leq R$, $|\hat{q}_{(1-\alpha)_m}(S_m \mid \vartheta_n) - q_{(1-\alpha)_m}(S \mid \vartheta_n)| \leq 2R$. By the layer cake representation of the expectation of a non-negative random variable $Z$, which is $\mathbb{E}[Z] = \int_0^\infty \mathbb{P}[Z \geq u] \, du$,

$$\mathbb{E}_{\vartheta_n, \mathcal{D}_{\text{cal}}} \left[ |\hat{q}_{(1-\alpha)_m}(S_m \mid \vartheta_n) - q_{(1-\alpha)_m}(S \mid \vartheta_n)| \, \Big| \, V \right]$$

$$= \int_0^{2R} \mathbb{P} \left[ |\hat{q}_{(1-\alpha)_m}(S_m \mid \vartheta_n) - q_{(1-\alpha)_m}(S \mid \vartheta_n)| \geq u \, \Big| \, V \right] du$$

$$\leq \int_0^{\beta/4} 2 \exp\left( -8m f_{\min}^2 u^2 \right) du + \int_{\beta/4}^{2R} 2 \exp\left( -8m f_{\min}^2 (\beta/4)^2 \right) du$$

$$\leq 2 \int_0^\infty \exp\left( -8m f_{\min}^2 u^2 \right) du + 4R \exp\left( -8 f_{\min}^2 (\beta/4)^2 m \right)$$

$$= \frac{\sqrt{\pi}}{2 f_{\min} \sqrt{2m}} + 4R \exp\left( -\frac{1}{2} f_{\min}^2 \beta^2 m \right).$$

Therefore, we have

$$\mathbb{E}_{\vartheta_n, \mathcal{D}_{\text{cal}}} \left[ |\hat{q}_{(1-\alpha)_m}(S_m \mid \vartheta_n) - q_{(1-\alpha)_m}(S \mid \vartheta_n)| \right]$$

$$\leq \mathbb{P}[V] \cdot \mathbb{E}_{\vartheta_n} \left[ |\hat{q}_{(1-\alpha)_m}(S_m \mid \vartheta_n) - q_{(1-\alpha)_m}(S \mid \vartheta_n)| \, \Big| \, V \right] + \mathbb{P}[V^c] \cdot 2R$$

$$\leq \frac{\sqrt{\pi}}{2 f_{\min} \sqrt{2m}} + 4R \exp\left( -\frac{1}{2} f_{\min}^2 \beta^2 m \right) + 2R\delta.$$

Picking $\delta = \frac{33AB^2}{\beta^2 n}$ completes the proof of Proposition A.5.

## A.5 PROOF OF THEOREM 3.2

**Theorem 3.2** (Efficiency of CQR-SGD). *For CQR-SGD, suppose Assumptions 3.1–3.3 hold. If $m > 8H/\min\{\alpha, 1-\alpha\}$, then for test sample $(X, Y)$ and $0 < \alpha \leq 1/2$,*

$$\mathbb{E}_{X, \vartheta_n, \mathcal{D}_{\text{cal}}} \left[ ||\mathcal{C}(X)| - |\mathcal{C}^*(X)|| \right] \leq \mathcal{O}\left( n^{-1/2} + (\alpha^2 n)^{-1} + m^{-1/2} + \exp(-\alpha^2 m) \right), \quad (17)$$

*where $H$ is the constant defined in (12).*

*Proof.* By the definition of the prediction set (9),

$$|\mathcal{C}(x)| = \max\left\{ 0, \, t_{1-\alpha/2}(x; \bar{\theta}_n) - t_{\alpha/2}(x; \underline{\theta}_n) + 2\hat{q}_{(1-\alpha)_m} \right\}$$

$$\leq \left| t_{1-\alpha/2}(x; \bar{\theta}_n) - t_{\alpha/2}(x; \underline{\theta}_n) + 2\hat{q}_{(1-\alpha)_m} \right|.$$

We further bound the right hand side by

$$\left| t_{1-\alpha/2}(x; \bar{\theta}_n) - t_{\alpha/2}(x; \underline{\theta}_n) + 2\hat{q}_{(1-\alpha)_m} \right|$$

$$= \Big| t_{1-\alpha/2}(x; \bar{\theta}_n) - t_{1-\alpha/2}(x; \bar{\theta}^*) + t_{1-\alpha/2}(x; \bar{\theta}^*) - t_{\alpha/2}(x; \underline{\theta}_n) + t_{\alpha/2}(x; \underline{\theta}^*) - t_{\alpha/2}(x; \underline{\theta}^*)$$

$$\quad + 2\hat{q}_{(1-\alpha)_m} \Big|$$

$$\leq \left| t_{1-\alpha/2}(x; \bar{\theta}_n) - t_{1-\alpha/2}(x; \bar{\theta}^*) \right| + \left| t_{\alpha/2}(x; \underline{\theta}_n) - t_{\alpha/2}(x; \underline{\theta}^*) \right| + 2 \left| \hat{q}_{(1-\alpha)_m} \right|$$

$$\quad + \left| t_{1-\alpha/2}(x; \bar{\theta}^*) - t_{\alpha/2}(x; \underline{\theta}^*) \right|$$

$$= \left| t_{1-\alpha/2}(x; \bar{\theta}_n) - t_{1-\alpha/2}(x; \bar{\theta}^*) \right| + \left| t_{\alpha/2}(x; \underline{\theta}_n) - t_{\alpha/2}(x; \underline{\theta}^*) \right| + 2 \left| \hat{q}_{(1-\alpha)_m} \right|$$

$$\quad + \left( t_{1-\alpha/2}(x; \bar{\theta}^*) - t_{\alpha/2}(x; \underline{\theta}^*) \right),$$

where the last equality follows because

$$t_{1-\alpha/2}(x; \bar{\theta}^*) = q_{1-\alpha/2}(Y \mid X) \geq q_{\alpha/2}(Y \mid X) = t_{\alpha/2}(x; \underline{\theta}^*).$$

Hence,

$$|\mathcal{C}(X)| - \left( t_{1-\alpha/2}(x; \bar{\theta}^*) - t_{\alpha/2}(x; \underline{\theta}^*) \right)$$

$$\leq \left| t_{1-\alpha/2}(x; \bar{\theta}_n) - t_{1-\alpha/2}(x; \bar{\theta}^*) \right| + \left| t_{\alpha/2}(x; \underline{\theta}_n) - t_{\alpha/2}(x; \underline{\theta}^*) \right| + 2 \left| \hat{q}_{(1-\alpha)_m} \right|.$$

We also have

$$
\begin{aligned}
&- \left( |\mathcal{C}(X)| - \left( t_{1-\alpha/2}\left(x;\bar{\theta}^*\right) - t_{\alpha/2}\left(x;\underline{\theta}^*\right) \right) \right) \\
&= \left( t_{1-\alpha/2}\left(x;\bar{\theta}^*\right) - t_{\alpha/2}\left(x;\underline{\theta}^*\right) \right) - \max\left\{ 0,\ t_{1-\alpha/2}\left(x;\bar{\theta}_n\right) - t_{\alpha/2}\left(x;\underline{\theta}_n\right) + 2\hat{q}_{(1-\alpha)_m} \right\} \\
&\leq t_{1-\alpha/2}\left(x;\bar{\theta}^*\right) - t_{\alpha/2}\left(x;\underline{\theta}^*\right) - t_{1-\alpha/2}\left(x;\bar{\theta}_n\right) + t_{\alpha/2}\left(x;\underline{\theta}_n\right) - 2\hat{q}_{(1-\alpha)_m} \\
&\leq \left| t_{1-\alpha/2}\left(x;\bar{\theta}_n\right) - t_{1-\alpha/2}\left(x;\bar{\theta}^*\right) \right| + \left| t_{\alpha/2}\left(x;\underline{\theta}_n\right) - t_{\alpha/2}\left(x;\underline{\theta}^*\right) \right| + 2\left| \hat{q}_{(1-\alpha)_m} \right|.
\end{aligned}
$$

Therefore,

$$
\begin{aligned}
&\left| |\mathcal{C}(X)| - \left( t_{1-\alpha/2}\left(x;\bar{\theta}^*\right) - t_{\alpha/2}\left(x;\underline{\theta}^*\right) \right) \right| \\
&\qquad \leq \left| t_{1-\alpha/2}\left(x;\bar{\theta}_n\right) - t_{1-\alpha/2}\left(x;\bar{\theta}^*\right) \right| + \left| t_{\alpha/2}\left(x;\underline{\theta}_n\right) - t_{\alpha/2}\left(x;\underline{\theta}^*\right) \right| + 2\left| \hat{q}_{(1-\alpha)_m} \right|.
\end{aligned}
$$

Hence, for test sample $(X, Y)$,

$$
\begin{aligned}
&\mathbb{E}_{X,\vartheta_n,\mathcal{D}_{\mathrm{cal}}}\left[ \left| |\mathcal{C}(X)| - t_{1-\alpha/2}\left(X;\bar{\theta}^*\right) - t_{\alpha/2}\left(X;\underline{\theta}^*\right) \right| \right] \\
&\leq \mathbb{E}_{X,\vartheta_n}\left[ \left| t_{1-\alpha/2}\left(X;\bar{\theta}_n\right) - t_{1-\alpha/2}\left(X;\bar{\theta}^*\right) \right| \right] + \mathbb{E}_{X,\vartheta_n}\left[ \left| t_{\alpha/2}\left(X;\underline{\theta}_n\right) - t_{\alpha/2}\left(X;\underline{\theta}^*\right) \right| \right] \\
&\qquad + 2\mathbb{E}_{\vartheta_n,\mathcal{D}_{\mathrm{cal}}}\left[ \left| \hat{q}_{(1-\alpha)_m}\left(S_m \mid \vartheta_n\right) \right| \right] \\
&\leq \mathbb{E}_{X,\vartheta_n}\left[ \left| t_{1-\alpha/2}\left(X;\bar{\theta}_n\right) - t_{1-\alpha/2}\left(X;\bar{\theta}^*\right) \right| \right] + \mathbb{E}_{X,\vartheta_n}\left[ \left| t_{\alpha/2}\left(X;\underline{\theta}_n\right) - t_{\alpha/2}\left(X;\underline{\theta}^*\right) \right| \right] \\
&\qquad + 2\mathbb{E}_{\vartheta_n}\left[ \left| q_{1-\alpha}\left(S \mid \vartheta_n\right) \right| \right] + 2\mathbb{E}_{\vartheta_n,\mathcal{D}_{\mathrm{cal}}}\left[ \left| q_{1-\alpha}\left(S \mid \vartheta_n\right) - \hat{q}_{(1-\alpha)_m}\left(S_m \mid \vartheta_n\right) \right| \right] \\
&\leq \mathbb{E}_{X,\vartheta_n}\left[ \left| t_{1-\alpha/2}\left(X;\bar{\theta}_n\right) - t_{1-\alpha/2}\left(X;\bar{\theta}^*\right) \right| \right] + \mathbb{E}_{X,\vartheta_n}\left[ \left| t_{\alpha/2}\left(X;\underline{\theta}_n\right) - t_{\alpha/2}\left(X;\underline{\theta}^*\right) \right| \right] \\
&\qquad + 2\mathbb{E}_{\vartheta_n}\left[ \left| q_{1-\alpha}\left(S \mid \vartheta_n\right) \right| \right] + 2\mathbb{E}_{\vartheta_n}\left[ \left| q_{1-\alpha}\left(S \mid \vartheta_n\right) - q_{(1-\alpha)_m}\left(S \mid \vartheta_n\right) \right| \right] \\
&\qquad + 2\mathbb{E}_{\vartheta_n,\mathcal{D}_{\mathrm{cal}}}\left[ \left| q_{(1-\alpha)_m}\left(S \mid \vartheta_n\right) - \hat{q}_{(1-\alpha)_m}\left(S_m \mid \vartheta_n\right) \right| \right].
\end{aligned}
$$

By Theorem 3.1,

$$
\begin{aligned}
\mathbb{E}_{X,\theta_n}\left[ \left| t_\gamma\left(X;\theta_n\left(\gamma\right)\right) - t_\gamma\left(X;\theta^*\left(\gamma\right)\right) \right| \right] &\leq \sqrt{\mathbb{E}_{X,\theta_n}\left[ \left( t_\gamma\left(X;\theta_n\left(\gamma\right)\right) - t_\gamma\left(X;\theta^*\left(\gamma\right)\right) \right)^2 \right]} \\
&\leq \frac{2\lambda_{\max}\sqrt{f_{\max}d}}{\lambda_{\min}f_{\min}\sqrt{\lambda_{\min}n}}.
\end{aligned}
$$

By Proposition A.2,A.3,A.5,

$$
\begin{aligned}
&\mathbb{E}_{X,\vartheta_n,\mathcal{D}_{\mathrm{cal}}}\left[ \left| |\mathcal{C}(X)| - t_{1-\alpha/2}\left(X;\bar{\theta}^*\right) - t_{\alpha/2}\left(X;\underline{\theta}^*\right) \right| \right] \\
&\leq \left( \frac{4\lambda_{\max}\sqrt{f_{\max}d}}{\lambda_{\min}f_{\min}\sqrt{\lambda_{\min}}} + \frac{2B\,\lambda_{\max}\sqrt{2f_{\max}d}}{\lambda_{\min}^2 f_{\min}} \right)\sqrt{\frac{1}{n}} + \frac{1}{f_{\min}m} \\
&\qquad + \frac{\sqrt{\pi}}{2f_{\min}\sqrt{2m}} + 4R\exp\left( -\frac{1}{2}f_{\min}^2\beta^2 m \right) + \frac{66AB^2R}{\beta^2 n} \\
&= \left( \frac{4\lambda_{\max}\sqrt{f_{\max}d}}{\lambda_{\min}f_{\min}\sqrt{\lambda_{\min}}} + \frac{2B\,\lambda_{\max}\sqrt{2f_{\max}d}}{\lambda_{\min}^2 f_{\min}} \right)\sqrt{\frac{1}{n}} + \frac{\sqrt{\pi}}{2f_{\min}\sqrt{2}}\sqrt{\frac{1}{m}} + \frac{1}{f_{\min}m} \\
&\qquad + 4R\exp\left( -\frac{\min\{\alpha^2,(1-\alpha)^2\}f_{\min}^2}{8f_{\max}^2}m \right) + \frac{1056\lambda_{\max}^2 f_{\max}^3 B^2 R}{\min\{\alpha^2,(1-\alpha)^2\}\lambda_{\min}^4 f_{\min}^2 n}. \qquad (41)
\end{aligned}
$$

This completes the proof of Theorem 3.2. $\qquad\qquad\square$

# B  PROOFS OF RESULTS IN CMR

To prove Theorem 4.1, the goal is to upper bound

$$
\begin{aligned}
&\mathbb{E}_{X,\breve{\theta}_n,\mathcal{D}_{\mathrm{cal}}}\left[ \left| 2\,\hat{q}_{(1-\alpha)_m}\left(S \mid \breve{\theta}_n\right) - \left( q_{1-\alpha/2}\left(Y \mid X\right) - q_{\alpha/2}\left(Y \mid X\right) \right) \right| \right] \\
&= 2\,\mathbb{E}_{X,\breve{\theta}_n,\mathcal{D}_{\mathrm{cal}}}\left[ \left| \hat{q}_{(1-\alpha)_m}\left(S \mid \breve{\theta}_n\right) - \left( q_{1/2}\left(Y \mid X\right) - q_{\alpha/2}\left(Y \mid X\right) \right) \right| \right].
\end{aligned}
$$

Further decompose it, and we have

$$\left| \hat{q}_{(1-\alpha)_m} \left( S \mid \check{\theta}_n \right) - \left( q_{1/2} \left( Y \mid X \right) - q_{\alpha/2} \left( Y \mid X \right) \right) \right|$$
$$= \left| \hat{q}_{(1-\alpha)_m} \left( S \mid \check{\theta}_n \right) - q_{(1-\alpha)_m} \left( S \mid \check{\theta}_n \right) + q_{(1-\alpha)_m} \left( S \mid \check{\theta}_n \right) - q_{1-\alpha} \left( S \mid \check{\theta}_n \right) \right.$$
$$\left. + q_{1-\alpha} \left( S \mid \check{\theta}_n \right) - \left( q_{1/2} \left( Y \mid X \right) - q_{\alpha/2} \left( Y \mid X \right) \right) \right|$$
$$\leq \left| \hat{q}_{(1-\alpha)_m} \left( S \mid \check{\theta}_n \right) - q_{(1-\alpha)_m} \left( S \mid \check{\theta}_n \right) \right| + \left| q_{(1-\alpha)_m} \left( S \mid \check{\theta}_n \right) - q_{1-\alpha} \left( S \mid \check{\theta}_n \right) \right|$$
$$+ \left| q_{1-\alpha} \left( S \mid \check{\theta}_n \right) - \left( q_{1/2} \left( Y \mid X \right) - q_{\alpha/2} \left( Y \mid X \right) \right) \right|.$$

Thus, the expectation is decomposed into three parts as follows, and we will upper bound each of them in Proposition B.4, B.3, and B.1:

$$\mathbb{E}_{X,\check{\theta}_n,\mathcal{D}_{\mathrm{cal}}} \left[ \left| 2\, \hat{q}_{(1-\alpha)_m} \left( S \mid \check{\theta}_n \right) - \left( q_{1-\alpha/2} \left( Y \mid X \right) - q_{\alpha/2} \left( Y \mid X \right) \right) \right| \right]$$
$$= 2\, \mathbb{E}_{\check{\theta}_n,\mathcal{D}_{\mathrm{cal}}} \left[ \left| \hat{q}_{(1-\alpha)_m} \left( S \mid \check{\theta}_n \right) - q_{(1-\alpha)_m} \left( S \mid \check{\theta}_n \right) \right| \right]$$
$$+ 2\, \mathbb{E}_{\check{\theta}_n} \left[ \left| q_{(1-\alpha)_m} \left( S \mid \check{\theta}_n \right) - q_{1-\alpha} \left( S \mid \check{\theta}_n \right) \right| \right]$$
$$+ 2\, \mathbb{E}_{X,\check{\theta}_n} \left[ \left| q_{1-\alpha} \left( S \mid \check{\theta}_n \right) - \left( q_{1/2} \left( Y \mid X \right) - q_{\alpha/2} \left( Y \mid X \right) \right) \right| \right]$$
$$\leq \frac{\sqrt{\pi}}{f_{\min}\sqrt{2m}} + 8R \exp\left( -\frac{f_{\min}^2 \min\{\alpha^2,(1-\alpha)^2\}}{8 f_{\max}^2} m \right) + \frac{2056 R \lambda_{\max}^2 f_{\max}^3 B^2 d}{\lambda_{\min}^4 f_{\min}^2 \min\{\alpha^2,(1-\alpha)^2\} n}$$
$$+ \frac{2}{f_{\min} m} + \frac{4B\, \lambda_{\max}\sqrt{f_{\max}d}}{\lambda_{\min}^2 f_{\min}} \sqrt{\frac{1}{n}}. \tag{42}$$

To proceed, we define some random variables for simplicity.

$$\Delta\left(X,\check{\theta}_n\right) := \left| t_{1/2}\left(X;\check{\theta}_n\right) - t_{1/2}\left(X;\check{\theta}^*\right) \right| \geq 0; \tag{43}$$
$$S^*\left(X,Y\right) := \left| q_{1/2}(Y \mid X) - Y \right|; \tag{44}$$
$$M\left(\check{\theta}_n\right) := \left\| \left(\check{\theta}_n - \check{\theta}^*\right) \right\|_2. \tag{45}$$

## B.1 PROOF OF PROPOSITION B.1

**Proposition B.1.** *In CMR, suppose Assumption 4.2 holds, we have*

$$\left| q_{1-\alpha}\left(S \mid X,\check{\theta}_n\right) - \zeta \right| \leq B \cdot M\left(\check{\theta}_n\right). \tag{46}$$

*If Assumptions 4.1,3.2,3.3 further hold, then*

$$\mathbb{E}_{X,\check{\theta}_n} \left[ \left| q_{1-\alpha}\left(S \mid \check{\theta}_n\right) - \left( q_{1/2}\left(Y \mid X\right) - q_{\alpha/2}\left(Y \mid X\right) \right) \right| \right] \leq \frac{2B\, \lambda_{\max}\sqrt{f_{\max}d}}{\lambda_{\min}^2 f_{\min}} \sqrt{\frac{1}{n}}. \tag{47}$$

*Proof.* Notice that

$$S\left(X,Y;\check{\theta}_n\right) := \left| t_{1/2}\left(X;\check{\theta}_n\right) - Y \right|$$
$$\leq \left| q_{1/2}(Y \mid X) - Y \right| + \left| t_{1/2}\left(X;\check{\theta}_n\right) - q_{1/2}(Y \mid X) \right|$$
$$= S^*\left(X,Y\right) + \Delta\left(X,\check{\theta}_n\right).$$

Similarly, $S\left(X,Y;\check{\theta}_n\right) \geq S^*\left(X,Y\right) - \Delta\left(X,\check{\theta}_n\right)$. Hence,

$$\left| S\left(X,Y;\check{\theta}_n\right) - S^*\left(X,Y\right) \right| \leq \Delta\left(X,\check{\theta}_n\right) \leq \|X\|_2 \left\| \left(\check{\theta}_n - \check{\theta}^*\right) \right\|_2 \leq B \cdot \left\| \left(\check{\theta}_n - \check{\theta}^*\right) \right\|_2.$$

Now we show that $q_{1-\alpha}\left(S^* \mid X\right) = q_{1/2}(Y \mid X) - q_{\alpha/2}(Y \mid X)$. Note that given $X$,

$$S^*\left(X,Y\right) \leq q_{1/2}(Y \mid X) - q_{\alpha/2}(Y \mid X)$$
$$\iff -\left( q_{1/2}(Y \mid X) - q_{\alpha/2}(Y \mid X) \right) \leq Y - q_{1/2}(Y \mid X) \leq q_{1/2}(Y \mid X) - q_{\alpha/2}(Y \mid X)$$
$$\iff q_{\alpha/2}(Y \mid X) \leq Y \leq q_{1-\alpha/2}(Y \mid X),$$

where the last step is from Assumption 4.2. Since $F_{Y|X}$ is continuous,

$$\mathbb{P}\left[ q_{\alpha/2}\left(Y \mid X\right) \leq Y \leq q_{1-\alpha/2}\left(Y \mid X\right) \mid X \right] = 1 - \alpha.$$

Hence,
$$\mathbb{P}[S^*(X,Y) \leq q_{1/2}(Y \mid X) - q_{\alpha/2}(Y \mid X)|X] = 1 - \alpha.$$

Let $q_{1-\alpha}(S^* \mid X)$ be the $(1-\alpha)$-quantile of $S^*$ given $X$. Since $X$ is given, and $F_{Y|X}$ is continuous, $F_{S^*|X}$ is continuous. Then, $q_{1-\alpha}(S^* \mid X) = q_{1/2}(Y \mid X) - q_{\alpha/2}(Y \mid X)$.

Conditioned on $X, \breve{\theta}_n, \Delta(X, \breve{\theta}_n)$ is deterministic. Thus,

$$\mathbb{P}\left[S\left(X,Y;\breve{\theta}_n\right) \leq u \mid X, \breve{\theta}_n\right] \geq \mathbb{P}\left[S^*(X,Y) + \Delta\left(X, \breve{\theta}_n\right) \leq u \mid X, \breve{\theta}_n\right]$$

$$\Rightarrow \mathbb{P}\left[S\left(X,Y;\breve{\theta}_n\right) \leq \Delta\left(X, \breve{\theta}_n\right) + q_{1/2}(Y \mid X) - q_{\alpha/2}(Y \mid X) \mid X, \breve{\theta}_n\right]$$

$$\geq \mathbb{P}\left[S^*(X,Y) \leq q_{1/2}(Y \mid X) - q_{\alpha/2}(Y \mid X) \mid X\right] = 1 - \alpha.$$

Then, $q_{1-\alpha}\left(S \mid X, \breve{\theta}_n\right) \leq \Delta\left(X, \breve{\theta}_n\right) + q_{1/2}(Y \mid X) - q_{\alpha/2}(Y \mid X)$. Similarly, we have

$$\mathbb{P}\left[S\left(X,Y;\breve{\theta}_n\right) \leq u \mid X, \breve{\theta}_n\right] \leq \mathbb{P}\left[S^*(X,Y) - \Delta\left(X, \breve{\theta}_n\right) \leq u \mid X, \breve{\theta}_n\right]$$

$$\Rightarrow \mathbb{P}\left[S\left(X,Y;\breve{\theta}_n\right) \leq -\Delta\left(X, \breve{\theta}_n\right) + q_{1/2}(Y \mid X) - q_{\alpha/2}(Y \mid X) \mid X, \breve{\theta}_n\right]$$

$$\leq \mathbb{P}\left[S^*(X,Y) \leq q_{1/2}(Y \mid X) - q_{\alpha/2}(Y \mid X) \mid X\right] = 1 - \alpha.$$

Then, $q_{1-\alpha}\left(S \mid X, \breve{\theta}_n\right) \geq -\Delta\left(X, \breve{\theta}_n\right) + q_{1/2}(Y \mid X) - q_{\alpha/2}(Y \mid X)$. Thus, by Assumption 4.2,

$$\left|q_{1-\alpha}\left(S \mid X, \breve{\theta}_n\right) - \left(q_{1/2}(Y \mid X) - q_{\alpha/2}(Y \mid X)\right)\right| \leq \Delta\left(X, \breve{\theta}_n\right)$$

$$\implies \left|q_{1-\alpha}\left(S \mid X, \breve{\theta}_n\right) - \zeta\right| \leq B \cdot M\left(\breve{\theta}_n\right).$$

Then we can remove the conditioning on $X$,

$$\mathbb{P}\left[S\left(X,Y;\breve{\theta}_n\right) \leq \zeta + B \cdot M\left(\breve{\theta}_n\right) \mid \breve{\theta}_n\right]$$

$$= \mathbb{E}_{X,Y|\breve{\theta}_n}\left[\mathbb{1}\left\{S\left(X,Y;\breve{\theta}_n\right) \leq \zeta + B \cdot M\left(\breve{\theta}_n\right)\right\} \mid \breve{\theta}_n\right]$$

$$= \mathbb{E}_{X|\breve{\theta}_n}\left[\mathbb{E}_{Y|X,\breve{\theta}_n}\left[\mathbb{1}\left\{S\left(X,Y;\breve{\theta}_n\right) \leq \zeta + B \cdot M\left(\breve{\theta}_n\right)\right\} \mid X, \breve{\theta}_n\right] \mid \breve{\theta}_n\right]$$

$$= \mathbb{E}_{X|\breve{\theta}_n}\left[\mathbb{P}\left[S\left(X,Y;\breve{\theta}_n\right) \leq \zeta + B \cdot M\left(\breve{\theta}_n\right) \mid X, \breve{\theta}_n\right] \mid \breve{\theta}_n\right]$$

$$\geq \mathbb{E}_{X|\breve{\theta}_n}\left[1 - \alpha \mid \breve{\theta}_n\right] = 1 - \alpha.$$

Hence, $q_{1-\alpha}\left(S \mid \breve{\theta}_n\right) \leq \zeta + B \cdot M\left(\breve{\theta}_n\right)$. And by similar arguments as below, $q_{1-\alpha}\left(S \mid \breve{\theta}_n\right) \geq \zeta - B \cdot M\left(\breve{\theta}_n\right)$. Specifically,

$$\mathbb{P}\left[S\left(X,Y;\breve{\theta}_n\right) \geq \zeta - B \cdot M\left(\breve{\theta}_n\right) \mid \breve{\theta}_n\right]$$

$$= \mathbb{E}_{X,Y|\breve{\theta}_n}\left[\mathbb{1}\left\{S\left(X,Y;\breve{\theta}_n\right) \geq \zeta - B \cdot M\left(\breve{\theta}_n\right)\right\} \mid \breve{\theta}_n\right]$$

$$= \mathbb{E}_{X|\breve{\theta}_n}\left[\mathbb{E}_{Y|X,\breve{\theta}_n}\left[\mathbb{1}\left\{S\left(X,Y;\breve{\theta}_n\right) \geq \zeta - B \cdot M\left(\breve{\theta}_n\right)\right\} \mid X, \breve{\theta}_n\right] \mid \breve{\theta}_n\right]$$

$$= \mathbb{E}_{X|\breve{\theta}_n}\left[\mathbb{P}\left[S\left(X,Y;\breve{\theta}_n\right) \geq \zeta - B \cdot M\left(\breve{\theta}_n\right) \mid X, \breve{\theta}_n\right] \mid \breve{\theta}_n\right]$$

$$\leq \mathbb{E}_{X|\breve{\theta}_n}\left[1 - \alpha \mid \breve{\theta}_n\right] = 1 - \alpha.$$

Therefore, $\left|q_{1-\alpha}\left(S \mid \breve{\theta}_n\right) - \zeta\right| \leq B \cdot M\left(\breve{\theta}_n\right)$.

Then, by Theorem 3.1,

$$\mathbb{E}_{\breve{\theta}_n}\left[\left|q_{1-\alpha}\left(S \mid \breve{\theta}_n\right) - \zeta\right|\right] \leq B \cdot \mathbb{E}_{\breve{\theta}_n}\left[M\left(\breve{\theta}_n\right)\right] \leq B\sqrt{\mathbb{E}_{\breve{\theta}_n}\left[\|(\underline{\theta}_n - \underline{\theta}^*)\|_2^2\right]}$$

$$\leq B\sqrt{\frac{4\lambda_{\max}^2 f_{\max}d}{\lambda_{\min}^4 f_{\min}^2 n}} = \frac{2B\,\lambda_{\max}\sqrt{f_{\max}d}}{\lambda_{\min}^2 f_{\min}}\sqrt{\frac{1}{n}},$$

i.e.,

$$\mathbb{E}_{X,\breve{\theta}_n}\left[\left|q_{1-\alpha}\left(S \mid \breve{\theta}_n\right) - \left(q_{1/2}(Y \mid X) - q_{\alpha/2}(Y \mid X)\right)\right|\right] \leq \frac{2B\,\lambda_{\max}\sqrt{f_{\max}d}}{\lambda_{\min}^2 f_{\min}}\sqrt{\frac{1}{n}}.$$

$\square$

## B.2 PROOF OF PROPOSITION B.2

**Proposition B.2.** *In CMR, suppose Assumption 4.1,3.2,3.3,4.2 hold. Define*

$$\beta := \min\left\{\frac{\alpha}{2f_{\max}}, \frac{1-\alpha}{2f_{\max}}\right\}, \qquad \epsilon_n := B\sqrt{\frac{A}{n\delta}}. \tag{48}$$

*If $\epsilon_n < \beta/4$, then with probability at least $1-\delta$, for any $s$ such that for $s \in \mathcal{I} := \{s \in \mathbb{R} : |s-\zeta| \leq \beta - \epsilon_n\}$, $f_{S|\check{\theta}_n}(s) \geq 2f_{\min}$, and $\left|q_{1-\alpha}\left(S \mid \check{\theta}_n\right) - \zeta\right| \leq \epsilon_n \leq \beta - \epsilon_n$.*

*Proof.* By the definition of $S$,

$$\mathbb{P}\left[S \leq s|X,\check{\theta}_n\right] = \mathbb{P}\left[t_{1/2}\left(X;\check{\theta}_n\right) - s \leq Y \leq t_{1/2}\left(X;\check{\theta}_n\right) + s \mid X,\check{\theta}_n\right].$$

Hence,

$$F_{S|X,\check{\theta}_n}(s) = F_{Y|X,\check{\theta}_n}\left(t_{1/2}\left(x;\check{\theta}_n\right) + s\right) - F_{Y|X,\check{\theta}_n}\left(t_{1/2}\left(x;\check{\theta}_n\right) - s\right). \tag{49}$$

We now show that with high probability, it holds for $s$ in the neighbourhood of $\zeta$ that

$$t_{1/2}\left(x;\check{\theta}_n\right) + s \in \mathcal{Y}, \quad t_{1/2}\left(x;\check{\theta}_n\right) - s \in \mathcal{Y}.$$

By Theorem 3.1, $\mathbb{E}_{\check{\theta}_n}\left[\|\check{\theta}_n - \check{\theta}^*\|_2^2\right] \leq \frac{A}{n}$ for $A := \frac{4\lambda_{\max}^2 f_{\max} d}{\lambda_{\min}^4 f_{\min}^2}$. By Markov's inequality,

$$\mathbb{P}\left[\|\check{\theta}_n - \check{\theta}^*\|_2 \leq \sqrt{\frac{A}{n\delta}}\right] \geq 1 - \delta.$$

Hence, with probability at least $1 - \delta$,

$$\sup_x \Delta\left(x, \check{\theta}_n\right) \leq B\|\check{\theta}_n - \check{\theta}^*\|_2 \leq B\sqrt{\frac{A}{n\delta}} =: \epsilon_n.$$

In this case, by (46),

$$\left|q_{1-\alpha}\left(S \mid \check{\theta}_n\right) - \zeta\right| \leq \epsilon_n. \tag{50}$$

Then, for every $s$ such that $|s - \zeta| \leq \beta - \epsilon_n$, i.e., $s \in \mathcal{I}$, it holds that

$$t_{1/2}\left(x;\check{\theta}_n\right) + s \leq q_{1/2}\left(Y|X\right) + \epsilon_n + \zeta + \beta - \epsilon_n = q_{1/2}\left(Y|X\right) + \zeta + \beta = q_{1-\alpha/2}(Y|X) + \beta \leq y_{\max};$$
$$t_{1/2}\left(x;\check{\theta}_n\right) + s \geq q_{1/2}\left(Y|X\right) - \epsilon_n + \zeta - \beta + \epsilon_n = q_{1/2}\left(Y|X\right) + \zeta - \beta = q_{1-\alpha/2}\left(Y|X\right) - \beta \geq y_{\min};$$
$$t_{1/2}\left(x;\check{\theta}_n\right) - s \leq q_{1/2}\left(Y|X\right) + \epsilon_n - \zeta + \beta - \epsilon_n = q_{1/2}\left(Y|X\right) - \zeta + \beta = q_{\alpha/2}(Y|X) + \beta \leq y_{\max};$$
$$t_{1/2}\left(x;\check{\theta}_n\right) - s \geq q_{1/2}\left(Y|X\right) - \epsilon_n - \zeta - \beta + \epsilon_n = q_{1/2}\left(Y|X\right) - \zeta - \beta = q_{\alpha/2}\left(Y|X\right) - \beta \geq y_{\min}.$$

Thus, $t_{1/2}\left(x;\check{\theta}_n\right) + s \in \mathcal{Y}, t_{1/2}\left(x;\check{\theta}_n\right) - s \in \mathcal{Y}$.

By (49), if $\epsilon_n < \beta/4$, then with probability at least $1-\delta$, we have for any $s$ such that $|s-\zeta| \leq \beta-\epsilon_n$,

$$f_{S|X,\check{\theta}_n}(s) = f_{Y|X,\check{\theta}_n}\left(t_{1/2}\left(x;\check{\theta}_n\right) + s\right) + f_{Y|X,\check{\theta}_n}\left(t_{1/2}\left(x;\check{\theta}_n\right) - s\right) \geq 2f_{\min}. \tag{51}$$

Since $\left|q_{1-\alpha}\left(S \mid \check{\theta}_n\right) - \zeta\right| \leq \epsilon_n \leq \beta - \epsilon_n < \frac{3}{4}\beta$, after taking expectation over $X$, we have $f_{S|\check{\theta}_n}\left(q_{1-\alpha}\left(S \mid \check{\theta}_n\right) - \zeta\right) \geq 2f_{\min}$. $\quad\square$

## B.3 PROOF OF PROPOSITION B.3

**Proposition B.3.** *In CMR, suppose Assumption 4.1,3.2,3.3,4.2 hold. If*

$$m > \frac{8f_{\max}}{f_{\min}\min\{\alpha,(1-\alpha)\}}. \tag{52}$$

*then*

$$\mathbb{E}_{\check{\theta}_n}\left[\left|q_{(1-\alpha)_m}\left(S \mid \check{\theta}_n\right) - q_{1-\alpha}\left(S \mid \check{\theta}_n\right)\right|\right] \leq \frac{1}{f_{\min}m} + \frac{514R\lambda_{\max}^2 f_{\max}^3 B^2 d}{\lambda_{\min}^4 f_{\min}^2 \min\{\alpha^2,(1-\alpha)^2\}n}, \tag{53}$$

*and if furthermore $n > \frac{256\lambda_{\max}^2 f_{\max}^3 B^2 d}{\lambda_{\min}^4 f_{\min}^2 \min\{\alpha^2,(1-\alpha)^2\}\delta}$, then with probability at least $1 - \delta$,*

$$\left|q_{(1-\alpha)_m}\left(S \mid \check{\theta}_n\right) - q_{1-\alpha}\left(S \mid \check{\theta}_n\right)\right| \leq \frac{1}{f_{\min}m} < \frac{\beta}{4}.$$

*Proof.* Notice that

$$0 < \frac{1-\alpha}{m} \leq |(1-\alpha)_m - (1-\alpha)| < \frac{2-\alpha}{m} < \frac{2}{m}.$$

If let $m > \frac{4}{\beta f_{\min}}$ for $\beta$ defined in (48), then

$$|(1-\alpha)_m - (1-\alpha)| < \frac{2}{m} < 2f_{\min} \cdot \frac{\beta}{4}.$$

According to Lemma A.4, since $|q_{1-\alpha}(S \mid \check{\theta}_n) - \zeta| \leq \epsilon_n < \frac{\beta}{4}$ by Proposition B.2, the distance from $\mathcal{I}^c$ is $r_0 > \frac{\beta}{2}$. Thus, by Lemma A.4, $|q_{(1-\alpha)_m}(S \mid \check{\theta}_n) - q_{1-\alpha}(S \mid \check{\theta}_n)| \leq \frac{1}{f_{\min}m} < \frac{\beta}{4}$, and hence, $|q_{(1-\alpha)_m}(S \mid \check{\theta}_n) - \zeta| < \frac{\beta}{2}$.

Therefore, if $\epsilon_n < \beta/4$ and $m > \frac{4}{f_{\min}\beta}$, then

$$\mathbb{E}_{\check{\theta}_n}\left[|q_{(1-\alpha)_m}(S \mid \check{\theta}_n) - q_{1-\alpha}(S \mid \check{\theta}_n)|\right] \leq \frac{1}{f_{\min}m} + 2R\delta.$$

Taking $\delta = \frac{257\lambda_{\max}^2 f_{\max}^3 B^2 d}{\lambda_{\min}^4 f_{\min}^2 \min\{\alpha^2,(1-\alpha)^2\}n}$, and we get

$$\mathbb{E}_{\check{\theta}_n}\left[|q_{(1-\alpha)_m}(S \mid \check{\theta}_n) - q_{1-\alpha}(S \mid \check{\theta}_n)|\right] \leq \frac{1}{f_{\min}m} + \frac{514R\lambda_{\max}^2 f_{\max}^3 B^2 d}{\lambda_{\min}^4 f_{\min}^2 \min\{\alpha^2,(1-\alpha)^2\}n}.$$

$\square$

### B.4 PROOF OF PROPOSITION B.4

**Proposition B.4.** *In CMR, suppose Assumption 4.1,3.2,3.3,4.2 hold. If*

$$m > \frac{8H}{\min\{\alpha,(1-\alpha)\}} \tag{54}$$

*for $H$ in (12), then*

$$\mathbb{E}_{\check{\theta}_n,\mathcal{D}_{\mathrm{cal}}}\left[|\hat{q}_{(1-\alpha)_m}(S_m \mid \check{\theta}_n) - q_{(1-\alpha)_m}(S \mid \check{\theta}_n)|\right]$$
$$\leq \frac{\sqrt{\pi}}{2f_{\min}\sqrt{2m}} + 4R\exp\left(-\frac{f_{\min}^2 \min\{\alpha^2,(1-\alpha)^2\}}{8f_{\max}^2}m\right) + \frac{514Rf_{\max}^3\lambda_{\max}^2 B^2 d}{\min\{\alpha^2,(1-\alpha)^2\}\lambda_{\min}^4 f_{\min}^2 n}.$$

The proof of Proposition B.4 is essentially the same as the proof of Proposition A.5. We include here for completeness.

*Proof.*

**Lemma B.1.** *In CMR, under the same setting of Proposition B.2, if the high probability event $V$ in Proposition B.2 occurs, for any $u \in [0,\beta/4]$, if*

$$\sup_s \left| F_{S|\check{\theta}_n}(s) - \hat{F}_{S|\check{\theta}_n}^{(m)}(s) \right| \leq 2f_{\min}u,$$

*then $|\hat{q}_{(1-\alpha)_m}(S_m \mid \check{\theta}_n) - q_{(1-\alpha)_m}(S \mid \check{\theta}_n)| \leq u$.*

*Proof.* For simplicity, in the proof we denote $q_p(S \mid \check{\theta}_n)$ by $q_p$. By Proposition B.3, for $u \in [0,\beta/4]$, $|q_{(1-\alpha)_m} - \zeta - u| \leq 3\beta/4$ and $|q_{(1-\alpha)_m} - \zeta + u| \leq 3\beta/4$, i.e., $q_{(1-\alpha)_m} - u \in \mathcal{I}$ and $q_{(1-\alpha)_m} + u \in \mathcal{I}$ for $\mathcal{I}$ defined in Proposition B.2. Hence, in this case,

$$F_{S|\check{\theta}_n}\left(q_{(1-\alpha)_m} - u\right) \leq F_{S|\check{\theta}_n}\left(q_{(1-\alpha)_m}\right) - 2f_{\min}u = (1-\alpha)_m - 2f_{\min}u;$$
$$F_{S|\check{\theta}_n}\left(q_{(1-\alpha)_m} + u\right) \geq F_{S|\check{\theta}_n}\left(q_{(1-\alpha)_m}\right) + 2f_{\min}u = (1-\alpha)_m + 2f_{\min}u.$$

By assumption,

$$\left| F_{S|\check{\theta}_n} \left( q_{(1-\alpha)_m} - u \right) - \hat{F}_{S|\check{\theta}_n}^{(m)} \left( q_{(1-\alpha)_m} - u \right) \right| \le 2 f_{\min} u;$$

$$\left| F_{S|\check{\theta}_n} \left( q_{(1-\alpha)_m} + u \right) - \hat{F}_{S|\check{\theta}_n}^{(m)} \left( q_{(1-\alpha)_m} + u \right) \right| \le 2 f_{\min} u.$$

Then

$$\hat{F}_{S|\check{\theta}_n}^{(m)} \left( q_{(1-\alpha)_m} - u \right) \le (1-\alpha)_m, \qquad \hat{F}_{S|\check{\theta}_n}^{(m)} \left( q_{(1-\alpha)_m} + u \right) \ge (1-\alpha)_m.$$

Since $\hat{F}_{S|\check{\theta}_n}^{(m)}$ is non-decreasing, we have

$$\hat{q}_{(1-\alpha)_m} \left( S_m \mid \check{\theta}_n \right) := \inf\{u' \in \mathcal{S}_m : \hat{F}_{S|\check{\theta}_n}^{(m)} (u') \ge (1-\alpha)_m\} \in \left[ q_{(1-\alpha)_m} - u, q_{(1-\alpha)_m} + u \right].$$

where $\mathcal{S}_m$ is the set of scores of the calibration data. Then, $|\hat{q}_{(1-\alpha)_m} \left( S_m \mid \check{\theta}_n \right) - q_{(1-\alpha)_m} \left( S \mid \check{\theta}_n \right)| \le u$. $\qquad\square$

By the Dvoretzky–Kiefer–Wolfowitz Inequality (Lemma A.7),

$$\mathbb{P}\left[ \sup_s \left| F_{S|\check{\theta}_n}(s) - \hat{F}_{S|\check{\theta}_n}^{(m)}(s) \right| \ge 2 f_{\min} u \right] \le 2 \exp\left( -8 m f_{\min}^2 u^2 \right).$$

Thus, by Lemma A.6, given that the event $V$ occurs,

$$\mathbb{P}\left[ |\hat{q}_{(1-\alpha)_m} \left( S_m \mid \check{\theta}_n \right) - q_{(1-\alpha)_m} \left( S \mid \check{\theta}_n \right)| \ge u \mid V \right] \le 2 \exp\left( -8 m f_{\min}^2 u^2 \right), \quad u \in [0, \beta/4].$$

Specifically,

$$\mathbb{P}\left[ |\hat{q}_{(1-\alpha)_m} \left( S_m \mid \check{\theta}_n \right) - q_{(1-\alpha)_m} \left( S \mid \check{\theta}_n \right)| \ge \beta/4 \mid V \right] \le 2 \exp\left( -8 m f_{\min}^2 (\beta/4)^2 \right).$$

Then, for any $u > \beta/4$,

$$\mathbb{P}\left[ |\hat{q}_{(1-\alpha)_m} \left( S_m \mid \check{\theta}_n \right) - q_{(1-\alpha)_m} \left( S \mid \check{\theta}_n \right)| \ge u \mid V \right] \le 2 \exp\left( -8 m f_{\min}^2 (\beta/4)^2 \right).$$

Since $|S| \le R$, $|\hat{q}_{(1-\alpha)_m} \left( S_m \mid \check{\theta}_n \right) - q_{(1-\alpha)_m} \left( S \mid \check{\theta}_n \right)| \le 2R$. By the layer cake representation of the expectation of a non-negative random variable $Z$, which is $\mathbb{E}[Z] = \int_0^\infty \mathbb{P}[Z \ge u] \, du$,

$$\mathbb{E}_{\check{\theta}_n} \left[ |\hat{q}_{(1-\alpha)_m} \left( S_m \mid \check{\theta}_n \right) - q_{(1-\alpha)_m} \left( S \mid \check{\theta}_n \right)| \mid V \right]$$

$$= \int_0^{2R} \mathbb{P}\left[ |\hat{q}_{(1-\alpha)_m} \left( S_m \mid \check{\theta}_n \right) - q_{(1-\alpha)_m} \left( S \mid \check{\theta}_n \right)| \ge u \mid V \right] du$$

$$\le \int_0^{\beta/4} 2 \exp\left( -8 m f_{\min}^2 u^2 \right) du + \int_{\beta/4}^{2R} 2 \exp\left( -8 m f_{\min}^2 (\beta/4)^2 \right) du$$

$$\le 2 \int_0^\infty \exp\left( -8 m f_{\min}^2 u^2 \right) du + 4R \exp\left( -8 f_{\min}^2 (\beta/4)^2 m \right)$$

$$= \frac{\sqrt{\pi}}{2 f_{\min} \sqrt{2m}} + 4R \exp\left( -\frac{1}{2} f_{\min}^2 \beta^2 m \right).$$

Therefore, we have

$$\mathbb{E}_{\check{\theta}_n, \mathcal{D}_{\mathrm{cal}}} \left[ |\hat{q}_{(1-\alpha)_m} \left( S_m \mid \check{\theta}_n \right) - q_{(1-\alpha)_m} \left( S \mid \check{\theta}_n \right)| \right]$$

$$\le \mathbb{P}\left[ V \right] \cdot \mathbb{E}_{\check{\theta}_n} \left[ |\hat{q}_{(1-\alpha)_m} \left( S_m \mid \check{\theta}_n \right) - q_{(1-\alpha)_m} \left( S \mid \check{\theta} n \right)| \mid V \right] + \mathbb{P}\left[ V^c \right] \cdot 2R$$

$$\le \frac{\sqrt{\pi}}{2 f_{\min} \sqrt{2m}} + 4R \exp\left( -\frac{f_{\min}^2 \min\{\alpha^2, (1-\alpha)^2\}}{8 f_{\max}^2} m \right) + 2R\delta.$$

Picking $\delta = \frac{257 \lambda_{\max}^2 f_{\max}^3 B^2 d}{\lambda_{\min}^4 f_{\min}^2 \min\{\alpha^2, (1-\alpha)^2\} n}$ completes the proof of Proposition B.4. $\qquad\square$

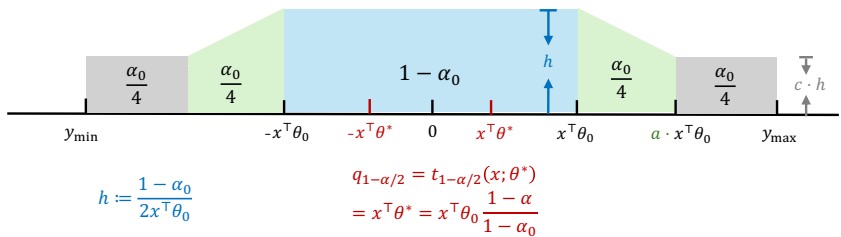

Figure 4: The probability density function of $Y|X = x$ for synthetic dataset.

## C  ADDITIONAL EXPERIMENTS ON SYNTHETIC DATA

### C.1  DATA GENERATION IN SECTION 6

The sampler of the data distribution $\mathcal{P}$ is constructed as follows. A vector $\theta_0$ is first drawn from $\theta_0 \sim \text{Uniform}([1,2]^2)$. The covariate $X$ is sampled uniformly from $\mathcal{X} = [1,20]^2$, i.e., $X \sim \text{Uniform}([1,20]^2)$. Then, the probability density function of the conditional distribution $Y|X = x$ is constructed over support $[y_{\min}, y_{\max}]$, where $y_{\max} = [20,20]^\top \theta_0$ and $y_{\min} = -y_{\max}$. The conditional p.d.f., illustrated in Figure 4, is piecewise affine with five segments, symmetric about zero. The central segment carries probability mass $(1-\alpha_0)$, and each the other four segments carries $\alpha_0/4$, where $\alpha_0 = 0.005$ is chosen to be smaller than the smallest miscoverage level considered in the experiments. The model is well-specified (Assumption 3.1) for $\gamma \in \{\alpha/2, 1 - \alpha/2\}$ and all $\alpha \in (\alpha_0, 1/2)$ by taking $\theta^*(\gamma) = \frac{1-2(1-\gamma)}{1-\alpha_0}\theta_0$, and hence the true quantile functions $t_\gamma(x;\theta^*(\gamma)) = \frac{1-2(1-\gamma)}{1-\alpha_0}\theta_0^\top x$. Then we can draw $y \sim Y|X = x$ from reject sampling to obtain $(x,y)$.

### C.2  VALIDATING REGIME OF $\mathcal{O}(1/(n\alpha^2))$

In the regime where $\alpha = o(n^{-1/4})$ and $\alpha = \omega(n^{-1/2})$, theory predicts that the length deviation should scale as $\mathcal{O}(1/(n\alpha^2))$, corresponding to the middle regime (green) in Figure 2. To validate this dependence, we pick $\alpha$ at several small values $\alpha = \{0.01, 0.02, 0.025, 0.03\}$ and vary the training size $n$, plotting the length deviation against $1/(n\alpha^2)$ on a log–log scale. The fitted regression line (red) in Figure 5 yields a slope of approximately $0.92$, which is close to the theoretical value of $1$. The empirical results support the predicted theoretical scaling, indicating the upper bound accurately captures the observed dependence.

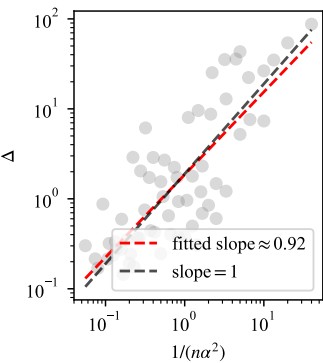

Figure 5: Log–log regression of length deviation $\Delta$ versus $1/(n\alpha^2)$ for relatively small $\alpha$.

### C.3  TRAINING VIA ALTERNATIVE OPTIMIZERS

To demonstrate that our analytical framework extends directly to alternative optimization algorithms by substituting the corresponding estimation error rate, Figure 6 reports the empirical results ob-

tained using *SGD with heavy-ball momentum* (Polyak, 1964). Theoretically, SGD with momentum achieves the same convergence rate as vanilla SGD, up to improved constants. According to Remark 3.3, the efficiency with SGD with momentum scales in the same order as SGD. Consistent with this prediction, the empirical results show that the phase transition phenomenon identified in our analysis persists under SGD with momentum as well. Specifically, in Figure 6 (c), the slope of curves in Figure 6 (a) changes from $-1$ to $-0.5$ as $\alpha$ increases.

Moreover, to demonstrate that our theoretical insights are not tied to optimizers with established convergence guarantee, Figure 7 reports the empirical results obtained using AdamW (Loshchilov & Hutter, 2019). From Figure 7 (c), we observe the phase transition phenomenon, where the slope of curves in Figure 7 (a) changes from $-1$ to $-0.5$ as $\alpha$ increases.

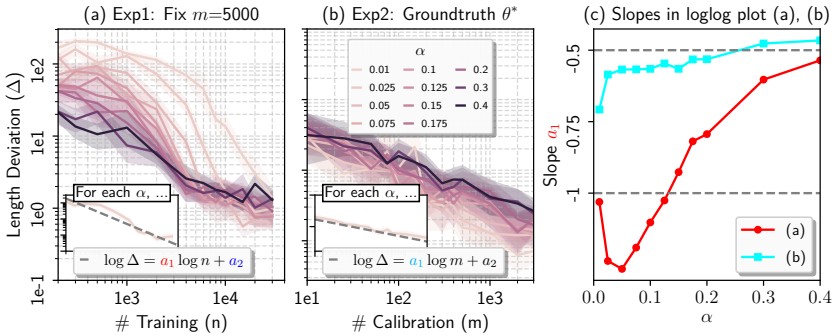

Figure 6: The length deviation of conformalized quantile regression with training via SGD with momentum.

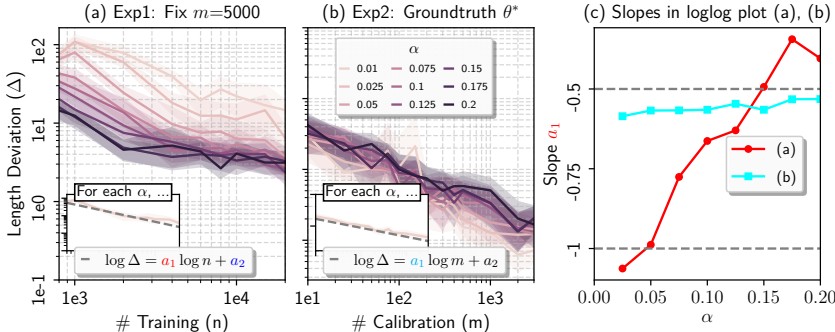

Figure 7: The length deviation of conformalized quantile regression with training via AdamW.

### C.4 NON-LINEAR GROUND-TRUTH QUANTILE FUNCTIONS

To empirically show that our theoretical insights extend beyond linear models, we conducted experiments in a setting where the ground-truth quantile functions are no longer linear. This is achieved by applying Gaussian convolution kernels to the original linear conditional probability density functions, thereby introducing controlled non-linear distortion. As shown in Figure 8, even with this non-linear distortion, the phase transition phenomenon persists, indicating that our theoretical insights remain valid in a broader setting.

### C.5 ALTERNATIVE LOSS FUNCTIONS

To provide empirical evidence that similar efficiency scaling behavior persists for other convex models satisfying our assumptions, we report results using $\ell_1$-regularization during training in Figure 9 and Huber penalty (Huber, 1964) during training in Figure 10. In both cases, the phase transition phenomenon remains clearly visible (the slope of (a) changes from $-1$ to $-0.5$ as $\alpha$ increases), further validating the generality of our theoretical insights.

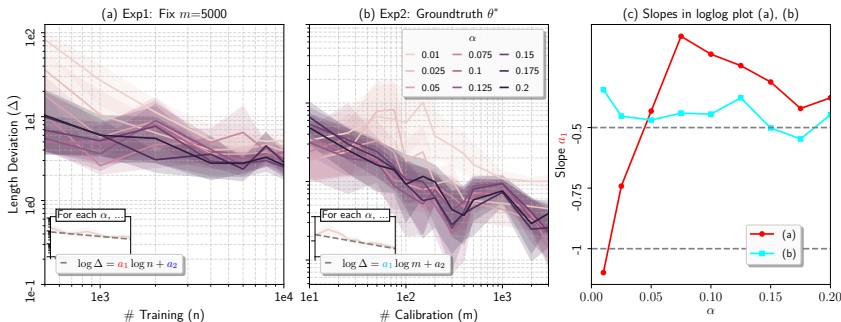

Figure 8: The length deviation of conformalized quantile regression on the data distribution with non-linear conditional quantile functions, where Gaussian convolution kernels are applied to the linear conditional probability density functions. We set $\sigma$ to be $0.1\times$ conditional quantile.

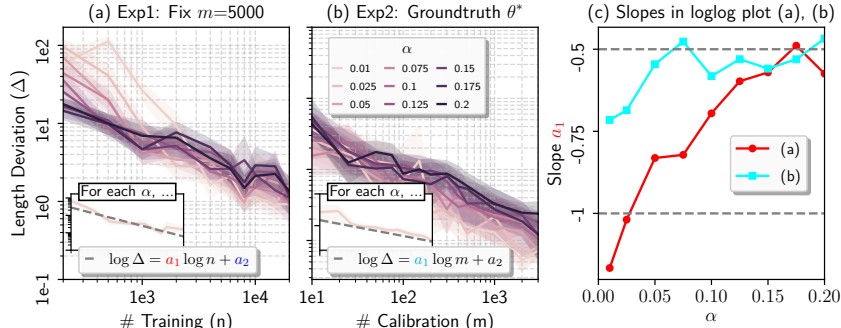

Figure 9: The length deviation of conformalized quantile regression, training with $\ell_1$ regularization. We set the coefficient of the regularization term to be $0.001$.

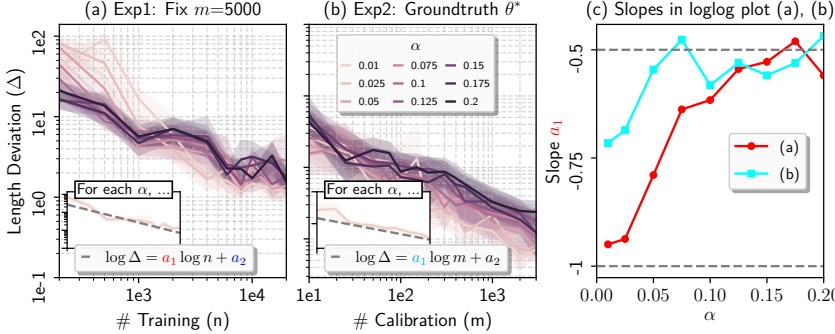

Figure 10: The length deviation of conformalized quantile regression, training with Huber penalty. We set Huber $\delta$ to be $0.5$, and Huber $\lambda$ to be $0.1$.

## D  EXPERIMENTS ON REAL-WORLD DATA

### D.1  STATISTICS OF DATASETS

We list the statistics of multiple popular real world regression datasets used in this paper in Table 1.

The Medical Expenditure Panel Survey (MEPS) Panels 19[1] and 20[2] are standard datasets used for benchmarking and comparative analysis in the quantile regression literature. Each sample consists of 139 features, including 2 categorical features, 4 continuous features, and 133 boolean features.

Table 1: Statistics of datasets.

| Dataset | # Features | # Number Samples |
|---|---|---|
| MEPS 19 | 139 | 15,785 |
| MEPS 20 | 139 | 17,541 |
| cpusmall (Chang & Lin, 2011) | 12 | 8,192 |
| abalone (Chang & Lin, 2011) | 8 | 4,177 |
| California Housing (Pace & Barry, 1997) | 8 | 20,640 |

### D.2  EMPIRICAL EVALUATION OF LENGTH DEVIATION

#### D.2.1  EXPERIMENTAL SETTINGS

We examine the effect of the training set size $n$ and the calibration set size $m$ on the prediction set length, comparing the empirical results with the theoretical bound in Theorem 3.2. Since the oracle quantile interval length $|\mathcal{C}^*(X)| = q_{1-\alpha/2}(Y|X) - q_{\alpha/2}(Y|X)$ depends on $\alpha$, we evaluate the expected absolute deviation $\mathbb{E}[||\mathcal{C}(X)| - |\mathcal{C}^*(X)||]$ for $\alpha \in [0.01, 0.05, 0.1, 0.2]$, where the interval length $|\mathcal{C}^*(X)|$ is approximated by its estimate with same $\alpha$ and largest training and calibration sample sizes. We reserve 20% of the dataset for testing length deviation. The remaining 80% data was partitioned for 80% training data and 20% calibration data: the training size $n$ varied from 10% to 80% in increments of 10%, while the calibration $m$ was chosen from $5\%, 10\%, 15\%, 20\%$ of the remaining data. Throughout experiments, models are trained with a step size tuned by successive halving for 1 epoch.

#### D.2.2  EMPIRICAL RESULTS WITH VARIOUS OPTIMIZERS

Figure 11 presents an empirical evaluation of length deviation of CMR and CQR under different optimizers on real-world datasets, comparing SGD, SGD with momentum (SGDM) (Polyak, 1964), Adam (Kingma & Ba, 2015), and AdamW (Loshchilov & Hutter, 2019). Although our theory is based on analyzing SGD and directly extends to SGD with momentum, we include Adam and AdamW due to their widespread practical use.

The results confirm two key insights from our theoretical analysis. First, increasing the calibration set size $m$ reduces the expected length deviation. Second, for a fixed sample size, a larger miscoverage level $\alpha$ leads to a smaller deviation with lower variance, which aligns with the $\alpha$-dependence in the theoretical rate. Consistent with Theorem 4.1, we observe that smaller values of $\alpha$ yield significantly larger length deviations.

Among the optimizers, we observe that Adam and AdamW generally achieve better efficiency (lower length deviation) but exhibit higher volatility, likely due to their scaled gradient norms. SGD decays more smoothly, providing a more consistent reduction in length deviation as the number of training samples increases. For Adam and AdamW, the benefit of additional training data can be less pronounced or even reversed on certain datasets (e.g., MEPS), where fast convergence makes efficiency more sensitive to stochasticity. On other datasets such as abalone and cpusmall, however, Adam also exhibits a clear decreasing trend, indicating a dataset-dependent behavior.

---

[1]https://meps.ahrq.gov/mepsweb/data_stats/download_data_files_detail.jsp?cboPufNumber=HC-181

[2]https://meps.ahrq.gov/mepsweb/data_stats/download_data_files_detail.jsp?cboPufNumber=HC-192

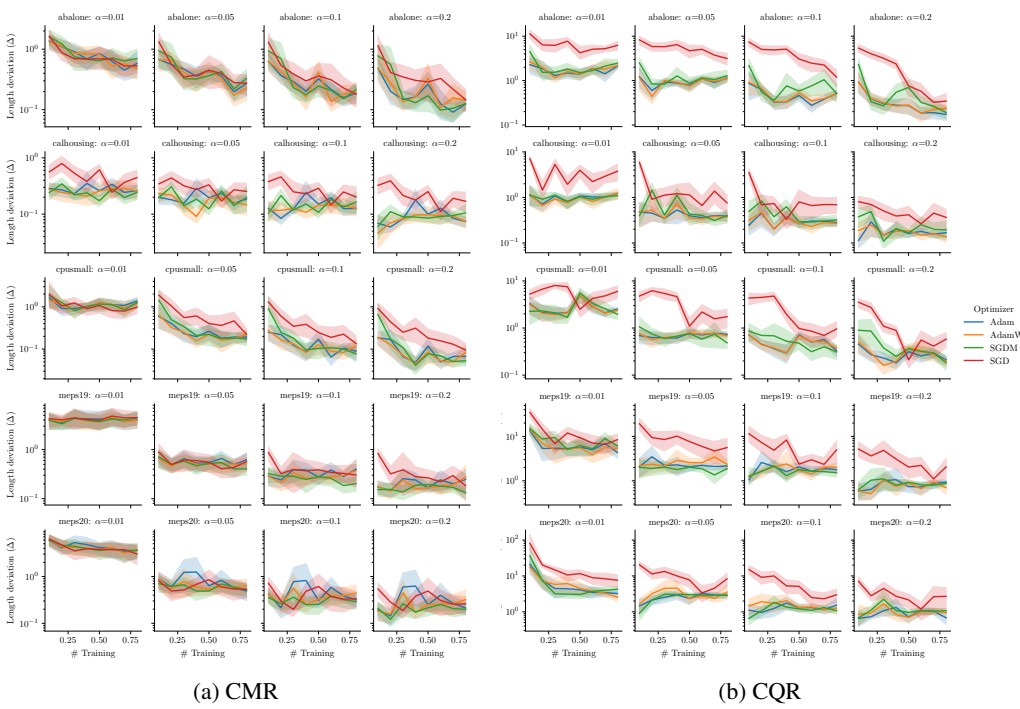

(a) CMR            (b) CQR

Figure 11: **Efficiency of conformalized regression under different training optimizers on real-world datasets**: `MEPS 19`, `MEPS 20`, `California Housing` (Pace & Barry, 1997), `cpusmall`, and `abalone` (Chang & Lin, 2011). For each optimizer, the learning rate is selected via successive halving, while all other hyperparameters (e.g., momentum=0.9 for SGD with momentum) follow the PyTorch defaults.

### D.2.3 EMPIRICAL PROBING ON NON-LINEAR MODELS

We conduct empirical probing of non-linear models on real-world datasets, and report the results in Figure 12. We observe that the length deviation remains consistent across non-linear and linear model architectures, suggesting a potential practical relevance of our findings beyond linear models.

## D.3 EMPIRICAL DATA ALLOCATION GUIDANCE

We empirically investigate how to allocate data on `cpusmall` dataset from LIBSVM (Chang & Lin, 2011). The training ratio $r_n$ takes values from [0.01, 0.05, 0.1, 0.2, 0.3, 0.4, 0.5, 0.6, 0.7, 0.8, 0.9, 0.95, 0.99], the calibration ratio $r_m$ is set as $1 - r_n$, the miscoverage level $\alpha$ takes values from [0.001, 0.002, 0.003, 0.004, 0.005, 0.006, 0.007, 0.008, 0.009, 0.01, 0.02, 0.03, 0.04, 0.05, 0.06, 0.07, 0.08, 0.09, 0.1, 0.15, 0.2].

**The left plot** in Figure 13 shows the length of the prediction interval versus $\alpha$ of CMR, grouping curves by training ratio (0%–20%, 20%–80%, 80%–100%). We observe that two "elbows" occur at approximately $\alpha = 0.045$ and $\alpha = 0.003$, at which points, reducing $\alpha$ leads to a substantially sharper increase in interval length than before. Notably, before the first elbow, e.g., when reducing $\alpha$ from 0.2 to 0.05, the prediction interval length increases only mildly.

**The right plot** in Figure 13 shows the length of the prediction interval versus the training ratio, with each curve corresponding to a different miscoverage level $\alpha$ (lighter color representing smaller $\alpha$). We observe that:

- The curves largely concentrate around interval lengths of approximately 2.5 and 15, respectively, which correspond to the two elbow locations in the left plot.

- For most cases where $\alpha$ is not extremely small, the interval length stays below 15, and the curves exhibit a wide U-shape. This indicates that allocating an excessively large portion of data to either training or calibration tends to degrade efficiency, whereas a more balanced split yields better efficiency. For reasonably large $\alpha$, say $\alpha > 0.04$, the number of calibration samples has less influence on the interval than the number of training samples, suggesting that allocating more data for training is generally beneficial.

- For very small miscoverage levels ($\alpha \leq 0.003$), which correspond to the three curves above the dashed line at length $= 15$, the interval length behaves erratically and no longer follows the U-shaped trend observed for larger $\alpha$. It is likely due to insufficient sample size at such small $\alpha$, and the prediction interval length is a trivial upper bound of the oracle interval length rather than its approximation. This phase of extremely small $\alpha$ may correspond to the regime $\alpha = \omega(n^{-1/2})$, where our upper bound is non-vanishing (Figure 2).

**Takeaway.** The empirical results suggest that practitioners may leverage the elbow points in the left plot of Figure 13 to select $\alpha$ values that yield good efficiency while maintaining reasonable coverage guarantees. In particular, for extremely small $\alpha$, the prediction interval becomes trivially large due to insufficient sample size, whereas in the regime of large $\alpha$, decreasing the miscoverage level results in only a mild increase in interval length. In terms of data allocation, the results are consistent with the practical rule of thumb that the amount of training and calibration data should be roughly of the same order, while allocating slightly more data for training is generally beneficial.

## E THE USE OF LARGE LANGUAGE MODELS (LLMs)

This paper uses a large language model to polish writing.

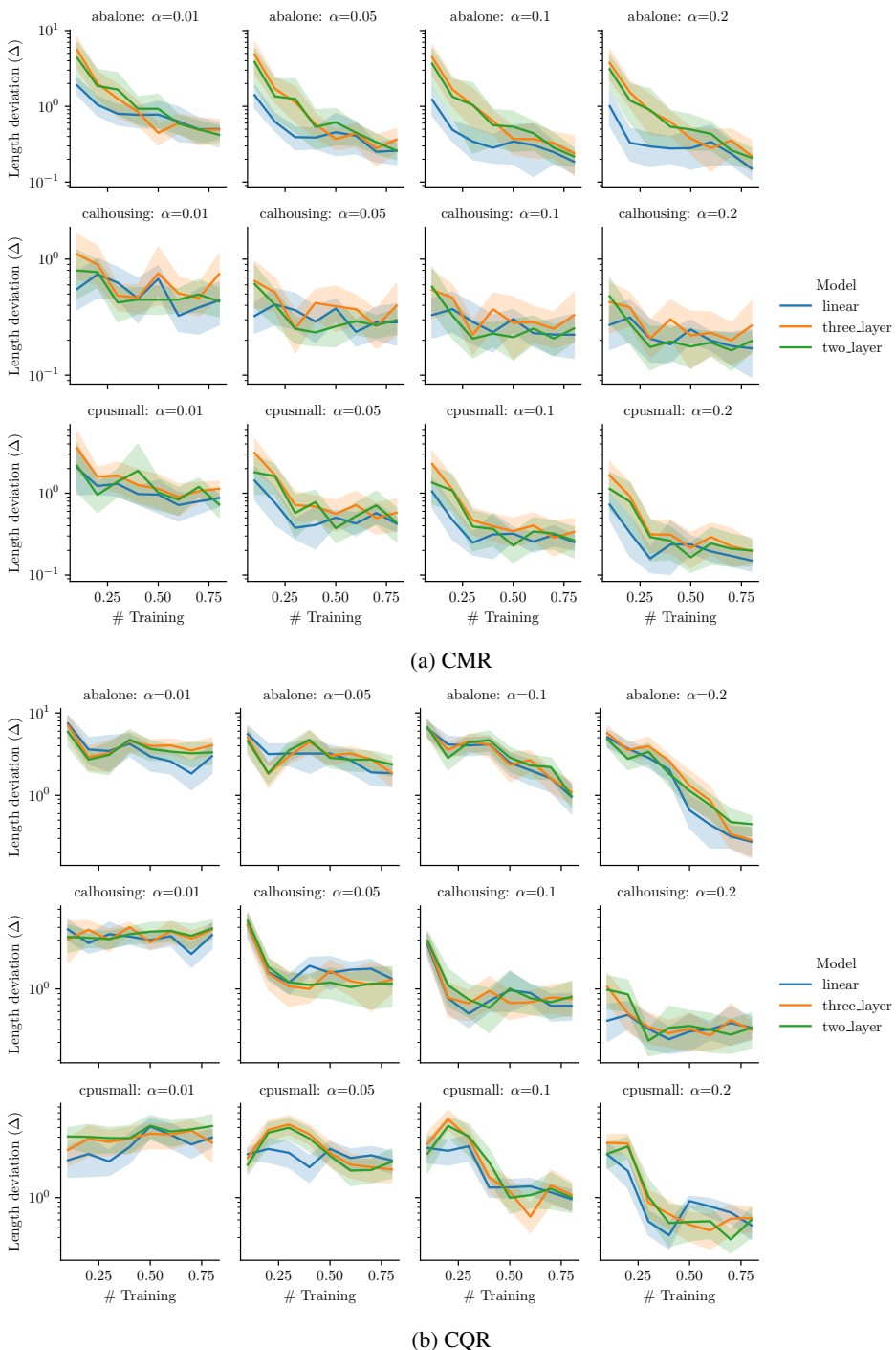

Figure 12: **Efficiency of conformalized regression with linear and non-linear models trained via SGD on real-world datasets**: `MEPS 19`, `MEPS 20`, `California Housing` (Pace & Barry, 1997), `cpusmall`, and `abalone` (Chang & Lin, 2011). The two-layer neural network has one hidden layer with 10 ReLU neurons, and the three-layer network has two hidden layers, each with 10 ReLU neurons.

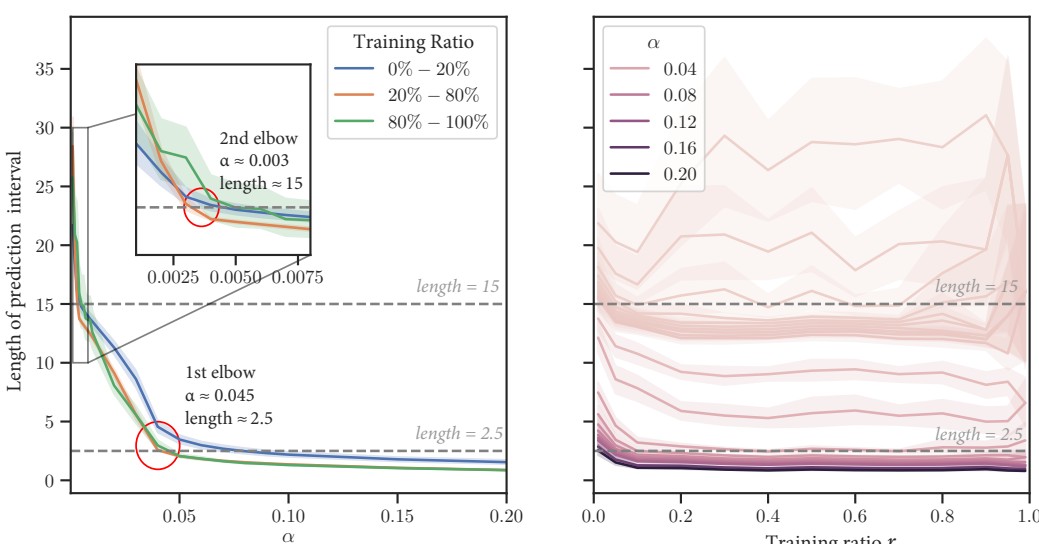

Figure 13: The effect of training ratio $r_n$, calibration ratio $r_m$, and miscoverage level $\alpha$ on `cpusmall` dataset. The training ratio $r_n$ takes values from $0.01$ to $0.99$, the calibration ratio $r_m$ is $1-r_n$, and $\alpha$ takes values from $0.001$ to $0.2$. See Appendix D.3 for detailed discussion on empirical data allocation. **The left plot** shows the length of the prediction interval versus $\alpha$, grouping curves by training ratio (0%–20%, 20%–80%, 80%–100%). We observe that there are two "elbows" around $\alpha = 0.045$ and $\alpha = 0.003$, at which points, reducing $\alpha$ leads to sharper rise of the interval length than before. **The right plot** shows the length of the prediction interval versus the training ratio, with each curve corresponding to a different miscoverage level $\alpha$ (lighter color representing smaller $\alpha$).

