# OpenReview forum: "Non-Asymptotic Analysis of Efficiency in Conformalized Regression"
_ICLR.cc/2026/Conference — ICLR 2026 Poster_

### Official Review · Reviewer_3qyg · 2025-10-19

**Soundness:** 3
**Presentation:** 3
**Contribution:** 2
**Rating:** 4
**Confidence:** 3

**Summary:**

This paper presents a rigorous non-asymptotic analysis of the efficiency of conformalized regression, specifically for Conformalized Quantile Regression (CQR) and Conformalized Median Regression (CMR) trained via Stochastic Gradient Descent (SGD). The core objective is to quantify the efficiency, measured by the expected deviation between the length of the constructed prediction interval and the ideal oracle interval length.

**Contributions**
The main contributions focus on a novel theoretical upper bound for this length deviation. Unlike prior work that often treats the miscoverage level $\alpha$ as a fixed constant, this paper derives the first bound that explicitly captures the joint dependence on the training set size $n$, the calibration set size $m$, and $\alpha$. The derived bound is of the order $O(n^{-1/2} + (\alpha^2n)^{-1} + m^{-1/2} + \exp(-\alpha^2m))$.
This result yields several key insights.

1. It introduces the term $(\alpha^2n)^{-1}$, revealing that efficiency degrades significantly when requiring very high confidence levels (i.e., very small $\alpha$).

2. It uncovers phase transitions in the convergence rates depending on the regime of $\alpha$; for instance, the rate can shift from $O(n^{-1/2})$ to a much slower $O((\alpha^2n)^{-1})$ as $\alpha$ becomes smaller than a threshold related to $n$.
3. These theoretical findings provide concrete guidance for optimally allocating data between training and calibration to achieve a desired level of efficiency. The analysis is supported by empirical results that closely align with the theoretical predictions.

**Strengths:**

*   **Originality:** The paper shows the first non-asymptotic efficiency analysis for conformalized regression that jointly models the impact of training size ($n$), calibration size ($m$), and the miscoverage level ($\alpha$). The identification of the crucial $(\alpha^2n)^{-1}$ term and the resulting "phase transitions" in convergence rates is an original theoretical contribution, moving beyond prior results that typically treated $\alpha$ as a fixed constant.

*   **Quality:** The technical quality of the work is very high. The theoretical analysis is rigorous, providing a complete, end-to-end derivation under clearly stated and relatively mild assumptions on the data distribution. The quality of the empirical validation is exceptional; experiments are designed to isolate and verify specific components and scaling behaviors predicted by the theory, which provides strong evidence for the claims.

*   **Clarity:** The paper is written with outstanding clarity. The problem setup, main theoretical results, and their implications are presented in a logical and accessible manner. The section dedicated to "phase transitions" is particularly effective, translating a complex mathematical bound into intuitive regimes and actionable insights. The figures are well-designed and clearly illustrate the alignment between theory and empirical results.

*   **Significance:** The work carries significant practical and theoretical implications. For practitioners, it offers the first concrete, theory-backed guidance on how to optimally allocate data between training and calibration based on the desired confidence level. For the research community, it fundamentally deepens the understanding of efficiency in conformal prediction, revealing a critical trade-off that was previously not formally understood. The framework is also potentially extensible to other optimizers and models, enhancing its long-term impact.

**Weaknesses:**

*   **Linear Model and Well-Specification Assumption:** The theoretical analysis is developed under the strong assumptions of a linear model class and well-specification. This limits the direct applicability of the derived rates to modern, often overparameterized, and non-linear models, such as neural networks. The paper would be strengthened by a discussion of the challenges in extending the analysis to these more complex settings, perhaps by outlining how model approximation errors would enter the final efficiency bounds.

*   **Focus on Quantile-Based Methods Over Simpler Alternatives:** The analysis centers on CQR/CMR, which requires training quantile regressors via the pinball loss. A widely used and often simpler alternative is to train a single point predictor (e.g., for the conditional mean) and calibrate using the absolute residuals $|Y - \hat{\mu}(X)|$. The paper does not discuss how its theoretical findings, particularly the critical $(\alpha^2n)^{-1}$ term, would apply to this common residual-based method. A comparison would be highly valuable, as the efficiency of the residual method often relies on different assumptions (e.g., homoscedasticity).

*   **Exclusion of Classification Tasks:** The paper's scope is strictly limited to regression, with efficiency measured by interval length. An analogous and equally important problem exists in classification, where efficiency is measured by the size of the prediction set. The work misses an opportunity to discuss how its core insights—especially the critical role of $\alpha$ in determining the required training data, might be applied to the classification setting, where the relationship between model confidence scores and the final set size is a key factor.

*   **Empirical Validation Primarily on Synthetic Data:** The main experiments that provide the most direct validation of the theoretical scaling laws are conducted exclusively on synthetic data. While Appendix includes real-world experiments, they play a minimal role in the main paper and cannot be used to verify the precise rates since the oracle is unknown. Adding an experiment on synthetic data that mildly violates the assumptions (e.g., with slight non-linearity) would provide valuable insights into the robustness of the theory.

**Questions:**

1.  **On the Impact of Model Misspecification:** Your theoretical analysis relies on the assumption of a well-specified linear model. This provides a very clean and insightful result. However, in practice, all models are misspecified. Could you elaborate on how your efficiency bounds might change in the presence of model misspecification? Specifically, if the true conditional quantile function is non-linear, an unavoidable approximation error term arises. How would this error propagate through your analysis? Would it be a simple additive term, or would it interact with the statistical rates you derived in a more complex manner?

2.  **Generalizability to Residual-Based Conformal Methods:** The paper provides a deep analysis of Conformalized Quantile Regression (CQR). A very common and often simpler alternative in practice is to train a single point predictor (e.g., for the conditional mean) and then calibrate using the absolute residuals, i.e., scores are $S_i = |Y_i - \hat{\mu}(X_i)|$. The efficiency of this method is known to be sensitive to heteroscedasticity. Do you hypothesize that your core finding—the critical dependence on $(\alpha^2n)^{-1}$—would also emerge in an efficiency analysis of this simpler residual-based method?

3.  **Implications for Classification:** While your work is focused on regression, the fundamental insight about the trade-off between the required confidence level ($\alpha$) and the training data size ($n$) seems universally important. Could you speculate on how this principle might manifest in the context of conformalized classification, where efficiency is measured by set size? For example, would the analysis involve the steepness of the model's softmax output distribution, and would a smaller $\alpha$ similarly demand much more training data to reliably distinguish between classes with very close scores?

4.  **Tightness of the Bound and Practical Data Allocation:** Your guidance on data allocation is derived by balancing the dominant terms in your theoretical upper bound. This is a powerful and practical contribution. However, since it is based on an upper bound, the tightness of the various terms matters. Could there be situations where the (often unknown) constant factors hidden in the Big-O notation are substantially different for the training and calibration error terms? If so, the truly optimal split might deviate from the one suggested by balancing the rates alone. Could you comment on the potential gap between the "optimal split for the bound" and the "true optimal split" in practice?

5. **Practical Relevance of the $\alpha = O(n^{-1/4})$ Regime:** Your analysis reveals a critical "phase transition" around $\alpha = \Omega(n^{-1/4})$. This is theoretically fascinating. However, in many practical applications, $\alpha$ is a small, fixed constant (e.g., 0.1, 0.05) and does not scale with the sample size $n$. Could you comment on the practical settings where such a scaling of $\alpha$ with $n$ might be relevant? For example, are there high-stakes domains (like autonomous systems or medical diagnostics) where practitioners might demand increasingly higher confidence levels as more data becomes available, thus creating a scenario where these extreme $\alpha$ regimes and their efficiency implications are of practical concern? As I know, there is some literature about extreme error rate cases in conformal prediction. Maybe you should discuss them.

---

> ### Author Response · Authors · 2025-11-23
> **Response to Reviewer 3qyg [Part 1/3]**
>
> We thank Reviewer 3qyg for their effort in reviewing the work, and for the detailed recognition of its originality, technical quality, clarity, and significance. We address each of the comments below.
>
> > W1: **Linear Model and Well-Specification Assumption …**
>
> We thank the reviewer for this comment regarding the linearity and well-specification assumptions. We have now included a dedicated Section 7 in the manuscript discussing the linearity assumption. We kindly note that they are standard in theoretical analyses of this type, as they provide a tractable foundation for deriving non-asymptotic rates and are a necessary first step.
>
> While a full theoretical analysis of misspecification is beyond our scope, its practical effect is captured in our empirical evaluations. As shown in Figure 12 in Appendix E.2.3 and Figure 11 in Appendix E.2.2, CQR and CMR perform robustly even when applied to non-linear models (like neural networks) on various real-world datasets, where strict well-specification is unlikely to hold. Moreover, the experiments in Appendix D.4 on synthetic data show that our theoretical findings extend to data distributions with non-linear quantiles. These results suggest that the insights from our theory have relevance beyond the idealized linear setting.
>
>
> > W2: **Focus on Quantile-Based Methods Over Simpler Alternatives …**
>
> We would like to kindly clarify the misunderstanding here. Our analysis on CMR already covers the case of training a single point predictor and calibrating absolute residuals, as CMR corresponds exactly to using the conditional median as the point estimate. Thus, CMR is a residual-based method in the sense highlighted by the reviewer, and Theorem 4.1 provides a finite-sample efficiency bound of the same flavour as our CQR analysis, in which case we do need a strong homoscedasticity assumption (Assumption 4.2).
>
> > W3: **Exclusion of Classification Tasks …**
>
> We thank the reviewer for pointing out the connection to classification tasks. While classification efficiency is indeed an important and related problem, the non-asymptotic analysis of conformalized classification requires fundamentally different tools. The key objects in the current analysis are the estimation errors of the conditional quantiles and the distribution of the nonconformalty scores. In contrast, classification efficiency is governed by marginal distributions of the data and softmax scores, and does not admit much similarity of conditional quantile analysis. Because the assumptions, notation, and technical arguments differ substantially, combining the two settings would fall outside the scope of a single paper. This may also explain why prior work on efficiency, such as Lei et al. (2018) and Bars & Humbert (2025), focuses exclusively on regression rather than developing a unified treatment.
>
> References:
> - Lei, Jing, et al. "Distribution-free predictive inference for regression." Journal of the American Statistical Association 113.523 (2018): 1094-1111.
> - Bars, Batiste Le, and Pierre Humbert. "On volume minimization in conformal regression." International Conference on Machine Learning (2025).
>
> > W4: **Empirical Validation Primarily on Synthetic Data**: *... Adding an experiment on synthetic data that mildly violates the assumptions …*
>
> We thank the reviewer for this advice. To empirically show that our theoretical insights extend beyond linear models, we have conducted experiments in a setting where the ground-truth quantile functions are no longer linear. This is achieved by applying Gaussian convolution kernels to the original linear conditional probability density functions, thereby introducing controlled non-linear distortion. As shown in Figure 8 in Appendix D.4, even with this non-linear distortion, the phase transition phenomenon persists, indicating that our theoretical insights remain valid in a broader setting.

---

> ### Author Response · Authors · 2025-11-23
> **Response to Reviewer 3qyg [Part 2/3]**
>
> > Q1: **On the Impact of Model Misspecification…**
>
> We refer to the above discussion on W1 in the weakness section.
>
> > Q2: **Generalizability to Residual-Based Conformal Methods…**
>
> We refer to our clarification to W2. We do have established a similar upper bound of this simpler residual-based method (Theorem 4.1).
>
> > Q3: **Implications for Classification: While your work …**
>
> We appreciate the reviewer’s follow-up. Although a formal non-asymptotic theory for conformalized classification lies outside the scope of this work (discussed in our response to W3), a similar qualitative phenomenon is expected: when $\alpha$ becomes very small, the prediction set is expected to expand, possibly becoming not informative.
>
> > Q4: **Tightness of the Bound and Practical Data Allocation**
>
> > Q4.1: *Could there be situations where the (often unknown) constant factors hidden in the Big-O notation are substantially different for the training and calibration error terms?*
>
> We would like to clarify that our non-asymptotic upper bounds come with explicit constants, given in Eq. (41) and Eq. (42). These expressions show that the training error term depends on the density bounds of the data distribution as well as the condition number of the covariance matrix, whereas the calibration error term is governed only by the density bounds. Therefore, when the covariance is not severely ill-conditioned, the constant factors associated with the two dominant terms are not substantially different.
>
> To further verify this, we computed the coefficients on our synthetic data (also asked by Reviewer N2xt in their W2). The coefficient of the training-error term is approximately $16\times$​ the coefficient of the calibration error for CQR, and $9\times$​ for CMR. In this setting, the constants of the two leading terms are of comparable magnitudes, suggesting that balancing the rates provides a practically relevant data-allocation strategy.
>
> > Q4.2: *Could you comment on the potential gap between the "optimal split for the bound" and the "true optimal split" in practice?*
>
> We agree with the reviewer that the optimal split derived from the upper bounds does not necessarily coincide exactly with the true optimal split for every data distribution, since the bounds capture worst-case behavior. Our goal is therefore not to provide the exact optimal split, but a theoretical guidance by identifying the correct scaling laws and the phase transitions of efficiency as a function of $(n, m, \alpha)$. Importantly, the empirical results support the practical relevance of this theoretical guidance. In Figure 3 and the additional results in Appendix D.3,D.4,D.5, the slopes in log-log plots match the theoretical transitions (changing from $−1$ to $−1/2$ for $n$ and approaches $-1/2$ for $m$ as $\alpha$ increases). In Figure 4, we further observe the predicted $O(1/(\alpha^2 n))$ regime.

---

> ### Author Response · Authors · 2025-11-23
> **Response to Reviewer 3qyg [Part 3/3]**
>
> > Q5: **Practical Relevance of the $\alpha=O(n^{-1/4})$ Regime**: *… comment on the practical settings where such a scaling of $\alpha$ with $n$ might be relevant …*
>
> We thank the reviewer for raising this important point about high-stakes scenarios. We would like to offer three clarifications to address this concern.
>
> - **Analysis of Small $\alpha$ Regimes**: Our original submission already analyzed the extreme small-$\alpha$ regime by partitioning the regime of $\alpha$ into scaling regimes relative to the order of $n$ (partitioned by three scaling orders $n^{-1}$, $n^{-1/2}$, $n^{-1/4 }$ in Figure 2). The $O(n^{-1/4})$ regime is the **largest regime** of $\alpha$ for which the upper bound is small. When $\alpha$ becomes extremely small, e.g. below $\omega(n^{-1/2})$, the upper bound is non-vanishing. We have now added an experiment on a real-world dataset (Figure 13, Appendix E.3), which confirms that for extremely small $\alpha$, the resulting prediction intervals can become trivially large. In high-stakes domains with fixed but very small $\alpha$, this implies that collecting more data or adopting strategies such as active learning may be necessary.
>
> - **Interpretation of "scaling" of** $\alpha$ **with** $n$. Our analysis does **not** assume that $\alpha$ is a function of $n$, but rather the *relative magnitude* of the two quantities. Therefore, Our analysis fully applies to the case where $\alpha$ is a fixed small constant while $n$ increases.
>
> - **Relation to extreme-level conformal prediction literature.** The literature on ``extreme error-rate regime” typically studies settings where $(1−\alpha)$ is extremely close to 1 **relative to the calibration sample size $m$**, so that classical conformal prediction yields infinitely wide sets, motivating the use of extreme-value extrapolation (Pasche et al., 2025). This setting is distinct from ours: extreme-level works focus on extrapolating tail behavior to control coverage under extremely high confidence levels, whereas our analysis investigates the efficiency behavior of standard conformalized regression as explicit functions of $(n,m,\alpha)$.
>
> References:
> - Pasche, Olivier C., Henry Lam, and Sebastian Engelke. "Extreme Conformal Prediction: Reliable Intervals for High-Impact Events." arXiv preprint arXiv:2505.08578 (2025).
>
> We thank the reviewer again for their comments, and we have updated the manuscript accordingly. We hope our responses adequately address the reviewer's concerns.

---

> > ### Comment · Reviewer_3qyg · 2025-11-25
> >
> > Thank you for your patient explanation. Most of my questions have been resolved. I will increase the rating.
> >
> > -- There is one point that needs to be explained: in fact, the efficiency analysis is not limited to classification problems, just as you cited as Lei et al. 2018, their team also studied the efficiency of classification problems, based on similar technologies, and proved the optimal classifier and efficiency (Sadinle, Lei, and Wasserman 2019). Though, as you explained, this is a theoretical analysis that can focus on regression tasks, which is already enough.
> >
> > > 1. Lei, Jing, Max G’Sell, Alessandro Rinaldo, Ryan J. Tibshirani, and Larry Wasserman. 2018. “Distribution-Free Predictive Inference for Regression.” Journal of the American Statistical Association 113 (523): 1094–1111.
> >
> > > 2. Sadinle, Mauricio, Jing Lei, and Larry Wasserman. 2019. “Least Ambiguous Set-Valued Classifiers with Bounded Error Levels.” Journal of the American Statistical Association 114 (525): 223–34.

---

> > > ### Author Response · Authors · 2025-11-25
> > > **Response to Reviewer 3qyg**
> > >
> > > We sincerely thank Reviewer 3qyg for their valuable feedback and for increasing the score. We really appreciate the reviewer’s clarification on the techniques for classification problems, which is very helpful for extending the ideas of this paper to classification tasks. We have incorporated it in the related works.

---

### Official Review · Reviewer_oFDM · 2025-10-28

**Soundness:** 3
**Presentation:** 3
**Contribution:** 3
**Rating:** 8
**Confidence:** 3

**Summary:**

This paper investigates the length of prediction intervals obtained from quantile regression models trained via stochastic gradient descent (SGD). The main contribution is a bound on the deviation of the empirical interval length from its true (population) counterpart. The analysis assumes a linear conditional model for $Y|X$, together with regularity and compact support of the underlying densities. While these assumptions may appear restrictive, they are reasonable in the context of deriving precise theoretical guarantees.

**Strengths:**

The main strength of this paper lies in the completeness of its analysis. The authors derive bounds on the expected deviation of the prediction interval length that explicitly account for all key parameters — namely, the size of the training data, the size of the calibration data, and the confidence level.

The paper is clearly written and well structured, leaving a positive overall impression both in the main text and in the detailed appendices and proofs.

**Weaknesses:**

The paper would benefit from a dedicated discussion of its limitations at the end. In particular, while the analysis focuses on the expected length deviation, it would be useful to comment on situations where the oracle interval itself may not be optimal — for instance, in the presence of multimodal conditional distributions. Furthermore, it would be interesting to discuss whether and how the linearity assumption on the conditional distribution Y∣X could be relaxed or extended.

**Questions:**

1. On the oracle interval and multimodality. You control the expected length deviation, but the optimality of the oracle interval is only briefly discussed in Remark 3.5. Under multimodality, the interval defined by quantiles does not coincide with the highest-density regions. A discussion of this limitation would be valuable.

2. On notation and readability. The paper uses numerous notations, some of which could potentially be simplified. In $t_{\gamma}(x,\theta) = x^{\top}\theta$, is the subscript $\gamma$ really necessary? The notation $\overline{\theta}$ and $\underline{\theta}$ already indicates whether the upper or lower quantile is considered. In addition, it might improve clarity to make explicit in Assumption 3.1 that the model is linear by writing, for instance, $X^{\top}\theta(\gamma)$.

3. On definitions and practical details.  Line 90: Is a set-valued definition of quantiles necessary? The standard generalized inverse given in Eq.(2) is well known and likely sufficient for your analysis.
Line 146: Please specify the learning rate used in the proofs of the main theorems. The expression seems to depend on parameters that are probably unknown in practice. What step size do you use empirically?

4. On related literature.The discussion around Theorem 3.1 could be expanded to better situate the work within the existing literature. Several studies investigate quantile regression trained by SGD under linear assumptions (see Shen et al, 2025 and references therein). It would be useful to compare your setting and assumptions (e.g., Assumptions~3.2--3.3) with these works, especially regarding the role of strong convexity, which Shen et al (2025) claims is typically absent.

5. On missing or unclear notation. In Theorem 3.1, the constant $d$ appears for the first time without prior definition. As later sections suggest, $d$ denotes the dimension of the covariates $X$; this should be stated explicitly before the theorem. Similarly, the notation for the $\Sigma$-indexed norm should be defined when it first appears.

6. On quantile crossing and model assumptions.  Remark 3.1 and Proposition B.4 are somewhat confusing. The SGD procedure learns two parameters, $\overline{\theta}$ and $\underline{\theta}$, corresponding to the upper and lower quantiles. Proposition~B.4 shows that, if $n$ is large enough, quantile crossing cannot occur, i.e., the lines $x \mapsto \overline{\theta}^{\top}x$ and $x \mapsto \underline{\theta}^{\top}x$ do not intersect.
Does this imply that the two ground-truth lines are parallel? Does this follow directly from Assumption 3.1, which (as I understand it) corresponds to the linear model $Y = \theta^{\top} X+ $noise?

7. On classical results.  Lemma B.4 is a standard result; see, for instance, Proposition A.25 in [2]. Please include appropriate references and mention when results are classical.

8. On writing and presentation. Throughout the paper (particularly in the appendices), punctuation should be added at the end of equations (commas or periods) so that each displayed equation is integrated into a grammatically correct sentence.

References


 [1] Online Quantile Regression, Shen et al. (2025), arXiv:2402.04602.

 [2] One-Dimensional Empirical Measures, Order Statistics, and Kantorovich Transport Distances, Bobkov & Ledoux (2016).

---

> ### Author Response · Authors · 2025-11-23
> **Response to Reviewer oFDM [Part 1/2]**
>
> We thank Reviewer oFDM for their thorough and positive assessment of our work. We sincerely appreciate their recognition of the paper's analytical completeness and clarity. Their insightful questions have helped us improve the manuscript. Below we address each point in detail.
>
> > W: **Dedicated discussion of limitations**: *… comment on situations where the oracle interval itself may not be optimal … whether and how the linearity assumption could be relaxed or extended*
>
> We thank the reviewer for highlighting this important point. In the revised manuscript, we have added **a dedicated Section 7** of *Limitations, Discussion, and Future Work* on page 10. While some of these aspects were mentioned implicitly throughout the original version, they are now consolidated and discussed more explicitly.
>
> **Oracle intervals may not be optimal in certain distributions.**
>
> We agree with the reviewer that an oracle interval $[q_{\alpha/2}(Y | X),  q_{1-\alpha/2} (Y | X) ]$ is not necessarily efficiency-optimal for certain conditional distributions. This limitation stems from the fact that standard non-conformity scores like CMR and CQR inherently produce a single interval, which cannot efficiently capture such complex distributional structures. We have added a discussion of this important limitation to the manuscript. One way to improve efficiency in these settings is to move beyond fixed score functions and consider parameterized nonconformity scores that adapt to the data. For instance, recent work such as Braun et al. (2025) employs an optimization-driven framework targeting volume minimization to learn the parametrization. This is a promising direction for future research.
>
> **Role and limitations of the linearity assumption.**
>
> The linearity assumption is essential for our theoretical guarantees. This assumption is standard in the theoretical analysis of quantile regression (Koenker, 2005; Pan & Zhou, 2021; Shen et al., 2024), as it ensures convexity of the objective and therefore the consistency of the SGD estimator as the training data size $n$ grows. While relaxing this assumption is in principle possible, it typically requires additional assumptions on the complexity of the function class or on the estimation error bounds, which may be difficult to verify in practice (Bars & Humbert, 2025).
>
> References:
> - Braun, Sacha, et al. "Minimum volume conformal sets for multivariate regression." arXiv preprint arXiv:2503.19068 (2025).
> - Koenker, Roger. Quantile regression. Vol. 38. Cambridge university press, 2005.
> - Pan, Xiaoou, and Wen-Xin Zhou. "Multiplier bootstrap for quantile regression: non-asymptotic theory under random design." Information and Inference: A Journal of the IMA 10.3 (2021): 813-861.
> - Shen, Yinan, Dong Xia, and Wen-Xin Zhou. "Online quantile regression." arXiv preprint arXiv:2402.04602 (2024).
> - Bars, Batiste Le, and Pierre Humbert. "On volume minimization in conformal regression." International Conference on Machine Learning (2025).

---

> ### Author Response · Authors · 2025-11-23
> **Response to Reviewer oFDM [Part 2/2]**
>
> > Q1: *On the oracle interval and multimodality…*
>
> Please see our response to Weakness.
>
> > Q2: *On notation and readability … is the subscript $\gamma$ really necessary? … improve clarity to make linearity explicit in Assumption 3.1 …*
>
> We thank the reviewer for the thoughtful comments on notation. We chose to keep the subscript $\gamma$ in $t_{\gamma}(x, \theta)$ because it explicitly indicates that the function estimates the $\gamma$-quantile of the conditional distribution $Y \mid X$. In some cases, the target quantile level is self-evident (as the reviewer noted), but in others it is not. Keeping the subscript avoids ambiguity in those instances. Moreover, removing $\gamma$ would implicitly strengthen Assumption 3.1 by requiring the conditional quantile functions for **all** quantile levels are linear in $X$, whereas our current setup assumes linearity only for the two specific quantiles of interest.
>
> Following the reviewer’s suggestion, we have added an explicit equation in Assumption 3.1 to clearly state the linearity assumption (Page 4, Line 186).
>
> > Q3.1: *Line 90: Is a set-valued definition of quantiles necessary …*
>
> We thank the reviewer for these helpful comments. The reviewer is correct that the standard generalized inverse in Eq. (2) is sufficient for most readers and is in fact the notion used throughout our analysis. For clarity and simplicity, we have now removed the set-valued definition (Page 2, Line 90).
>
> > Q3.2: *Line 146: Please specify the learning rate used in the proofs …*
>
> Regarding the learning rate, in the proofs of the main theorems, the step size $\mu_k$ at iteration $k$ is specified in Corollary B.1 as $\mu_k = 1 / (\lambda_{\min} f_{\min} k)$, which is the classical choice used to obtain the convergence guarantees following Rakhlin et al. (2012). These constants appear only in the theoretical analysis to derive explicit non-asymptotic bounds and are not required to run the algorithm in practice. This is similar to Lipschitz-smoothness for learning rates in standard SGD theory. Empirically, we can select the learning rate via successive halving or grid search.
>
> References:
> - Rakhlin, Alexander, Ohad Shamir, and Karthik Sridharan. "Making Gradient Descent Optimal for Strongly Convex Stochastic Optimization." International Conference on Machine Learning. 2012.
>
> > Q4: *On related literature … would be useful to compare your setting and assumptions with Shen et al 2025 …*
>
> We thank the reviewer for pointing out this literature. We agree that Shen et al. (2025) does not require global strong convexity. However, they assume conditional density functions are bounded away from zero in a neighborhood of the target quantile. This is very close to ours, except that we impose a global rather than local lower bound on the conditional density. We also would like to emphasize here that they do not consider conformal prediction and its efficiency analysis, which is the core contribution of our work. We have now included this discussion in the revised version. (Page 8 Line 390-391)
>
> References:
> - Shen, Yinan, Dong Xia, and Wen-Xin Zhou. "Online quantile regression." arXiv preprint arXiv:2402.04602 (2024).
>
> > Q5: *On missing or unclear notation …*
>
> We thank the reviewer for the feedback. We kindly clarify that the dimension $d$ is first defined in line 121. However, we are happy to incorporate reviewers’ suggestion and improve clarity and accessibility:
>
> - We have now explicitly restated​ the definition of $d$ in Assumption 3.1  (Page 4, Line 186).
> - We have added the definition of the $\Sigma$-norm when it is first mentioned in the Appendix (Page 15, Proposition B.1, Line 768).
>
> > Q6: *On quantile crossing and model assumptions … Does this imply that the two ground-truth lines are parallel …*
>
> We thank the reviewer for this keen observation. We note that because the covariate space $\mathcal{X}$ is bounded, the ground-truth lower and upper quantile functions cannot cross, even if they are not parallel. We have added a clarification in Remark 3.1 to reflect this point. (Page 4 Line 165-166)
>
> > Q7: *On classical results … include appropriate references …*
>
> We thank the reviewer for pointing out this important reference! While we suspected that a similar result already existed in the literature, we did not find it. We have now cited it and added a clarifying note before Lemma B.4 (Page 22, Line 1144) to acknowledge this prior result.
>
> References:
> - Bobkov, Sergey, and Michel Ledoux. One-dimensional empirical measures, order statistics, and Kantorovich transport distances. Vol. 261. No. 1259. American Mathematical Society, 2019.
>
> > Q8: *On writing and presentation … punctuation should be added at the end of equations …*
>
> We thank the reviewer for this careful reading. We have carefully checked all equations in the main text and appendix and have added the missing punctuation.

---

> > ### Comment · Reviewer_oFDM · 2025-11-24
> >
> > Thank you for your exhaustive answers.
> >
> > A comment regarding Q3.2: Indeed, this stepsize is common for SGD. But the statement of Theorem 3.1 should include the stepsize given in corollary B.1, as the result depends on this hypothesis. For a different stepsize, the result may not be true, right ? (Again, this is common for SGD, but it deserves to be mentioned explicitly in Theorem 3.1). In addition to this modification, can you specify some guideline in your paper with your numerical experiments? I just want to make sure that this will be in the revised version.

---

> > > ### Author Response · Authors · 2025-11-25
> > >
> > > We thank Reviewer oFDM for their valuable feedback. We have explicitly added the required step size in Theorem 3.1 (Page 5, Line 227-228), as suggested. We have also added a new Section 6.1 (Roadmap of Experiments) on page 9 to give a high level description of numerical experiments. We deeply appreciate the constructive comments, which have helped us improve the paper. We hope these additions address the reviewer's concerns.

---

### Official Review · Reviewer_fXxb · 2025-11-01

**Soundness:** 4
**Presentation:** 4
**Contribution:** 3
**Rating:** 6
**Confidence:** 3

**Summary:**

This paper gives finite-sample efficiency bounds for split conformal regression, both CMR and CQR under stochastic gradient descent training. They derive an upper bound on expected deviation of prediction set length from an oracle interval for CQR-SGD. Additionally they derive a non-asymptotic upper bound for homoscedastic tasks within CMR-SGD by leveraging the fact that intervals are symmetric and of constant lengths across inputs. The upper bounds derived are functions of training and calibration set size, and miscoverage level, which allows them to identify phase transition with respect to miscoverage levels. The authors show how to use this to “guide” train and calibration data splitting both theoretically and empirically for a limited regime of model and data types.

**Strengths:**

Main ideas and contributions are presented well and the proofs/the story the authors are telling with the results are clear. The paper combines strong theoretical backing, but also provides some practical implications of this analysis relating to data splits that practitioners can use. To the best of my knowledge, theoretical results differ enough from previous works mostly in the subtleties involving relaxation of assumptions on the score distribution and the fully non-asymptotic nature of their derivations. While other works have presented in depth analysis on expected size of conformal prediction sets and “optimal” data splitting, I think the theoretical backing makes this work novel enough.

**Weaknesses:**

Empirical results are lacking. The paper is absent of thorough and diverse experiments across broad model and data types. While I understand the upper bound derivations rely on standard error rates of *optimizer of choice (i.e. SGD) I think more can be done empirically to show a) robustness of method and b) the extent to which one can relax some assumptions on the SGD error rates or characteristics of the data distribution and still have upper bounds empirically hold. It’s unclear to me the scope of real world data in addition to the model type that their theoretical findings would truly apply to. Further empirical probing of a broader variety of both linear models on more complex data types would make this analysis more convincing.

The authors don’t provide any empirical evidence on the optimizer-agnostic claim. While it makes sense in theory, some small experiments showing this would be a nice addition.

**Questions:**

Your experiments primarily use a small subclass of linear models (i.e. ridge regression) on MEPS data, could you provide empirical evidence that the same alpha-scaling behavior persists for other convex models that satisfy the outlined assumptions?

MEPS are primarily tabular with many bounded/indicator features (likely roughly aligning with your assumptions after some light preprocessing). Could you discuss sensitivity to datasets with genuinely heavy-tailed or unbounded covariates, where boundedness barely holds approximately (at best)?

---

> ### Author Response · Authors · 2025-11-23
> **Response to Reviewer fXxb [Part 1/2]**
>
> We thank Reviewer fXxb for their thorough and positive assessment of our work, and for their recognition of its novelty, clarity, theoretical contribution, and practical significance. Below, we provide point-by-point responses to their valuable comments and clarify the points they raised.
>
>
> > W1: **Empirical results are lacking**: *… showing this would be a nice addition.*
>
> We thank the reviewer for this constructive comment. We now have conducted extensive experiments on various optimizers, data types, models, and datasets.
>
> **Data types with non-linear quantile functions.**
>
> To empirically show that our theoretical insights extend beyond linear models, we conducted experiments in a setting where the ground-truth quantile functions are no longer linear. This is achieved by applying Gaussian convolution kernels to the original linear conditional probability density functions, thereby introducing controlled non-linear distortion. As shown in Figure 8 in Appendix D.4, even with this non-linear distortion, the phase transition phenomenon persists, indicating that our theoretical insights remain valid in a broader setting.
>
> **On non-linear models, alternative datasets and optimizers.**
>
>
> We have conducted additional experiments using two non-linear models: a two-layer and a three-layer perceptron with ReLU activations, evaluated on the `cpusmall` and `abalone` regression datasets from LIBSVM (Chang & Lin, 2011), and `California Housing` dataset (Pace & Barry, 1997). The results, presented in Figure 12 (Appendix E.2.3), demonstrate that the length deviation remains consistent across these different model architectures, suggesting a potential practical relevance of our findings beyond linear models.
>
> We have also conducted additional experiments using alternative real-world datasets. These results are presented in Figure 12 (Appendix E.2.3) and Figure 11 (Appendix E.2.2).
>
> References:
> - Chang, Chih-Chung, and Chih-Jen Lin. "LIBSVM: A library for support vector machines." ACM transactions on intelligent systems and technology (TIST) 2.3 (2011): 1-27.
> - Pace, R. Kelley, and Ronald Barry. "Sparse spatial autoregressions." Statistics & Probability Letters 33.3 (1997): 291-297.
>
> > W2: **On alternative optimizers.**: *The authors don’t provide any empirical evidence on the optimizer-agnostic claim …*
>
> We thank the reviewer for this suggestion. To demonstrate that our analytical framework extends directly to alternative optimization algorithms by substituting the corresponding estimation error rate, Figure 6 in Appendix D.3 reports the empirical results obtained using SGD with heavy-ball momentum (Polyak 1964). Theoretically, SGD with momentum achieves the same convergence rate as vanilla SGD, up to improved constants. According to Remark 3.4, the efficiency with SGD with momentum scales in the same order as SGD. Consistent with this prediction, the empirical results show that the phase transition phenomenon identified in our analysis persists under SGD with momentum as well. Specifically, in Figure 6 (c), the slope of curves in Figure 6 (a) changes from $-1$ to $-0.5$ as $\alpha$ increases.
>
> Moreover, to demonstrate that our theoretical insights are not tied to optimizers with established convergence guarantee, Figure 7 in Appendix D.3 reports the empirical results obtained using AdamW (Loshchilov & Hutter 2019). From Figure 7 (c), we observe the phase transition phenomenon, where the slope of curves in Figure 7 (a) changes from $-1$ to $-0.5$ as $\alpha$ increases.
>
> We believe the behavior of SGD with momentum can help to verify that our theoretical analysis can be directly extended to other optimizers, and the behavior of AdamW can help to support the robustness of the phase transition phenomenon, even for optimizers without a convergence guarantee.
>
> References:
> - Polyak, Boris T. "Some methods of speeding up the convergence of iteration methods." (1964)
> - Loshchilov, Ilya, and Frank Hutter. "Decoupled Weight Decay Regularization." (​2019)

---

> ### Author Response · Authors · 2025-11-23
> **Response to Reviewer fXxb [Part 2/2]**
>
> > Q1: *Your experiments primarily use a small subclass … provide empirical evidence that the same alpha-scaling behavior persists for other convex models …*
>
> We thank the reviewer for this question. To provide empirical evidence that the same efficiency scaling behavior persists for other convex models satisfying our assumptions, in Appendix D.5, we report results using $\ell_1$-regularization during training in Figure 9 and Huber penalty (Huber, 1964) during training in Figure 10. In both cases, the phase transition phenomenon remains clearly visible (the slope of (a) changes from $-1$ to $-0.5$ as $\alpha$ increases), further validating the generality of our theoretical insights.
>
> References:
> - Huber, Peter J. "Robust estimation of a location parameter." Breakthroughs in statistics: Methodology and distribution. New York, NY: Springer New York, 1992. 492-518.
>
> > Q2: *`MEPS` are primarily tabular … sensitivity to datasets with genuinely heavy-tailed or unbounded covariates …?*
>
> We thank the reviewer for this insightful question regarding the sensitivity of our method to genuinely heavy-tailed covariates. The reviewer is correct that `MEPS` features are primarily bounded indicators (Figure 14 in Appendix E.4).
>
> To address this directly, we conducted experiments on the `California Housing` Dataset (Pace & Barry, 1997), which contains features with heavier tails. We systematically identified heavy-tailed features using the Hill estimator (Hill, 1975) to calculate the extreme value index $\xi$. Following the framework of Voitalov et al. (2019), we conservatively classified features with $\xi > 1/4$ as heavy-tailed, identifying three such features (`total_rooms`, `total_bedrooms`, `population`). This classification is supported by both the statistical estimate in Figure 15 (Appendix E.4) and the visual evidence from the QQ plots in Figure 16 (Appendix E.4). We extended this analysis to the `abalone` and `cpusmall` datasets (Figures 17 and 18, Appendix E.4), which exhibit a broader spectrum of tail behavior.
>
> The overall performance across these datasets is shown in the updated Figure 12 (Appendix E.2.3) and Figure 11 (Appendix E.2.2). While it is difficult to precisely isolate the effect of tail heaviness from other dataset properties, the results show that the length deviation in general behaves consistently among datasets, suggesting that CQR and CMR remain robust even in the presence of heavy-tailed features.
>
> From a theoretical perspective, our analysis accounts for this effect: as covariates become more heavy-tailed, the maximum eigenvalue $\lambda_{\max}$ of the covariance matrix increases. This, in turn, inflates the constants in the terms dependent on $n$ in Eq. (41), providing a theoretical explanation for a potential increase in training error.
>
> References:
> - Pace, R. Kelley, and Ronald Barry. "Sparse spatial autoregressions." Statistics & Probability Letters 33.3 (1997): 291-297.
> - Voitalov I, Van Der Hoorn P, Van Der Hofstad R, et al. Scale-free networks well done[J]. Physical Review Research, 2019, 1(3): 033034.
> - Hill B M. A simple general approach to inference about the tail of a distribution[J]. The annals of statistics, 1975: 1163-1174.
>
> We thank the reviewer again for the constructive comments, and we have incorporated the discussion in the updated version.

---

> > ### Comment · Reviewer_fXxb · 2025-11-26
> > **Thanks for replies**
> >
> > Thanks to the authors for address all the issues I raised. I am satisfied by the responses and added experiments and have changed my score accordingly.

---

> > > ### Author Response · Authors · 2025-11-27
> > >
> > > We sincerely thank Reviewer fXxb for their constructive feedback and valuable suggestions, which have helped us improve the paper. We also appreciate the increased score and the positive assessment of our work.

---

### Official Review · Reviewer_N2xt · 2025-11-04

**Soundness:** 3
**Presentation:** 3
**Contribution:** 3
**Rating:** 6
**Confidence:** 4

**Summary:**

This paper studies the expected length of prediction intervals produced by split conformal methods for regression, focusing on conformalized quantile regression (CQR) and conformalized median regression (CMR). Authors examine the case of linear models  trained with SGD and derive a bound on expected deviation between the learned prediction interval length and the oracle length given by the true conditional quantile. Specifically, authors provide non‑asymptotic upper bounds depending on training and calibration set sizes as well as miscoverage level $\alpha$. The analysis assumes bounded covariates and outcome, as well as conditional density bounded above and below.

**Strengths:**

Originality: to my knowledge, the first explicit non-asymptotic bound of this type (depending on train and calibration test sizes and miscoverage).
Quality: explicit versions of the bounds (equations 41 and 42) make the dependence on problem constants transparent. Experiments showcase key theoretical findings.
Clarity: Assumptions are explicit and clear, and key constants are defined.
Significance: finite‑sample results relating dataset splitting and miscoverage level are immediately actionable for conformal prediction practitioners, complementing earlier work on CQR and asymptotic efficiency.

**Weaknesses:**

- Abstract mentioned “offering guidance for allocating data to control excess prediction set length". I assume the Data Allocation and preceding sections cover that. In my opinion, the practitioners would benefit from a more clear and explicit recommendations on choosing sizes/miscoverage level, e.g., starting from a fixed training (or calibration) set size $n$ ($m$) or a required level of $\alpha$. In fact, authors themselves follow similar setups for their own experiments in the Appendix.
- Large constants in equations 41 and 42. Computing them for the presented synthetic data model may improve the understanding and presentation of the bounds.
- Experiments are mainly linear with one real dataset family (MEPS). Including at least one nonlinear (e.g., NN) backbone (even if theory doesn’t directly cover it) would illustrate the practical applicability of the bounds beyond linear models.

**Questions:**

- Figure 6 in the appendix shows a very strange behavior: performance oscillates with increasing training set size. Why is that happening?

- Figure 7 in the appendix is much more stable, but also has problems: length deviation increases with increasing training set size for many $\alpha$ and calibration set sizes. Can you explain this behavior?

---

> ### Author Response · Authors · 2025-11-23
> **Response to Reviewer N2xt [Part 1/2]**
>
> We thank Reviewer N2xt for the thoughtful review and the recognition of our paper's originality, quality, clarity, and significance. We address the comments below.
>
> > W1: **Explicit recommendation on data allocation**. *“the practitioners would benefit from more clear and explicit recommendations on choosing sizes/miscoverage levels …”*
>
> As the reviewer notes, Section 3.2.1 provides high-level guidance for data allocation based on how efficiency scales with $(n, m, \alpha)$. To complement this, we have conducted an experiment and included the results in Figure 13 and discussion in Appendix E.3, demonstrating a practical way for empirically selecting the miscoverage level $\alpha$ and the training sample ratio. The key insight is that practitioners may leverage the elbow points in the left plot of Figure 13 to select $\alpha$ values that yield good efficiency while maintaining reasonable coverage guarantees. In particular, for extremely small $\alpha$, the prediction interval becomes trivially large due to insufficient sample size, whereas in the regime of large $\alpha$, decreasing the miscoverage level results in only a mild increase in interval length. In terms of data allocation, the results are consistent with the practical rule of thumb that the amount of training and calibration data should be roughly of the same order, while allocating slightly more data for training is generally beneficial.
>
>
> > W2: **Large constants in equations 41 and 42**: *Computing them for the presented synthetic data model …*
>
> The reviewer is correct that the constants in our bounds could be large. Large constants are common in theoretical upper bounds, as the rates are usually derived for **worst case scenarios**, necessitating large constants such as condition numbers of the covariance matrix, the density bounds of the data distribution. However, these constants **do not affect the scaling behavior of the key parameters,** which is the primary focus of our non-asymptotic analysis.
>
>
> To illustrate the practical implications of the bounds,  we calculate the constants for the scenario in Figure 3 (seed=0, n=200, m=5000). The constants are:  $B=28.28$, $K=2.83$, $f_{\max}=0.11$, $f_{\min}=2.34\times 10^{-5}$, $\lambda_{\max}= 273.57$, $\lambda_{\min}= 35.70$. Using these values, the relative magnitudes of the coefficients of the leading terms of $\frac{1}{\sqrt{n}}$ and $\frac{1}{\sqrt{m}}$ in Eq. (41) and Eq. (42) are as follows:
>
>
> - In Eq. (41), the coefficient of $\frac{1}{\sqrt{n}}$ is roughly $16.59 \times$ that of $\frac{1}{\sqrt{m}}$;
> - In Eq. (42), the coefficient of $\frac{1}{\sqrt{n}}$ is roughly $9.03 \times$ that of $\frac{1}{\sqrt{m}}$.
>
>
> These results align with the widely held view that, when the training and calibration sample sizes are of comparable order, allocating slightly more data to training generally leads to better efficiency.
>
> > W3: **Experiments are mainly linear with one real dataset family (`MEPS`)**: *Including at least one nonlinear (e.g., NN) backbone (even if theory doesn’t directly cover it) would illustrate the practical applicability …*
>
> We thank the reviewer for the suggestion. We have conducted additional experiments using two non-linear models: a two-layer and a three-layer perceptron with ReLU activations, evaluated on the `cpusmall` and `abalone` regression datasets from LIBSVM (Chang & Lin, 2011), and `California Housing` dataset (Pace & Barry, 1997). The results, presented in Figure 12 (Appendix E.2.3), demonstrate that the length deviation remains consistent across these different model architectures, suggesting a potential practical relevance of our findings beyond linear models.
>
> References:
> - Chang, Chih-Chung, and Chih-Jen Lin. "LIBSVM: A library for support vector machines." ACM transactions on intelligent systems and technology (TIST) 2.3 (2011): 1-27.
> - Pace, R. Kelley, and Ronald Barry. "Sparse spatial autoregressions." Statistics & Probability Letters 33.3 (1997): 291-297.

---

> ### Author Response · Authors · 2025-11-23
> **Response to Reviewer N2xt [Part 2/2]**
>
> > Q1 &  Q2: **Figure 6 … performance oscillates …**, **Figure 7… length deviation increases with increasing training set size …**.
>
> We thank the reviewer for these insightful observations regarding Figures 6 and 7 in the original version. We would like to clarify that **each curve in these figures represents the final length deviation from a grid of independent training runs, not the trajectory of a single run**. To investigate this further, we conducted additional experiments with the following modifications:
>
> - **Randomized Data Splits**:​ We changed from a fixed seed to using different seeds for splitting training and calibration sets (while keeping the test set fixed), which reduces spurious correlations.
> - **Extended Datasets**:​ We included three additional real-world regression datasets (details in Appendix E.1).
> - **Optimizer Comparison**:​ We compared SGD, SGD with momentum (SGDM), Adam, and AdamW. While our theory is based on analyzing SGD and directly extends to SGD with momentum, we included Adam and AdamW due to their widespread practical use (Adam was used in the original Figures 6 and 7), a point also raised by Reviewer fXxb.
>
> The new results are presented in Figure 11 in Appendix E.2.2.
>
> - Addressing original Figure 6 (Oscillations):​ The oscillations are now significantly reduced with randomized train-calibration split. The Adam/AdamW Optimizer generally achieves a better (lower) length deviation but with more volatility due to its scaled gradient norms, while SGD decays more smoothly.
>
>
> - Addressing original Figure 7 (Increasing Deviation):​ The results show that SGD provides a more consistent reduction in length deviation with more data. For Adam/AdamW, the benefit of additional training samples can be less pronounced or even reversed on some datasets (e.g., `MEPS`), as it may converge quickly and be more affected by its inherent stochasticity. However, on other datasets like `abalone` and `cpusmall`  , Adam also shows a clear reduction with more samples, indicating dataset-dependent behavior.
>
> We thank the reviewer again for the insightful comments that helped us improve the paper. We have revised the manuscript to incorporate the new results, and we hope these additions adequately address the reviewer’s concerns.

---

### Author Response · Authors · 2025-11-29

Dear AC,

We would like to briefly summarize our rebuttal progress.

In response to the comments from reviewers, we have conducted additional experiments and clarified multiple technical details accordingly. So far, three reviewers (oFDM, 3qyg, and fXxb) have explicitly acknowledged that their concerns had been addressed. Among them, two reviewers increased their scores:
- **Reviewer 3qyg** increased the score **from 4 to 8 at 19:32 on 24 Nov 2025 (EST)**.
- **Reviewer fXxb** increased the score **from 6 to 8 at 14:15 on 26 Nov 2025 (EST)**.

For completeness, the reviewers’ ratings at the time were:
- **Reviewer 3qyg:** Rating: 8 / Confidence: 3
- **Reviewer oFDM:** Rating: 8 / Confidence: 3
- **Reviewer fXxb:** Rating: 8 / Confidence: 3
- **Reviewer N2xt:** Rating: 6 / Confidence: 4 (not replied yet)


We fully understand that AC has discretion in making decisions, and we provide the above information only for reference.

If any additional clarification would be helpful, we would be more than happy to provide it!

Authors of Submission 7737

---

### Meta-Review · Area_Chair_7WHh · 2026-01-06

**Summary:**

This paper provides finite-sample bounds on how much the length of split-conformal prediction intervals from conformalized quantile/median regression (CQR/CMR) trained via SGD can exceed the oracle interval length.
The bound scales as $O\left(\frac{1}{\sqrt{n}}+\frac{1}{\alpha^{2}n}+\frac{1}{\sqrt{m}}+e^{-\alpha^{2}m}\right)$, making explicit the joint role of training size (n), calibration size (m), and miscoverage level ($\alpha$), and it reveals ($\alpha$)-dependent phase transitions that guide how to allocate data to control excess interval length.

Reviewers’ main concerns are around (i) scope of the theoretical setting and assumptions, (ii) practical interpretability of the bounds, and (iii) empirical breadth/robustness.
Multiple reviewers noted that the strongest formal guarantees are derived for linear, well-specified quantile/median models under boundedness and conditional density regularity conditions (and, for the median/residual-style result, an additional homoscedastic-type assumption), leaving uncertainty about how directly the theory transfers to common misspecified and nonlinear regimes.
On practicality, reviewers requested clearer “how-to” guidance on train–calibration allocation and alpha selection, and highlighted that constants in the explicit bounds can be large, as well as the need for theorem statements to transparently specify algorithmic requirements (e.g., stepsize conditions).
On the empirics, reviewers highlighted that the original experiments focused on a relatively narrow set of models/datasets, requested broader validation (including nonlinear backbones, additional datasets, and optimizer sensitivity), and questioned the irregular behaviors observed in some appendix plots (oscillations and non-monotone trends with increasing training size).

These issues primarily affected confidence in the real-world generality and the tightness/actionability of the recommendations derived from worst-case bounds, rather than the novelty of the core non-asymptotic dependence on (n, m, alpha).

**Reviewer Concerns:**

Addressed:
- Empirical breadth: Added experiments beyond the original setting, including nonlinear models and additional real datasets, plus explicit optimizer comparisons (SGD / momentum variants and adaptive methods). This directly responds to concerns that the evidence was too narrow and that optimizer sensitivity was unclear.
- “Strange” empirical behaviors: Clarified that earlier plots aggregated independent runs and strengthened experimental setting (e.g., randomized splits), reporting updated results that reduce oscillations and explain dataset/optimizer-dependent non-monotonicity.
- Actionable guidance and presentation: Added clearer discussion/roadmap for experiments and more explicit guidance on data allocation and alpha selection, addressing requests for practitioner-oriented recommendations.

Partially resolved:
- Practical tightness of worst-case constants: Although constants were discussed/illustrated, the bounds may still be conservative; guidance derived by balancing dominant terms can deviate from truly optimal splitting when constants differ substantially across training vs calibration terms.

**Reviewer Scores:**

N2xt: no changes.

fXxb: +2 to 8.

oFDM: no changes.

3qyg: +2 to 6.

---

### Decision · Program_Chairs · 2026-01-26

Accept (Poster)